# CAN LLMs GENERATE NOVEL RESEARCH IDEAS?
## A LARGE-SCALE HUMAN STUDY WITH 100+ NLP RESEARCHERS

**Chenglei Si, Diyi Yang, Tatsunori Hashimoto**
Stanford University
`{clsi,diyiy,thashim}@stanford.edu`

### ABSTRACT

Recent advancements in large language models (LLMs) have sparked optimism about their potential to accelerate scientific discovery, with a growing number of works proposing research agents that autonomously generate and validate new ideas. Despite this, no evaluations have shown that LLM systems can take the very first step of producing novel, expert-level ideas, let alone perform the entire research process. We address this by establishing an experimental design that evaluates research idea generation while controlling for confounders and performs the first comparison between expert NLP researchers and an LLM ideation agent. By recruiting over 100 NLP researchers to write novel ideas and blind reviews of both LLM and human ideas, we obtain the first statistically significant conclusion on current LLM capabilities for research ideation: we find LLM-generated ideas are judged as more novel ($p < 0.05$) than human expert ideas while being judged slightly weaker on feasibility. Studying our agent baselines closely, we identify open problems in building and evaluating research agents, including failures of LLM self-evaluation and their lack of diversity in generation. [1]

## 1 INTRODUCTION

The rapid improvement of LLMs, especially in capabilities like knowledge and reasoning, has enabled many new applications in scientific tasks, such as solving challenging mathematical problems (Trinh et al., 2024), assisting scientists in writing proofs (Collins et al., 2024), retrieving related works (Ajith et al., 2024; Press et al., 2024), and generating code to solve analytical or computational tasks (Huang et al., 2024; Tian et al., 2024). While these are useful applications that can potentially increase the productivity of researchers, it remains an open question whether LLMs can take on the more creative and challenging parts of the research process.

We focus on this problem of measuring the *research ideation* capabilities of LLMs and ask: are current LLMs capable of generating novel ideas that are comparable to expert humans? Although ideation is only one part of the research process, this is a key question to answer, as it is the very first step to the scientific research process and serves as a litmus test for the possibility of autonomous research agents that create their own ideas. Evaluating expert-level capabilities of LLM systems is challenging (Bakhtin et al., 2022; Collins et al., 2024), and research ideation takes this to an extreme. Qualified expert researchers are difficult to recruit at scale, evaluation criteria can be highly subjective, and it is difficult even for experts to judge the quality of research ideas (Beygelzimer et al., 2021).

We address these challenges directly, recognizing that for important, high-stakes tasks like research ideation, there is no substitute for a large-scale expert evaluation. We design a carefully controlled comparison of human and LLM ideas that overcomes sample size and baseline problems present in earlier small-scale evaluation studies. Our study recruited a large pool of over 100 highly qualified NLP researchers to produce human baseline ideas and perform blind reviews of human and LLM ideas. To reduce the possibility that confounding variables affect our outcome measures, we enforce strict controls that standardize the styles of human and LLM ideas and match their topic distribution.

We compare our human expert baseline with a simple and effective LLM agent that incorporates retrieval augmentation and adopts recent ideas in inference-time scaling, such as overgenerating and

---

[1] We release our agent implementation and all human review scores at: `https://github.com/NoviScl/AI-Researcher`. The last two authors advised this project equally.

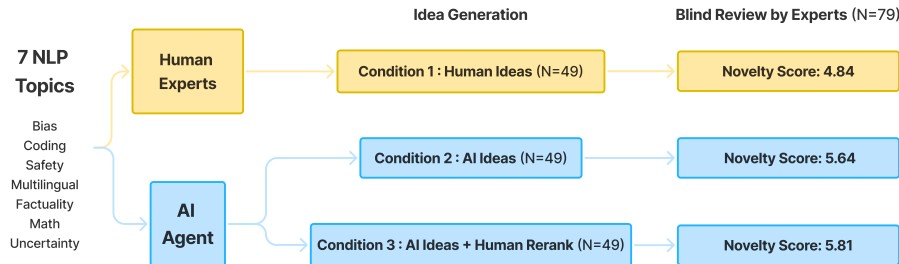

Figure 1: Overview: we recruit 79 expert researchers to perform blind review of 49 ideas from each of the three conditions: expert-written ideas, AI-generated ideas, and AI-generated ideas reranked by a human expert. We standardize the format and style of ideas from all conditions before the blind review. We find AI ideas are judged as significantly more novel than human ideas ($p < 0.05$).

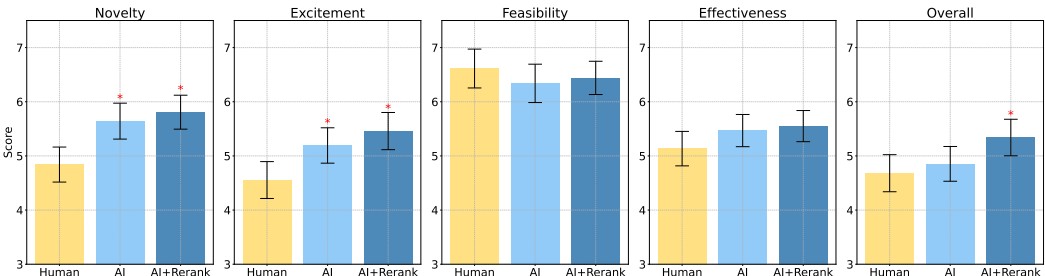

Figure 2: Comparison of the three experiment conditions across all review metrics. Red asterisks indicate that the condition is statistically better than the `Human` baseline with two-tailed Welch's t-tests and Bonferroni correction. All scores are on a 1 to 10 scale. More detailed results are in Section 5.

reranking LM outputs. These measures allow us to make statistically rigorous comparisons between human experts and state-of-the-art LLMs (Figure 1).

Our evaluation-centric approach complements many recent methods-centric works that attempt to instantiate research agents. These works rely on fast and lower-cost evaluation surrogates – either by decreasing the number of expert reviewers (Baek et al., 2024; Li et al., 2024; Wang et al., 2024; Yang et al., 2024), constraining the length and detailedness of the ideas (Wang et al., 2024; Yang et al., 2024), or relying on LLM-as-a-judge (Lu et al., 2024). They do not perform the large-scale human comparison studies that are needed to answer the motivating question of our work. Our work takes the opposite approach, performing a year-long and high-cost evaluation that provides human expert baselines and a standardized evaluation protocol to serve as a foundation for future follow-up studies and methods work.

Through nearly 300 reviews across all our conditions, we find that AI-generated ideas are judged as more novel than human expert ideas ($p < 0.05$), which holds robustly under multiple hypothesis correction and across different statistical tests (Figure 2). Apart from evaluating the ideas, we also analyze the LLM agent, showing limitations and open problems – despite excitement about inference-time scaling of LLMs, we find that they lack idea diversity when we scale up idea generation, and they cannot currently serve as reliable evaluators.

## 2 PROBLEM SETUP

The central experiment of our work is a comparison of human- and LLM-generated ideas. While this goal is simple, there is no existing consensus on how to formulate the task of research ideation and evaluation, and we begin by defining the key aspects of our experiment design.

We think of research idea evaluation as consisting of three separate components: 1). the idea itself, generated in response to our instructions, 2). the writeup which communicates the idea, and 3). the evaluation of the writeup by experts. We outline our experiment design in each of these three parts with particular focus on potential confounders, such as the area of research, the format of a research idea, and the evaluation process.

**Ideation Scope and Instructions**    Any experiment on ideation must carefully balance the realisticness and interestingness of a research idea with the practical realities of eliciting ideas from a large population. In our case, these tradeoffs are even more pronounced, as we have designed our ideation experiments so that the resulting ideas can be executed by experts in a follow-up set of experiments.

These constraints have led us to study prompting-based NLP research as a testbed for our study. Prompting research has been popular in recent years of NLP and AI research Schulhoff et al. (2024). This class of projects strikes a reasonable trade-off among our constraints. The most impactful prompting projects like chain-of-thought have had a major influence on LLM performance (Wei et al., 2022), and prompting projects are executable with minimal computing hardware.

We further structure our ideation process to avoid selection-bias-based confounders in ideation. If we simply ask LLMs and humans to produce ideas on 'prompting topics', we may find that LLMs and humans differ in the types of research ideas they produce (for example, LLMs may naturally suggest more projects on safer topics, which might be judged as less exciting by humans). This would lead us to simply measure misalignment in research topic preference between LLMs and humans, which is not the goal of our study. To address this possibility, we define a set of seven specific research topics extracted from the Call For Papers page of recent NLP conferences such as COLM. Specifically, our topics include: Bias, Coding, Safety, Multilinguality, Factuality, Math, and Uncertainty (see Appendix A.3 for a complete description of these topics).

Each human and LLM participant of the ideation experiment receives the same set of natural language instructions including the same topic description, idea template, and demonstration example to ensure a fair comparison. For human participants, we additionally allow them to select a preferred topic from the list, and for each selected topic, we generate a corresponding LLM idea. This exactly matches the idea topic distribution between the LLM and human participants, while ensuring that human experts are able to select topics according to their expertise.

**Idea Writeup**    An idea can only be evaluated if it is written up to be communicated, but this writing process introduces many additional potential confounders. Human researchers may write in ways that subtly signal quality research, such as including more examples and implementation details. The format of the writeup functions as a way to scaffold what contents should be included and the level of detailedness. Ideally, we want both human and LLM participants to provide all the necessary implementation details for their generated ideas.

We take inspiration from guidelines used in grant submissions and introduce a template to specify the structure and detailedness of idea proposals. Specifically, we construct a template that includes fields for the title, problem statement, motivation, proposed method, step-by-step experiment plan, test case examples, and the fallback plan. Both the LLM agent and the human idea writers are instructed to follow this template and our provided demonstration examples to produce a project proposal as the output (see Appendix A.4 for the full template and Appendix A.5 for the demo example).

Even with these templates, there may be subtle writing style cues that affect the outcome measure. For example, humans may tend to write in a more engaging and informal tone. To reduce this possibility further, we developed a style normalization module that uses an LLM to convert all ideas into the same writing and formatting style without changing the original content. Our small-scale human study shows that such a normalization approach leads to a 50% accuracy for expert human judges who are asked to distinguish AI ideas from human ideas. Finally, the use of an LLM style anonymizer has the possibility of substantively changing the content of the ideas. To rule this out, the first author of this paper manually verified each human idea proposal to ensure all contents of the original ideas were preserved. We present the full prompt used in Appendix A.6.

**Review and Evaluation**    Reviewing research ideas is notoriously subjective, so we want to design a review form that defines all review criteria clearly to standardize and anchor the evaluations as much

as possible. At the same time, we want our review criteria and measured variables to capture all the desiderata of high-quality research ideas.

We follow best practices from AI conference reviewing (e.g., ICLR and ACL) when designing the review form, where we define four breakdown metrics including novelty, excitement, feasibility, and expected effectiveness, apart from the overall score. For each metric, we ask for a numerical score on a 1-10 scale along with a free-text rationale. We provide clear definitions and grounding for each numerical scale to calibrate all reviewers' standards (see Appendix A.7 for the full review form). In the next two sections, we instantiate how our LLM agent generates ideas and how our expert participants generate and review the ideas.

## 3 IDEA GENERATION AGENT

We build a simple but effective LLM ideation agent to compare with the human expert baseline. Rather than focusing on innovating the agent itself, we adhere to a minimalist design principle, aiming to understand the current capabilities of LLMs in idea generation. Our research ideation agent has three essential components: paper retrieval, idea generation, and idea ranking, which we will describe in detail below.

### 3.1 PAPER RETRIEVAL FOR RAG

To ground idea generation, the agent needs to retrieve papers related to the given research topic, so that it will be aware of related works when generating new ideas. To do so, we leverage retrieval-augmented generation (RAG), which has demonstrated effectiveness on many knowledge-intensive tasks (Lewis et al., 2020; Shi et al., 2024). Concretely, given a research topic (e.g., "novel prompting methods that can improve factuality and reduce hallucination of large language models"), we prompt an LLM to generate a sequence of function calls to the Semantic Scholar API. We use `claude-3-5-sonnet-20240620` as the backbone model for our agent but the pipeline should generalize to other LLMs as well. The paper retrieval action space includes: {`KeywordQuery(keywords)`, `PaperQuery(paperId)`, `GetReferences(paperId)`}. Each action generation is grounded on the previous actions and executed results. We keep the top $k = 20$ papers from each executed function call and stop the action generation when a max of $N = 120$ papers have been retrieved. We then use the LLM to score and rerank all retrieved papers based on three criteria: 1) the paper should be directly relevant to the specified topic; 2) the paper should be an empirical paper involving computational experiments; 3) the paper is interesting and can inspire new projects. The LLM is prompted to score each retrieved paper on a scale of 1 to 10 based on these criteria and we use the top-ranked papers for the next step of idea generation.

### 3.2 IDEA GENERATION

Our key insight for idea generation is to generate as many candidate ideas as possible. Our intuition is that only a small fraction of all generated ideas might be high-quality, and we should be willing to expend inference-time compute to generate more candidates so that we can later use a reranker to discover the "diamond in the rough". This aligns with existing results showing that scaling inference compute with repeated sampling can boost LLM performance on various coding and reasoning tasks (Li et al., 2022; Brown et al., 2024). Specifically, we prompt the LLM to generate 4000 seed ideas on each research topic. The idea generation prompt includes the demonstration examples and the retrieved papers. We craft $k = 6$ demonstration examples by manually summarizing exemplar papers (Yasunaga et al., 2024; Madaan et al., 2023; Weller et al., 2023; Weston & Sukhbaatar, 2023; Zheng et al., 2024; Dhuliawala et al., 2023) into our desired idea format. For retrieval augmentation, we randomly select $k = 10$ papers from the top-ranked retrieved papers and concatenate their titles and abstracts to prepend to the idea generation prompt. We also append the titles of all previously generated ideas to the prompt to explicitly ask the LLM to avoid repetitions.

To remove duplicated ideas from this large pool of candidate ideas, we first perform a round of deduplication by encoding all seed ideas with `all-MiniLM-L6-v2` from Sentence-Transformers (Reimers & Gurevych, 2020) and then computing pairwise cosine similarities. We set a similarity threshold of

0.8 for the idea deduplication based on manual inspection. [2] This leaves about 5% non-duplicated ideas out of all the generated seed ideas. We expand more on this duplication issue later in Section 7.1.

## 3.3 IDEA RANKING

The next step is for our ideation agent to rank all the remaining ideas so that we can find the best ones among them. To build such an automatic idea ranker, we use public review data as a proxy. Specifically, we scraped 1200 ICLR 2024 submissions related to LLMs (with keyword filtering) along with their review scores and acceptance decisions. We explored multiple ways of predicting the scores and decisions of these submissions and found that LLMs are poorly calibrated when asked directly to predict the final scores or decisions, but can achieve non-trivial accuracy when asked to judge which paper is better in pairwise comparisons.

We converted the ICLR submissions into our standard project proposal format and randomly paired up accepted and rejected papers and asked LLMs to predict which one is accepted. On this task, `Claude-3.5-Sonnet` achieves an accuracy of 71.4% with zero-shot prompting. For comparison, `GPT-4o` achieves 61.1% and `Claude-3-Opus` achieves 63.5%, and we do not observe significant gains from additional prompting techniques like few-shot or chain-of-thought prompting. We therefore choose the `Claude-3.5-Sonnet` zero-shot ranker.

| $N$ | Top-10 | Bottom-10 | Gap |
|---|---|---|---|
| 1 | 6.28 | 5.72 | 0.56 |
| 2 | 6.14 | 5.24 | 0.90 |
| 3 | 5.83 | 4.86 | 0.97 |
| 4 | 5.94 | 4.99 | 0.95 |
| 5 | 6.42 | 4.69 | 1.73 |
| 6 | 6.11 | 4.81 | 1.30 |

Table 1: Average ICLR review scores of top- and bottom-10 papers ranked by our LLM ranker, with different rounds ($N$) of pairwise comparisons.

In order to obtain reliable scores for all project proposals based on pairwise comparisons, we adopt a Swiss system tournament where all project proposals are paired with those whose accumulated scores are similar, and if the proposals are judged to be better, they gain an additional point. We repeat this for $N$ rounds so the total score of each project proposal will be within the $[0, N]$ range. As a sanity check, we use the `Claude-3.5-Sonnet` ranker to rank the 1.2K ICLR LLM-related submissions and compare the average review scores of the top 10 ranked papers and the bottom 10 ranked papers in Table 1. We see a clear separation between the top and bottom ranked papers, indicating the effectiveness of the LLM ranker. We choose $N = 5$ for all our experiments since it gives the best ranking result on this validation set. The top-ranked project proposals from the agent will be directly used for the `AI Ideas` condition of the human study.

Since our AI ranker is still far from perfect, we also introduce another experiment condition where the first author of this paper manually reranked the generated project proposals instead of relying on the LLM ranker, and we call this the `AI Ideas + Human Rerank` condition. 17 out of the 49 ideas in the `AI Ideas + Human Rerank` condition overlap with the `AI Ideas` ranked by the LLM agent (Table 8 in Appendix A.11), while the other 32 are different, indicating the discrepancy between the LLM ranker and the human expert reranking.

## 4 EXPERT IDEA WRITING AND REVIEWING

In this section, we shift focus to the human branch of idea generation comparison. We present the details of our human study, including information about the recruited experts, the human idea generation task, and the subsequent review process.

## 4.1 EXPERT RECRUITMENT

We recruit our expert participants (including for idea writing and reviewing) by sending sign-up forms to several channels, including: 1) the OpenNLP Slack channel with 1426 NLP researchers from 71 institutions; 2) Twitter (X); 3) Slack channels of various NLP groups by direct communication with the group members; and 4) official chat app of the NAACL 2024 conference. Our study including all recruitment materials has been approved by IRB.

---

[2]We provide randomly sampled idea pairs and their similarities in Appendix A.10. We also provide additional implementation details about the ideation agent in Appendix A.8.

| | Idea Writing Participants (N=49) | | | | | Idea Reviewing Participants (N=79) | | | | |
|---|---|---|---|---|---|---|---|---|---|---|
| Metric | Mean | Median | Min | Max | SD | Mean | Median | Min | Max | SD |
| papers | 12 | 10 | 2 | 52 | 9 | 15 | 13 | 2 | 52 | 10 |
| citations | 477 | 125 | 2 | 4553 | 861 | 635 | 327 | 0 | 7276 | 989 |
| h-index | 5 | 4 | 1 | 21 | 4 | 7 | 7 | 0 | 21 | 4 |
| i10-index | 5 | 4 | 0 | 32 | 6 | 7 | 5 | 0 | 32 | 6 |

Table 2: Research profile metrics of the idea writing and reviewing participants. Data are extracted from Google Scholar at the time of idea or review submission.

| Metric | Mean | Median | Min | Max | SD |
|---|---|---|---|---|---|
| `Human` Ideas | | | | | |
| Familiarity (1-5) | 3.7 | 4.0 | 1.0 | 5.0 | 1.0 |
| Difficulty (1-5) | 3.0 | 3.0 | 1.0 | 5.0 | 0.7 |
| Time (Hours) | 5.5 | 5.0 | 2.0 | 15.0 | 2.7 |
| Length (Words) | 901.7 | 876.0 | 444.0 | 1704.0 | 253.5 |
| `AI` Ideas | | | | | |
| Length (Words) | 1186.3 | 1158.0 | 706.0 | 1745.0 | 233.7 |
| `AI + Human Rerank` Ideas | | | | | |
| Length (Words) | 1174.0 | 1166.0 | 706.0 | 1708.0 | 211.0 |

Table 3: Statistics of the 49 ideas from each condition.

We performed screening on the participants based on their provided Google Scholar profiles and recruited $N = 49$ experts for writing ideas, and $N = 79$ experts for reviewing ideas. Each idea writer is asked to write one idea within 10 days and we compensate $300 for each, with a $1000 bonus for the top 5 ideas as scored by the expert reviewers. Each idea reviewer is assigned 2 to 7 ideas to review and we collected $N = 298$ unique reviews in total. They are given one week to finish the reviews and we compensated $25 for each review written by the idea reviewers.

## 4.2 EXPERT QUALIFICATIONS

Our pool of participants is highly qualified and diverse. The 49 idea writers come from 26 different institutions and 73% of them are current PhD students. The 79 reviewers come from 32 institutions and 87% of them are PhD students and Postdocs. We provide the detailed statistics in Appendix A.13. We use their Google Scholar profiles to extract several proxy metrics, including the number of papers, citations, h-index, and i10-index at the time of their submission. Table 2 shows that our idea writers have an average of 12 papers and 477 citations, while every reviewer has published at least two papers and has an average citation of 635 and h-index of 7. Moreover, based on their survey responses, 72 out of the 79 reviewers have previously reviewed for conferences. These statistics indicate that our participants are highly qualified and have substantial research experience.

## 4.3 IDEA WRITING

We report statistics of our idea writers' ideas to measure their quality. As shown in Table 3, idea writers indicate a moderately high familiarity with their selected topic (3.7 on a 1 to 5 scale), and indicate the task as moderately difficult (3 on a 1 to 5 scale). They spent an average of 5.5 hours on the task and their ideas are 902 words long on average. These indicate that participants are putting substantial effort into this task. We show the distribution of their selected topics in Appendix A.3.

## 4.4 IDEA REVIEWING

**Review Assignment** We let all reviewer participants select their top two preferred topics as well as their preferred reviewing load (from 2 to 7). We then randomly assign them to ideas within their selected topics and all ideas are anonymized. In the assignment, we balance the number of ideas from each condition for each reviewer and ensure that each reviewer gets at least one human idea and one AI idea. Every idea is reviewed by 2 to 4 different reviewers. We also avoid assigning ideas written by authors from the same institution to avoid any potential contamination. Each reviewer wrote an average of 3.8 reviews from 2 or 3 conditions, across 1 to 3 topics (full statistics in Appendix A.14).

| Metric | Mean | Median | Min | Max | SD |
|---|---|---|---|---|---|
| Ours | | | | | |
| Familiarity (1-5) | 3.7 | 3.0 | 1.0 | 5.0 | 0.9 |
| Confidence (1-5) | 3.7 | 4.0 | 1.0 | 5.0 | 0.7 |
| Time (Minutes) | 31.7 | 30.0 | 5.0 | 120.0 | 16.8 |
| Length (Word) | 231.9 | 208.0 | 41.0 | 771.0 | 112.1 |
| ICLR 2024 | | | | | |
| Confidence (1-5) | 3.7 | 4.0 | 1.0 | 5.0 | 0.8 |
| Length (Word) | 421.5 | 360.0 | 14.0 | 2426.0 | 236.4 |
| Length (Word; Strengths & Weaknesses) | 247.4 | 207.0 | 2.0 | 2010.0 | 176.4 |

Table 4: Statistics of our collected reviews, with ICLR 2024 reviews as a baseline (for the 1.2K submissions that mentioned the keyword "language models").

**Review Quality Check** Apart from ensuring reviewer qualifications, we also compute statistics to measure the quality of the reviews in Table 4. On average, the reviewers indicated a familiarity of 3.7 (out of 5) in their selected topic and a confidence of 3.7 (out of 5) in their reviews. This is comparable with the 1.2K ICLR 2024 submissions related to language models, where the reviewers also have an average confidence of 3.7 out of 5. Moreover, reviewers spent an average of 32 minutes on each review, with each review being about 232 words long.

Since our review forms are different from the ICLR review forms, we compare them with the ICLR reviews where we remove the summary and question sections and only count the lengths of the strengths and weaknesses sections. This way, the ICLR reviews have an average length of 247, similar to our collected reviews. As an additional measure of review quality, out of the 298 unique reviews that we have collected, 80 of them provided links to existing papers in their rationales to justify why the proposed method is not novel. These results further validate the high quality of our review data.

## 5 MAIN RESULT: AI IDEAS ARE RATED MORE NOVEL THAN EXPERT IDEAS

In this section, we present our main finding. Consistently across three different statistical tests accounting for the possible confounders, we find that AI ideas have higher novelty scores than human ideas while being comparable on all other metrics.

**Test 1: Treating Each Review as an Independent Data Point.** In Test 1, we treat each review as an independent data point and aggregate all reviews from the same condition. We treat the `Human Ideas` as the baseline condition and compare it with `AI Ideas` and `AI Ideas + Human Rerank` using two-tailed Welch's t-tests with Bonferroni correction. We show the barplot in Figure 2 and the detailed numerical results in Table 5. Both `AI Ideas` ($\mu = 5.64 \pm \sigma = 1.76$) and `AI Ideas + Human Rerank` ($\mu = 5.81 \pm \sigma = 1.66$) are significantly better than `Human Ideas` ($\mu = 4.84 \pm \sigma = 1.79$) on the novelty score ($p < 0.01$). In this particular test, the AI ideas in both conditions are also significantly better than human ideas on the excitement score ($p < 0.05$), and the `AI Ideas + Human Rerank` condition is also significantly better than `Human Ideas` in terms of the overall score ($p < 0.05$). We do not observe significant differences between AI-generated ideas and human-written ideas on the other metrics.

**Test 2: Treating Each Idea as an Independent Data Point.** Since we collect multiple reviews for each idea, one could argue that we should not treat each review as an independent data point. To account for this potential confounder, we perform Test 2 where we average the scores of each idea and treat each idea as one data point. This way, the sample size for every condition will be $N = 49$, namely the number of ideas. We treat the `Human Ideas` as the baseline condition and compare it with `AI Ideas` and `AI Ideas + Human Rerank` using two-tailed Welch's t-tests with Bonferroni correction. Under this test (Table 14 in Appendix A.15), we still see significant results ($p < 0.05$) where both `AI Ideas` ($\mu = 5.62 \pm \sigma = 1.39$) and `AI Ideas + Human Rerank` ($\mu = 5.78 \pm \sigma = 1.07$) have higher novelty scores than `Human Ideas` ($\mu = 4.86 \pm \sigma = 1.26$).

**Test 3: Treating Each Reviewer as an Independent Data Point.** Another possible confounder is that different reviewers might have different biases, for example, some reviewers may be more lenient

| Condition | Size | Mean | Median | SD | SE | Min | Max | p-value |
|---|---|---|---|---|---|---|---|---|
| **Novelty Score** | | | | | | | | |
| Human Ideas | 119 | 4.84 | 5 | 1.79 | 0.16 | 1 | 8 | – |
| AI Ideas | 109 | 5.64 | 6 | 1.76 | 0.17 | 1 | 10 | **0.00**\*\* |
| AI Ideas + Human Rerank | 109 | 5.81 | 6 | 1.66 | 0.16 | 2 | 10 | **0.00**\*\*\* |
| **Excitement Score** | | | | | | | | |
| Human Ideas | 119 | 4.55 | 5 | 1.89 | 0.17 | 1 | 8 | – |
| AI Ideas | 109 | 5.19 | 6 | 1.73 | 0.17 | 1 | 9 | **0.04**\* |
| AI Ideas + Human Rerank | 109 | 5.46 | 6 | 1.82 | 0.17 | 1 | 9 | **0.00**\*\* |
| **Feasibility Score** | | | | | | | | |
| Human Ideas | 119 | 6.61 | 7 | 1.99 | 0.18 | 1 | 10 | – |
| AI Ideas | 109 | 6.34 | 6 | 1.88 | 0.18 | 2 | 10 | 1.00 |
| AI Ideas + Human Rerank | 109 | 6.44 | 6 | 1.63 | 0.16 | 1 | 10 | 1.00 |
| **Expected Effectiveness Score** | | | | | | | | |
| Human Ideas | 119 | 5.13 | 5 | 1.76 | 0.16 | 1 | 8 | – |
| AI Ideas | 109 | 5.47 | 6 | 1.58 | 0.15 | 1 | 10 | 0.67 |
| AI Ideas + Human Rerank | 109 | 5.55 | 6 | 1.52 | 0.15 | 1 | 9 | 0.29 |
| **Overall Score** | | | | | | | | |
| Human Ideas | 119 | 4.68 | 5 | 1.90 | 0.17 | 1 | 9 | – |
| AI Ideas | 109 | 4.85 | 5 | 1.70 | 0.16 | 1 | 9 | 1.00 |
| AI Ideas + Human Rerank | 109 | 5.34 | 6 | 1.79 | 0.17 | 1 | 9 | **0.04**\* |

Table 5: Scores across all conditions by treating each review as an independent datapoint (Test 1). Size is the number of reviews for each condition and the p-values are computed with two-tailed Welch's t-tests with Bonferroni correction. We **bold** results that are statistically significant (\*$p < 0.05$;\*\* $p < 0.01$;\*\*\* $p < 0.001$). AI ideas are judged as significantly better than human ideas in terms of novelty and excitement while being comparable on all other metrics.

than others. To account for such reviewer biases, we perform Test 3 where we treat each reviewer as one data point and compute their average score on each condition. Then for each reviewer, we get their mean score difference between the AI Ideas condition and the Human Ideas condition, as well as the difference between the AI Ideas + Human Rerank condition and the Human Ideas condition. This way, we only analyze the differences among the different conditions. That is, if the differences are significantly higher than zero under the one-sample t-test, that indicates reviewers are giving higher scores to one condition compared to the other. Using this test (Table 15 in Appendix A.15), we also see significant results ($p < 0.05$) that AI ideas in both the AI Ideas and AI Ideas + Human Rerank conditions are rated more novel than Human Ideas. Therefore, we conclude that AI ideas generated by our ideation agent are judged as more novel than human expert generated ideas, consistently across all three different statistical tests. [3]

## 6 IN-DEPTH ANALYSIS OF THE HUMAN STUDY

In this section, we move beyond the statistical comparisons and dive into other aspects of our collected data. Specifically, we focus on the quality of human ideas and the extent of reviewer agreement.

### 6.1 HUMAN EXPERTS MAY NOT BE GIVING THEIR BEST IDEAS

We first investigate whether human experts are submitting their best ideas to us. We did a post-study survey to understand how idea-writing participants came up with their ideas. Out of the 49 participants, 37 of them came up with the idea on the spot, while the other 12 already had the idea before the study. Furthermore, we asked the survey question: "*How does this idea compare to your past research ideas (ideas that you actually worked on)? Please answer with a percentile. E.g., this idea is one of my top 10% ideas.*" Our participants indicated that on average their submitted ideas are about the top 43% of all their past ideas. This implies that our collected ideas are likely the median-level ideas from these expert researchers, which is reasonable given that most of them came up with the idea within the 10-day time constraint of the task.

---

[3]We also include results of fitting linear mixed-effects models in Appendix A.16, which reinforces our conclusions. Additionally, we plot the breakdown of all metrics by topic in Appendix A.17.

## 6.2 Reviewing Ideas is Inherently Subjective

Finally, we acknowledge that reviewing is inherently subjective, and reviewing based on ideas rather than executed papers might be even more subjective. We investigate this using inter-reviewer agreement. Specifically, we randomly split reviewers of each paper into half, use one half to rank the top and bottom 25% of all ideas, and then measure agreement with the held-out set of reviewers. As shown in the first block of Table 6, reviewers have a relatively low agreement (56.1%) despite the fact that we have provided detailed explanations for each metric in our review form. As a baseline comparison, the NeurIPS 2021 reviewer consistency experiment found 66.0% accuracy using this reviewer agreement metric in the balanced setting (Beygelzimer et al., 2021; Lu et al., 2024). We also computed the reviewer agreement using the same metric on the 1.2K ICLR 2024 submissions related to language models, which has a balanced accuracy of 71.9%. While our reviewer agreement is higher than random (50%), it is generally lower than conference reviewing, most likely due to the higher subjectivity involved when evaluating ideas without seeing the actual experiment results.

Apart from the above quantitative analysis, we also provide some qualitative analysis of our collected data. We provide a summary of free-text reviews in Appendix A.18, and provide four pairs of AI and human ideas along with full reviews in Appendix A.19.

# 7 LIMITATIONS OF LLMs

Our ideation agent is motivated by two potential strengths of LLMs: their ability to scale by generating a vast number of ideas - far more than any human could - and the possibility of filtering these ideas to extract the best ones from the large pool. In theory, this approach could lead to high-quality ideas by leveraging inference scaling. However, we present empirical evidence that this naive assumption about scaling idea generation has significant limitations.

## 7.1 LLMs Lack Diversity in Idea Generation

We adopted an over-generate and rank paradigm in idea generation. This raises the question: is there an upper limit to how many new ideas LLMs can generate? To answer this question, we take a closer look at 4000 generated seed ideas for each topic.

We encode all raw ideas with `all-MiniLM-L6-v2` from Sentence-Transformers. For each idea, we compute its cosine similarity with all previously generated ideas on the same topic. We consider an idea as a duplicate if it has a similarity of above 0.8 with any of the previously generated ideas. In Figure 3, we show that as the agent keeps generating new batches of ideas, the accumulated non-duplicate ideas eventually plateau. In fact, out of the 4000 generated seed ideas, there are only 200 non-duplicate unique ideas. This sets a bottleneck on our inference-time scaling since increasing the number of generated ideas simply leads to repeating duplicate ideas.

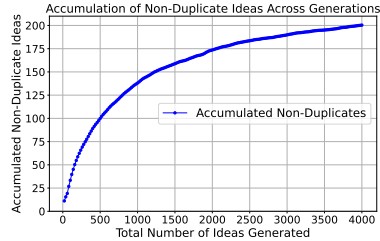

Figure 3: The accumulated non-duplicate ideas saturate as the agent keeps generating new ideas. All data points are averaged across all topics.

## 7.2 LLMs Cannot Evaluate Ideas Reliably

Most prior works have adopted *LLM-as-a-judge* for evaluating research ideas Lu et al. (2024) motivated by the observation that LLMs can have a higher agreement with human evaluators than the inter-human agreement. However, we offer some empirical evidence that LLMs cannot evaluate ideas reliably yet.

Concretely, we use the average review score of each idea to rank the top and bottom 25% of all our collected human and AI ideas, and use this to benchmark various LLM evaluators. Specifically, we obtain the LLM predicted scores of all ideas and set the median score as the threshold to measure their accuracy on our balanced idea ranking data.

In the second block of Table 6, we compare several different LLM evaluators: 1) directly giving the review criteria and prompting for a final score (Yang et al., 2024; Li et al., 2024; Baek et al., 2024); 2) our pairwise ranker as described in Section 3.3; and 3) the "AI Scientist" reviewer agent (Lu et al., 2024). All of these LLM evaluators **have a lower agreement than our expert reviewers' scores**. Even the best LLM evaluator — our own Claude-3.5 pairwise ranker — only achieves an accuracy of 53.3%, lower than our inter-reviewer consistency of 56.1%.

Even if AI-human agreement eventually matches or exceeds human-human agreement, simply meeting this baseline does not imply that AI-as-a-reviewer is meaningful, since we may be trading variance for bias, where AI reviewers are more consistent but rely on spurious correlations (Durmus et al., 2022). Our findings in Table 6 are consistent with these brittleness concerns, as we find a significant drop in AI-human agreement scores

|  | Consistency |
|---|---|
| Random | 50.0 |
| NeurIPS'21 | 66.0 |
| ICLR'24 | 71.9 |
| Ours | 56.1 |
| GPT-4o Direct | 50.0 |
| GPT-4o Pairwise | 45.0 |
| Claude-3.5 Direct | 51.7 |
| Claude-3.5 Pairwise | 53.3 |
| "AI Scientist" Reviewer | 43.3 |

Table 6: Review score consistency among human reviewers (first block) and between humans and AI (second block).

under our study compared to the original studies. Finally, even though Claude-3.5 pairwise agreements may seem close to human agreement, many other pieces of evidence throughout the paper leads us to be cautious about the use of LLM-as-a-judge in such a complex and subjective task. These include our findings on the significant discrepancy between the agent's top-ranked ideas and the human expert's top-ranked ideas (Appendix A.11) and how the `AI Ideas + Human Rerank` condition tends to score higher than the `AI Ideas` condition on all metrics in Section 5.

## 8 RELATED WORK

**Research idea generation and execution**. Several prior works explored methods to improve idea generation, such as iterative novelty boosting (Wang et al., 2024), multi-agent collaboration (Baek et al., 2024), and multi-module retrieval and revision (Yang et al., 2024). While some of them share similar components as our ideation agent, these works focus on improving the idea generation methods over vanilla prompting baselines, without comparisons to any human expert baselines. Beyond ideation, another line of work uses LLMs for executing experiments by generating code given the research problems (Huang et al., 2024; Tian et al., 2024), or combining idea generation with code generation to directly implement AI-generated ideas (Lu et al., 2024; Li et al., 2024). These works either use automatic evaluation on a pre-defined set of problems and benchmarks, setting a constrained problem space; or rely on proxy metrics like LLM evaluators, which are often unreliable.

**LLM for other research-related tasks**. LLMs have also been used for several other research-related tasks, such as generating code to perform data-driven discovery (Majumder et al., 2024; Hu et al., 2024; Guo et al., 2024; Gu et al., 2024; Ifargan et al., 2024), automatic review generation (D'Arcy et al., 2024; Liang et al., 2024), related work curation (Kang & Xiong, 2024; Ajith et al., 2024; Press et al., 2024; Lehr et al., 2024), experiment outcome prediction (Lehr et al., 2024; Zhang et al., 2024; Manning et al., 2024; Hewitt et al., 2024), and future work recommendation (Zhang et al., 2024). Unlike these works, we tackle the more creative and open-ended task of research ideation.

**Computational creativity**. Our work also connects to the line of work on examining AI's novelty and diversity in creative tasks. Previous findings include AI writings being less creative than professional writers (Chakrabarty et al., 2024); LLM generations lacking collective diversity (Zhou et al., 2024; Anderson et al., 2024); and human-AI collaboration reducing diversity (Padmakumar & He, 2024). In contrast, we focus on the human-AI comparison on the challenging task of research ideation with expert participants.

## 9 CONCLUSION

We compared research ideas generated by our AI agent with ideas written by expert researchers and observed the robust finding that expert reviewers rate AI ideas as statistically more novel than expert ideas. We recognize several limitations of the current study, including the quality of the human baseline, the subjectivity of idea evaluation, and the limited scope. We discuss future steps to address these limitations in Appendix A.1 and discuss various ethical considerations in Appendix A.2.

ACKNOWLEDGEMENT

We thank all participants who wrote and reviewed ideas for us. Many of them also provided insightful feedback on various aspects of this study. This project would not have been possible without their support. We thank Rose Wang, Dora Zhao, Irena Gao, Isabel Gallegos, Ken Liu, Aryaman Arora, Harshit Joshi, Shi Feng, Tianyu Gao, Xinran Zhao, Yangjun Ruan, Xi Ye, Mert Yuksekgonul, and members of Tatsu Lab and SALT Lab for their helpful feedback on the early version of this draft. We thank our undergraduate intern Isha Goswami and faculty administrator Eric Alejandro Pineda for assisting with review data collection and financial logistics.

This work was supported by gifts from Open Philanthropy, Tianqiao and Chrissy Chen Institute, Meta, IBM, and Amazon, and grants from ONR, NSF IIS-2247357, and CNS-2308994.

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

# A  APPENDIX

## A.1  DISCUSSION

In this section, we discuss some high-level questions readers might have and suggest ways to address them.

**Question 1: Do these collected expert ideas represent their best ideas?** One might argue that these ideas submitted by our idea-writing participants might not represent their best ideas as we discussed in subsection 6.1, since most of them came up with the idea on the spot within a short period. In order to address this concern, we have designed an experiment where we will compare AI ideas with papers accepted at top-tier AI conferences. To avoid any possible contamination, we target the upcoming EMNLP 2024 conference, which will release the accepted papers in October 2024. We have generated AI ideas with our agent on 23 topics from the EMNLP Call For Papers page in July 2024 and cached them. We pre-registered our analysis plan which also includes the link to the cached ideas. Apart from comparing the quality of these ideas, we will also compute the overlap between AI-generated ideas and accepted papers on the same topics.

**Question 2: Are evaluations based solely on ideas subjective?** In this current study, we focused solely on evaluating the ideas themselves. Ideas that sound novel and exciting might not necessarily turn into successful projects, and our results indeed indicated some feasibility trade-offs of AI ideas. We view the current study as a preliminary evaluation of AI-generated ideas. In the next phase, we will recruit researchers to execute some AI and human-generated ideas into full projects. This will enable reviewers to assess the complete experimental outcomes, providing a more reliable basis for evaluation. Furthermore, it will allow us to analyze whether our initial idea evaluations align with the assessments of the actual project outcomes.

**Question 3: Why do you focus only on prompting-based research in NLP?** The scope of our study is limited to prompting research ideas within NLP. We chose this design to facilitate the next phase of our execution experiment, where we prefer research ideas that are less resource-demanding and can be executed relatively quickly. We believe that the evaluation protocols we established should be applicable to other research domains as well, although the conclusions could be different depending on the research fields. Future work should consider extending such human study to other research domains and it would be interesting to compare how the conclusions differ.

**Question 4: Can you automate idea execution as well?** It is tempting to envision an end-to-end automated research pipeline where AI agents can implement AI-generated ideas to directly evaluate their effectiveness. Apart from speeding up scientific discovery, one could also imagine using such execution agents to automatically verify experiment results in existing papers or new submissions. We have also explored building an LLM agent to generate code to implement the generated ideas. Specifically, we provide a template codebase that consists of: (1) loading datasets from Huggingface or generating synthetic test examples; (2) implementing baseline methods; (3) implementing the proposed method; (3) loading or implementing the evaluation metrics; (4) running experiments on the testset with the baselines and the proposed method, so that the output of the agent will be a report of the baseline performance as well as the proposed method's performance. While this agent can generate code that compiles and executes, we find that the automated experiments can be **misleading** because the agent often skips or modifies steps in the baselines or proposed methods. In some cases, the metric functions are also not correctly defined. This highlights the core challenge: just comparing the final experiment results is not enough; we have to verify the faithfulness of the implementations as well. Performing such implementation verification is not a trivial task, and we leave it to future work. We provide detailed description of our idea execution agent in Appendix A.30.

A.2 ETHICAL CONSIDERATIONS

**Publication Policy.** The growing use of AI to generate research ideas raises serious concerns about the potential abuse of these technologies by students or researchers who may flood academic conferences with low-quality or poorly thought-out submissions. The availability of LLM-generated content could lead to a decline in the overall quality of academic discourse, as some individuals might take a lazy approach, relying on AI to both generate ideas and review submissions. This would undermine the credibility and integrity of the review process. The risks are real. Without proper oversight, we could see a deluge of submissions that lack depth or intellectual merit. To prevent this, it is essential to hold researchers accountable for the outputs generated through AI tools. Rigorous standards must be applied equally to both AI-assisted and human-generated research to ensure that the use of LLMs does not result in misleading, superficial, or unethical academic contributions.

**Intellectual Credit.** The use of LLMs to generate research ideas introduces significant ambiguity around the concept of intellectual credit. Traditional frameworks for attributing credit in research, based on human authorship and contribution, become less clear when AI plays a significant role in idea generation. Questions arise around how to distribute credit between the developers of the LLM, the researchers who designed the frameworks for its use, and the researchers who integrate AI-generated ideas into their work. Furthermore, it becomes increasingly difficult to trace the origins of AI-generated contributions, especially when they draw from vast datasets composed of numerous sources. This complexity calls for a broader rethinking of how intellectual credit is assigned in AI-driven research. While a complete overhaul of legal and academic norms is beyond the scope of this project, we advocate for the adoption of transparent documentation practices. Researchers should clearly disclose the role AI played in the idea generation process, specifying which models, data sources, and frameworks were used, and outlining the level of human involvement. This could ensure that the credit distribution in AI-supported research is as transparent and fair as possible.

**Potential for Misuse.** AI-generated research ideas, especially those that introduce novel concepts, have the potential to be misused in ways that could lead to harmful or destabilizing outcomes. For instance, ideation agents could be leveraged to generate adversarial attack strategies or other unethical applications. This concern aligns with broader arguments from those focused on existential risk (X-risk), who argue that AI-driven innovation could be a primary route to destabilizing the status quo, posing risks at a societal or even global level. Our stance is that such discussions on safety should be evidence-based to the extent that it is possible, and careful evaluation work is an important component of keeping these discussions grounded in actual, measured capabilities of these systems. We advocate for continued safety research specifically targeting these types of concerns—such as the development of Reinforcement Learning from Human Feedback (RLHF) systems or anti-jailbreak mechanisms for research ideation agents. Additionally, we believe it would be meaningful to create safety benchmarks that assess the ethical and safe application of AI-generated ideas.

**Idea Homogenization.** Our analysis showed that current LLMs lack diversity in idea generation. This raises important concerns that wide adoption of LLMs can result in idea homogenization, where the generated ideas only reflect a narrow set of perspectives or have systematic biases. Over time, this could lead to a reduction in the richness and diversity of research outputs globally. Future work should develop ways to either improve LLMs themselves or refine our idea generation methods to promote idea diversity. It's also important to note that our evaluation primarily assesses the quality of the typical ideas being generated, and may not fully capture the long tail of unique or novel ideas that would be truly transformative.

**Impact on Human Researchers.** The integration of AI into research idea generation introduces a complex sociotechnical challenge, as research is fundamentally a community-driven, collaborative effort. By introducing AI, particularly LLMs, into this social system, we risk unforeseen consequences. Overreliance on AI could lead to a decline in original human thought, while the increasing use of LLMs for ideation might reduce opportunities for human collaboration, which is essential for refining and expanding ideas. To mitigate these risks, future works should explore new forms of human-AI collaboration, and our results on human reranking of AI ideas show that even naive human-AI collaboration approaches can be effective. Beyond reranking, humans can play a critical role in the ideation process by providing intermediate feedback, taking AI-generated ideas as inspiration for further development, and bringing their unique expertise into the process. Understanding how to integrate LLMs into this collaborative process without disrupting the social fabric of research

will be an important ongoing problem, requiring careful consideration of the broader sociotechnical implications.

**Impact on Human Researchers.** The use of AI to generate research ideas raises concerns about the potential displacement of human researchers and the devaluation of human creativity. There is a risk that researchers may become overly reliant on AI, leading to a decline in original human thought and innovation. Furthermore, the dynamics of research collaboration could be fundamentally altered. For example, increasing use of LLMs for ideation might discourage collaboration among human researchers. To address this, we highlight the value of human-AI collaboration. We presented preliminary results where human reranking on top of AI-generated ideas can bring additional values. Apart from reranking, there are many other possible ways for humans to contribute to the collaborative ideation process, for example, by providing intermediate feedback to generated ideas, or taking AI ideas as inspirations for further improvement. Moreover, human researchers often brainstorm together and collaborative discussion helps refine ideas. How to adapt LLMs in collaborative idea generation is an interesting open question that we leave to future work.

### A.3 LIST OF RESEARCH TOPICS

We selected the following list of research topics for our research ideation task:

1. Bias: novel prompting methods to reduce social biases and stereotypes of large language models

2. Coding: novel prompting methods for large language models to improve code generation

3. Safety: novel prompting methods to improve large language models' robustness against adversarial attacks or improve their security or privacy

4. Multilingual: novel prompting methods to improve large language models' performance on multilingual tasks or low-resource languages and vernacular languages

5. Factuality: novel prompting methods that can improve factuality and reduce hallucination of large language models

6. Math: novel prompting methods for large language models to improve mathematical problem solving

7. Uncertainty: novel prompting methods that can better quantify uncertainty or calibrate the confidence of large language models

We use these topic descriptions to elicit ideas from both human participants and our LLM agent.

We show the distribution of our idea writing participants' selected topics in Table 7.

| Topic | Count |
|---|---|
| Bias | 4 |
| Coding | 9 |
| Safety | 5 |
| Multilingual | 10 |
| Factuality | 11 |
| Math | 4 |
| Uncertainty | 6 |
| Total | 49 |

Table 7: Idea topic distribution.

### A.4 PROJECT PROPOSAL TEMPLATE

We give the following project proposal template to both the AI agent and human idea writers.

**1. Title**: A concise statement of the main research question to be used as the paper title.

**2. Problem Statement**: Clearly define the problem your research intends to address. Explain clearly why this problem is interesting and important.

**3. Motivation**: Explain why existing methods are not good enough to solve the problem, and explain the inspiration behind the new proposed method. You should also motivate why the proposed method would work better than existing baselines on the problem.

**4. Proposed Method**: Explain how the proposed method works, describe all the essential steps.

**5. Step-by-Step Experiment Plan**: Break down every single step of the experiments, make sure every step is executable. Cover all essential details such as the datasets, models, and metrics to be used. If the project involves prompting, give some example prompts for each step.

**6. Test Case Examples**: Give at least two concrete examples. The first example should show how the baseline method fails on the test case. If there are multiple baselines, give examples for all of them. The second example should show how the proposed method succeeds on the test case. For each test case, include the input (test example and the full prompt) and the expected output. You should also provide an explanation for why the outputs from the proposed prompt are better. If the proposed method has multiple steps, break them down into intermediate steps.

**7. Fallback Plan**: Propose some alternative plans for what should the students do if the proposed method doesn't manage to satisfy the success criteria. For example, you can suggest additional analysis to help debug why the proposed method didn't work, which could inform alternative new methods, or just turn the project into an analysis paper instead by offering some interesting ablation and insights.

### A.5  PROJECT PROPOSAL DEMO EXAMPLE

We present a manually written demonstration example used for project proposal generation. The example is summarized from an existing paper (Dhuliawala et al., 2023). This same example is given to both the AI agent as well as the idea-writing experts.

**1. Title**:
Chain-of-Verification Reduces Hallucination in Large Language Models

**2. Problem Statement**:
Generation of plausible yet incorrect factual information, termed hallucination, is an unsolved issue in large language models.

**3. Motivation**:
A majority of the methods for reducing hallucination can be divided into roughly three categories: training-time correction, generation-time correction, and via augmentation (tool-use). We want to take a simpler approach that fully leverages the power of LLM itself. Our key motivation is that large language models, when suitably prompted, can both generate and execute a plan of how to verify themselves in order to check their own work, and finally incorporate this analysis into an improved response.

**4. Proposed Method**:
Our overall process, which we call Chain-of-Verification (CoVe), thus performs four core steps:

(1) **Generate Baseline Response**: Given a query, generate the response using the LLM.

(2) **Plan Verifications**: Given both query and baseline response, generate a list of verification questions that could help to self-analyze if there are any mistakes in the original response.

(3) **Execute Verifications**: Answer each verification question in turn, and hence check the answer against the original response to check for inconsistencies or mistakes.

(4) **Generate Final Verified Response**: Given the discovered inconsistencies (if any), generate a revised response incorporating the verification results.

Each of these steps is performed by prompting the same LLM in different ways to obtain the desired response.

**5. Step-by-Step Experiment Plan**:

1: **Gather Datasets**: We choose datasets that evaluate factual correctness, including the Multi-SpanQA dataset on closed-book QA and the FactScore dataset on generating biographies.

2: **Construct Prompts**: For the baseline, we use direct prompting where, given a query, we generate left-to-right as usual using the LLM, with no special tricks. Given that such baseline generations are typically prone to hallucination, CoVe attempts to identify these hallucinations and correct them in the following steps:

(1) **Plan Verifications**: Conditioned on the original query and the baseline response, the model is prompted to generate a series of verification questions that test the factual claims in the original baseline response.

(2) **Execute Verifications**: Given the planned verification questions, the next step is to answer them in order to assess if any hallucinations exist. The planning prompt conditions on the baseline response in the first step. The verification questions generated from planning are answered in the second step, where crucially the context given to the LLM prompt only contains the questions and not the original baseline response, hence preventing the LLM from repeating those answers directly.

(3) **Generate Final Verified Response**: Finally, the improved response that takes verification into account is generated. This is executed by a final few-shot prompt where the context takes into account all of the previous reasoning steps, the baseline response, and verification question-answer pairs, so that the corrections can take place.

3: **Select Models**: We test GPT-3.5 (Text-Davinci-003) and GPT-4 from the OpenAI API, as well as the open-source LLaMA-2-70B-chat.

**4: Get Results**: Get answer predictions from the models on these datasets with both the baselines and the proposed method.

**5: Analyze Results**: Compare whether the new method improves the performance of LLMs in these tasks as compared to the baselines.

## 6. Test Case Examples:

- **Baseline Prompt Input (Direct Prompting)**: Name some politicians who were born in NY, New York.

- **Baseline Prompt Expected Output (Direct Prompting)**: Here are some politicians who were born in NY, New York:
  1. Hillary Clinton - former secretary of state and former Democratic presidential nominee.
  2. Donald Trump - former president of the United States.
  3. Michael Bloomberg - former Mayor of New York City and former Democratic presidential candidate.

- **Proposed Prompt Input (Chain-of-Verification Prompting; Step 1: Plan Verifications)**: Generate a set of verification questions to verify each answer.

- **Proposed Prompt Expected Output (Chain-of-Verification Prompting; Step 1: Plan Verifications)**:
  - Where was Hillary Clinton born?
  - Where was Donald Trump born?
  - Where was Michael Bloomberg born?

- **Proposed Prompt Input (Chain-of-Verification Prompting; Step 2: Execute Verifications)**: Answer each of the verification questions.

- **Proposed Prompt Expected Output (Chain-of-Verification Prompting; Step 2: Execute Verifications)**:
  1. Hillary Clinton was born in Chicago, Illinois, United States on October 26, 1947.
  2. Donald Trump was born on June 14, 1946, in Queens, New York City, New York, United States.
  3. Michael Bloomberg was born on February 14, 1942, in Boston, Massachusetts, United States.

- **Proposed Prompt Input (Chain-of-Verification Prompting; Step 3: Final Response)**: (Prepend the original question and the baseline response and the verification QA pairs) Given the verification answers, generate a revised response incorporating the verification results.

- **Proposed Prompt Expected Output (Chain-of-Verification Prompting; Step 3: Final Response)**: Here are some politicians who were born in NY, New York:
  1. Donald Trump - former president of the United States.
  2. Alexandria Ocasio-Cortez - Democratic member of the U.S. House of Representatives.

- **Explanation**: Given a user query, a large language model with direct prompting generates a baseline response that may contain inaccuracies, e.g., factual hallucinations. To improve this, Chain-of-Verification first generates a plan of a set of verification questions to ask, and then executes that plan by answering them and hence checking for agreement. We find that individual verification questions are typically answered with higher accuracy than the original accuracy of the facts in the original longform generation. Finally, the revised response takes into account the verifications.

## 7. Fallback Plan:

If the proposed method does not help as compared to the baseline, analyze each step of the CoVe process to see if the verification questions are relevant, if the answers to the verification questions are correct, and whether the generated final verified response is indeed improved over the baseline response by considering the verification QA pairs. This can help us debug the proposed method or turn this into interesting analysis on the model's ability to verify and correct its own responses.

## A.6 STYLE STANDARDIZATION PROMPT

---

**Style Standardization Prompt**

You are a writing assistant specialized in editing academic writing. I will give you a student's research idea and an idea template. Your task is to edit the student's idea to follow the template's format.

**Student idea:** (Insert the student's idea here)

**Template:** (Insert the template idea here)

Make sure that you only edit the wording and formatting, including things like punctuation, capitalization, linebreaks, and bullet points. Also make sure to edit any informal wording and phrasing to use vocabulary that sounds like the template's writing style. No other changes are allowed beyond these.

The main subsections should be indexed clearly without indentation at the beginning. The title subsection does not need indexing; other subsections, including problem statement, motivation, proposed method, step-by-step experiment plan, test case examples, and fallback plan, should be indexed 1 to 6. Each subsection can then have sub-bullets for sub-subsections if applicable. Leave an empty line after each subsection.

You should use tab as indentation and make sure to use appropriate nested indentation for sub-bullets. All bullets should have a clear hierarchy so people can easily differentiate the sub-bullets. Only leave empty lines between subsections and remove any extra line breaks. If many bullet points are clustered together in a paragraph, separate them clearly with indentation and appropriate bullet point markers. Change to a new line for each new bullet point.

For the fallback plan, do not list a bunch of bullet points. Instead, condense them into one coherent paragraph.

For line breaks, avoid Raw String Literals or Double Backslashes when using "\n", and change them to spaces or tabs.

For in-line citations, if the citation mentioned the author's last name (like "(Si et al., 2023)" or "(An et al., 2024)"), you should keep them there; but if the citation is just a number (like "[1]" or "[3,4,5]"), you should just remove it and do some necessary rephrasing to make the sentence still sound coherent without the references.

Apart from minor rephrasing and changing formatting, do not change any content of the idea. You must preserve the exact meaning of the original idea, do not change, remove, or add any other details. Do not drop any subsections (including test case examples). Do not rename any models, datasets, or methods. Do not drop clarification or examples in brackets and do not drop any data source mentions (e.g., Chatbot Arena or Wildchat)! Note that when indexing test case examples, each test case example could have multiple steps of inputs and outputs and you shouldn't give separate indices to them. Each test case example should be a whole set of input-output pairs for the baseline(s) and proposed method.

For the proposed method subsection, avoid any big changes. If the subsection comes in as a coherent paragraph, you don't have to break it down into bullet points. If the subsection is already in bullet points, you should keep it that way. If the subsection is a mix of both, you should keep the bullet points and the coherent paragraph as they are.

Keep all the clarification and examples mentioned in all the subsections and do not remove any of them (including those in brackets).

For model selection, if any version of Claude is mentioned, change it to the latest version of Claude (Claude-3.5); if any version of LLaMA is mentioned, change it to the latest version LLaMA-3. Do not make any other model changes.

Now directly generate the edited student idea to match the format of the template.

---

## A.7 IDEA REVIEW FORM

We use the following review form to elicit reviews from all expert reviewers. Reviewers have one week of time to finish each review.

**1. Name**

**2. Institution**

**3. Email**

**4. Consent**

**5. Honor Code**: I confirm that I will not use ChatGPT, Claude, Gemini, or any other AI tools when writing my reviews.

**6. Familiarity**: Before reviewing the idea, please indicate how familiar you are with the given topic on a scale of 1 - 5 (this is just for us to understand potential confounders).

1. You have never read about this topic before
2. You have read at least one paper on this topic
3. You have read multiple papers on this topic but have not published any paper on it
4. You have co-authored at least one paper on this topic
5. You have co-authored multiple papers on this topic or have published at least one first-author paper on this topic

**7. Experience**: Have you reviewed for major NLP or AI conferences before (e.g., *ACL, COLING, NeurIPS, ICLR, ICML, AAAI)?

**8. Full Research Idea Proposal**

**9. Novelty Score**: Whether the idea is creative and different from existing works on the topic, and brings fresh insights. You are encouraged to search for related works online. You should consider all papers that appeared online prior to July 2024 as existing work when judging the novelty.

1. Not novel at all - there are many existing ideas that are the same
2. 
3. Mostly not novel - you can find very similar ideas
4. 
5. Somewhat novel - there are differences from existing ideas but not enough to turn into a new paper
6. Reasonably novel - there are some notable differences from existing ideas and probably enough to turn into a new paper
7. 
8. Clearly novel - major differences from all existing ideas
9. 
10. Very novel - very different from all existing ideas in a very interesting and clever way

**10. Novelty Rationale**: Short justification for your score. If you give a low score, you should specify similar related works. (Your rationale should be at least 2-3 sentences.)

**11. Feasibility Score**: How feasible it is to implement and execute this idea as a research project? Specifically, how feasible the idea is for a typical CS PhD student to execute within 1-2 months of time. You can assume that we have abundant OpenAI / Anthropic API access, but limited GPU compute.

1. Impossible: the idea doesn't make sense or the proposed experiments are flawed and cannot be implemented

2.

3. Very challenging: there are flaws in the proposed method or experiments, or the experiments require compute/human resources beyond any academic lab

4.

5. Moderately feasible: It can probably be executed within the given time frame but would require careful planning, efficient use of APIs or some advanced computational strategies to overcome the limited GPU resources, and would require some modifications to the original proposal to make it work

6. Feasible: Can be executed within the given constraints with some reasonable planning

7.

8. Highly Feasible: Straightforward to implement the idea and run all the experiments

9.

10. Easy: The whole proposed project can be quickly executed within a few days without requiring advanced technical skills

**12. Feasibility Rationale**: Short justification for your score. If you give a low score, you should specify what parts are difficult to execute and why. (Your rationale should be at least 2-3 sentences.)

**13. Expected Effectiveness Score**: How likely the proposed idea is going to work well (e.g., better than existing baselines).

1. Extremely Unlikely: The idea has major flaws and definitely won't work well

2.

3. Low Effectiveness: The idea might work in some special scenarios but you don't expect it to work in general

4.

5. Somewhat ineffective: There might be some chance that the proposed idea can work better than existing baselines but the improvement will be marginal or inconsistent

6. Somewhat effective: There is a decent chance that the proposed idea can beat existing baselines by moderate margins on a few benchmarks

7.

8. Probably Effective: The idea should offer some significant improvement over current methods on the relevant benchmarks

9.

10. Definitely Effective: You are very confident that the proposed idea will outperform existing methods by significant margins on many benchmarks

**14. Expected Effectiveness Rationale**: Short justification for your score. (Your rationale should be at least 2-3 sentences.)

**15. Excitement Score**: How exciting and impactful this idea would be if executed as a full project. Would the idea change the field and be very influential.

1. Poor: You cannot identify the contributions of this idea, or it's not interesting at all and you would fight to have it rejected at any major AI conference

2.

3. Mediocre: this idea makes marginal contributions and is very incremental

4.

5. Leaning negative: it has interesting bits but overall not exciting enough

6. Learning positive: exciting enough to be accepted at a major AI conference, but still has some weaknesses or somewhat incremental

7.

8. Exciting: would deepen the community's understanding or make major progress in this research direction

9.

10. Transformative: would change the research field profoundly and worth a best paper award at major AI conferences

**16. Excitement Rationale**: Short justification for your score. (Your rationale should be at least 2-3 sentences.)

**17. Overall Score**: Overall score: Apart from the above, you should also give an overall score for the idea on a scale of 1 - 10 as defined below (Major AI conferences in the descriptions below refer to top-tier NLP/AI conferences such as *ACL, COLM, NeurIPS, ICLR, and ICML.):

1. Critically flawed, trivial, or wrong, would be a waste of students' time to work on it

2. Strong rejection for major AI conferences

3. Clear rejection for major AI conferences

4. Ok but not good enough, rejection for major AI conferences

5. Decent idea but has some weaknesses or not exciting enough, marginally below the acceptance threshold of major AI conferences

6. Marginally above the acceptance threshold of major AI conferences

7. Good idea, would be accepted by major AI conferences

8. Top 50% of all published ideas on this topic at major AI conferences, clear accept

9. Top 15% of all published ideas on this topic at major AI conferences, strong accept

10. Top 5% of all published ideas on this topic at major AI conferences, will be a seminal paper

**18. Overall Rationale**: You should also provide a rationale for your overall score. (Your rationale should be at least 2-3 sentences.)

**19. Confidence**: Additionally, we ask for your confidence in your review on a scale of 1 to 5 defined as following:

1. Your evaluation is an educated guess
2. You are willing to defend the evaluation, but it is quite likely that you did not understand central parts of the paper
3. You are fairly confident that the evaluation is correct
4. You are confident but not absolutely certain that the evaluation is correct
5. You are absolutely certain that the evaluation is correct and very familiar with the relevant literature

**20. Time**: How many minutes did you spend on this task?

## A.8 IDEA GENERATION AGENT: ADDITIONAL IMPLEMENTATION DETAILS

**Seed Idea Generation**    Due to the max output length limit of the LLM API, we first generate a large number of shorter seed ideas. We keep the seed ideas short so that we can explore more different ideas given the same output token budget. We provide a demonstration example of the seed idea in Appendix A.9. Then, we perform duplication and expand each remaining seed idea into a full project proposal following our standard template in Appendix A.4.

**Retrieval Augmentation**    We apply retrieval augmentation to the idea generation prompt in order to increase diversity in the idea generation. To maximize diversity, we apply retrieval augmentation half of the time when generating seed ideas, and we randomly select $k = 10$ papers from the top 20 retrieved papers when applying retrieval augmentation.

**Idea Filtering**    After expanding seed ideas into full project proposals, we did some basic filtering to remove any project proposals that failed the novelty and feasibility checks:

1. Novelty: We use the literature review module to retrieve the top 10 most relevant papers to the generated idea and ask the LLM to compare each of them to the generated idea. The idea will be filtered as long as any one of the retrieved papers is judged as equivalent.

2. Feasibility: The idea will be filtered if it requires extensive manual labor or hardware resources beyond the capacity of a typical academic lab. The idea will also be filtered if it involves any inconsistency in the experimental setups or assumptions. For example, if the idea assumes only black-box API access of the LLMs, then it shouldn't involve experiments that need internal weight access.

This filtered out about 1% of the generated project proposals.

### A.9 DEMONSTRATION EXAMPLE: SEED IDEA GENERATION

We present a demonstration example used for seed idea generation. The example is summarized from an existing paper (Dhuliawala et al., 2023).

**Title**:
Chain-of-Verification Prompting

**Problem**:
Generation of plausible yet incorrect factual information, termed hallucination, is an unsolved issue in large language models.

**Existing Methods**:
A majority of the methods for reducing hallucination can be divided into roughly three categories: training-time correction; generation-time correction; and via augmentation (tool-use).

**Motivation**:
A key observation is that large language models, when suitably prompted, can both generate and execute a plan of how to verify themselves in order to check their own work, and finally incorporate this analysis into an improved response.

**Proposed Method**:
Our overall process, which we call Chain-of-Verification (CoVe), thus performs four core steps:

(1) **Generate Baseline Response**: Given a query, generate the response using the LLM.

(2) **Plan Verifications**: Given both query and baseline response, generate a list of verification questions that could help to self-analyze if there are any mistakes in the original response.

(3) **Execute Verifications**: Answer each verification question in turn, and hence check the answer against the original response to check for inconsistencies or mistakes.

(4) **Generate Final Verified Response**: Given the discovered inconsistencies (if any), generate a revised response incorporating the verification results.

Each of these steps is performed by prompting the same LLM in different ways to obtain the desired response.

**Experiment Plan**:
Compare with zero-shot prompting, Chain-of-Thought, and few-shot prompting on the MultiSpanQA dataset on closed-book QA and FactScore dataset on generating biographies.

A.10 GENERATED SEED IDEAS AND THEIR NEAREST NEIGHBORS

We present several randomly sampled generated seed ideas (see Appendix A.8 for the definition of seed ideas) on the topic of "novel prompting methods that can better quantify uncertainty or calibrate the confidence of large language models". For each idea, we show the most similar idea (nearest neighbor) based on the embedding similarity, along with the similarity score. In practice, we set a threshold threshold of 0.8 for determining whether two ideas are duplicates.

**Idea 1:**
**Title:** Adaptive Precision Boundary Probing
**Problem:** LLMs often provide uncertainty estimates that are either too coarse-grained or inappropriately precise, failing to adapt to the inherent ambiguity or precision requirements of different queries.
**Existing Methods:** Existing uncertainty quantification methods typically use fixed precision scales or calibration techniques that don't adapt to the specific context and precision requirements of each query.
**Motivation:** Human experts adjust the precision of their uncertainty estimates based on the nature of the question and the available evidence. We can incorporate this adaptive approach to improve LLM uncertainty quantification.
**Proposed Method:** We introduce Adaptive Precision Boundary Probing (APBP), a dynamic prompting technique that iteratively refines the precision of uncertainty estimates. Given a query, APBP starts with a coarse-grained confidence interval. It then prompts the model to assess whether this interval is appropriately precise given the query's context and the model's knowledge. If the model determines that greater precision is warranted, APBP iteratively narrows the interval, prompting the model at each step to justify the increased precision. Conversely, if the model recognizes high ambiguity or limited knowledge, APBP widens the interval. Throughout this process, the model is asked to explicitly reason about the factors influencing the appropriate level of precision, such as the specificity of the query, the reliability of relevant knowledge, and potential sources of ambiguity. The final output is an uncertainty estimate with a precision level tailored to the specific query and the model's knowledge state.
**Experiment Plan:** We will evaluate APBP on a diverse set of tasks with varying inherent precision requirements, including numerical estimation, date prediction, and open-ended text generation. We'll compare APBP against fixed-precision uncertainty estimation methods, measuring both calibration accuracy and the appropriateness of precision levels as judged by human experts.

**Nearest Neighbor of Idea 1:**
**Title:** Contextual Confidence Oscillation
**Problem:** Current methods for quantifying uncertainty in large language models often fail to capture the dynamic nature of confidence across different contexts within a single query.
**Existing Methods:** Most existing approaches use static confidence scores or calibration techniques that don't account for intra-query contextual shifts.
**Motivation:** Human confidence often fluctuates as we process different parts of a complex question or task. By mimicking this oscillation, we can potentially capture a more nuanced and accurate representation of model uncertainty.
**Proposed Method:** We propose Contextual Confidence Oscillation (CCO), a novel prompting technique that encourages the model to continuously re-evaluate and express its confidence as it processes a query. The prompt is structured as a series of checkpoints, where the model must pause its reasoning, reflect on its current confidence level, and explain any changes since the last checkpoint. This creates a confidence trajectory that can be analyzed for patterns, sudden drops, or gradual increases. Additionally, we introduce 'confidence disruptors' - intentionally ambiguous or challenging sub-queries inserted at various points to test the model's ability to recognize and express increased uncertainty when appropriate.
**Experiment Plan:** We will evaluate CCO against standard uncertainty quantification methods on a range of tasks, including multi-step reasoning problems, ambiguous queries, and long-form text analysis. We'll measure not just overall accuracy of uncertainty estimates, but also the correlation between confidence oscillations and human-annotated difficulty levels of different parts of each query. We'll also analyze how well the model's expressed confidence trajectory aligns with its actual performance across different segments of complex tasks.

**Similarity: 0.70**

**Idea 2:**
**Title:** Quantum Superposition Confidence Prompting
**Problem:** Current LLMs struggle to accurately quantify uncertainty across multiple possible answers, often defaulting to overconfidence in a single response.
**Existing Methods:** Existing approaches typically involve single-path reasoning or limited branching, failing to capture the full spectrum of uncertainty.
**Motivation:** Inspired by quantum mechanics, where particles can exist in multiple states simultaneously, we propose a method that allows LLMs to consider multiple answer possibilities concurrently.
**Proposed Method:** We introduce Quantum Superposition Confidence Prompting (QSCP), where the LLM is instructed to generate multiple potential answers simultaneously, assigning confidence scores to each. The prompt encourages the model to 'exist in multiple states,' exploring contradictory answers and their implications concurrently. For example: 'Imagine you are in a quantum superposition of multiple expert personas. Each persona will provide an answer to the following question, along with a confidence score (0-100%). Ensure the personas explore contradictory viewpoints. Question: [INSERT QUESTION]'. The LLM then generates responses from multiple personas, each with its own confidence score. The final uncertainty is derived from the distribution of these scores, providing a more nuanced understanding of the model's confidence across possible answers.
**Experiment Plan:** Compare QSCP against standard prompting, chain-of-thought, and other uncertainty quantification methods on diverse question-answering datasets. Evaluate using metrics such as calibration error, Brier score, and a novel 'quantum uncertainty score' that measures the spread and coherence of the generated answer superposition.

**Nearest Neighbor of Idea 2:**
**Title:** Quantum Superposition Prompting
**Problem:** Traditional methods for uncertainty quantification in large language models often fail to capture the full range of possible interpretations and outcomes, especially for queries with inherent ambiguity or multiple valid perspectives.
**Existing Methods:** Current approaches typically focus on generating a single response with an associated confidence score, or at best, a small set of discrete alternatives.
**Motivation:** Drawing inspiration from the principle of superposition in quantum mechanics, we propose a method to represent and reason about multiple possible outcomes simultaneously, providing a richer and more nuanced uncertainty quantification.
**Proposed Method:** We present Quantum Superposition Prompting (QSP), a novel framework for exploring and quantifying uncertainty in language model outputs. QSP begins by prompting the model to generate a 'superposition' of possible interpretations or approaches to the given query. Each element in this superposition is assigned a complex amplitude, representing both its probability and its relationship to other elements. The model is then guided through a series of 'measurement' prompts, designed to collapse this superposition along different bases of interpretation. These measurements yield probability distributions over outcomes, capturing different facets of uncertainty. QSP employs techniques inspired by quantum computing, such as interference and entanglement, to model how different interpretations interact and influence each other. The final uncertainty quantification is derived from the full set of measurements, providing a multi-dimensional representation of the model's uncertainty that captures ambiguity, conflicting evidence, and the interdependence of different interpretations.
**Experiment Plan:** We will evaluate QSP on tasks that inherently involve multiple valid perspectives or ambiguous interpretations, such as ethical dilemmas, creative writing prompts, and open-ended analytical questions. Metrics will include the diversity and coherence of generated superpositions, the ability to capture human-judged ambiguities, and improvements in uncertainty calibration compared to classical methods.

**Similarity: 0.77**

**Idea 3:**
**Title:** Fractal Uncertainty Decomposition

**Problem:** LLMs often provide overly simplistic uncertainty estimates that fail to capture the hierarchical and nested nature of uncertainty in complex knowledge domains.

**Existing Methods:** Current uncertainty quantification methods typically produce flat, single-dimensional confidence scores that don't reflect the multi-layered structure of knowledge and uncertainty.

**Motivation:** By recursively decomposing a query into sub-components and assessing uncertainty at multiple levels of granularity, we can construct a more comprehensive and structurally informed uncertainty estimate.

**Proposed Method:** We introduce Fractal Uncertainty Decomposition (FUD), a prompting technique that recursively breaks down a query into a hierarchical structure of sub-queries, assessing uncertainty at each level. Given an initial query, FUD prompts the model to identify key sub-components or aspects of the question. For each sub-component, the model provides an answer and a confidence estimate. If the confidence for a sub-component is below a certain threshold, FUD recursively applies the same decomposition process to that sub-component. This continues until either a maximum depth is reached or all sub-components have high confidence. The resulting structure is a tree of nested confidence estimates. FUD then aggregates these estimates bottom-up, using a combination of statistical methods and prompted meta-analysis by the model. The final output is both an overall uncertainty estimate and a detailed map of the uncertainty structure, showing how confidence varies across different aspects and levels of the query.

**Experiment Plan:** We will evaluate FUD on complex, multi-faceted tasks such as scientific explanation, historical analysis, and technical troubleshooting. We will compare its performance to flat confidence estimation methods and other hierarchical approaches. Evaluation metrics will include traditional calibration measures, as well as new metrics designed to assess the quality and informativeness of the uncertainty decomposition. We will also conduct case studies to demonstrate how FUD can provide more actionable and interpretable uncertainty information in real-world scenarios.

**Nearest Neighbor of Idea 3:**

**Title:** Semantic Fractal Decomposition

**Problem:** Current uncertainty quantification methods for large language models often fail to capture the hierarchical and self-similar nature of conceptual understanding, leading to inconsistent confidence estimates across different levels of abstraction.

**Existing Methods:** Existing approaches typically focus on flat, single-level uncertainty estimates or simple hierarchical decompositions that don't fully capture the complex, nested nature of semantic understanding.

**Motivation:** Drawing inspiration from fractal geometry, where patterns repeat at different scales, we propose a method that recursively decomposes concepts and queries into self-similar sub-components, allowing for a more nuanced and scale-invariant approach to uncertainty quantification.

**Proposed Method:** We present Semantic Fractal Decomposition (SFD), a prompting technique that guides the model to recursively break down a given query or concept into smaller, self-similar components. At each level of decomposition, the model is asked to provide a confidence estimate. The process continues until a predefined depth is reached or the model indicates it can no longer meaningfully decompose the concept. The final uncertainty estimate is then constructed by aggregating these multi-level confidence scores using a novel fractal dimension-inspired algorithm. This approach allows for capturing uncertainty that may be present at different semantic scales and provides a more robust and consistent measure of the model's confidence across varying levels of abstraction.

**Experiment Plan:** We will evaluate SFD on a diverse set of tasks ranging from simple factual queries to complex, multi-faceted questions in domains like philosophy, science, and law. We will compare its performance against traditional flat confidence estimation techniques and simpler hierarchical methods. Key metrics will include the consistency of uncertainty estimates across related queries at different levels of abstraction, the correlation between fractal-aggregated confidence scores and actual model performance, and the interpretability of the decomposition process.

**Similarity: 0.81**

### A.11 OVERLAP BETWEEN AI RANKING AND EXPERT RERANKING

We show the overlap between the `AI Ideas` condition and the `AI Ideas + Human Rerank` conditions in Table 8. We note that 17 out of the 49 ideas in the `AI Ideas + Human Rerank` condition are also ranked as top ideas in the `AI Ideas` condition by the AI ranker, while the other 32 are not.

| Topic | Overlap | New |
|---|---|---|
| Bias | 2 | 2 |
| Coding | 4 | 5 |
| Safety | 2 | 3 |
| Multilingual | 5 | 5 |
| Factuality | 2 | 9 |
| Math | 2 | 2 |
| Uncertainty | 1 | 5 |
| Total | 18 | 31 |

Table 8: Overlap of ideas between `AI + Human Rerank` and `AI` conditions, broken down by topic.

### A.12 QUALITY CONTROL OF HUMAN EXPERT IDEAS

Each expert is instructed to choose one of the seven specified topics and write one idea on it within 10 days, following the given template in the annotation document. We included an honor code statement to ask the participants to not use any AI tools in their idea writing. We collected $N = 50$ ideas originally and manually checked all of them for quality control. We filtered out one of them as being essentially a paraphrase of an existing paper's abstract. We compensated the participant nevertheless but excluded them from the review task.

### A.13 PARTICIPANT DETAILS

We show the detailed position breakdown of our 49 idea-writing participants in Table 9 and the positions of our 79 reviewer participants in Table 10.

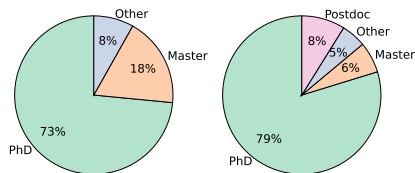

Figure 4: Positions of our idea writer (left) and reviewer (right) participants.

| Position | Count |
|---|---|
| Postdoc | 1 |
| PhD | 36 |
| Master | 9 |
| Undergraduate | 1 |
| Research Scientist | 1 |
| Machine Learning Engineer | 1 |

Table 9: Positions of the 49 idea writing participants.

| Position | Count |
|---|---|
| Postdoc | 7 |
| PhD | 63 |
| Master | 5 |
| Research Scientist | 3 |
| Machine Learning Engineer | 1 |

Table 10: Positions of the 79 idea reviewing participants.

We show the institutions of the idea writing participants in Table 11.

| Institution | Count |
|---|---|
| Stanford University | 11 |
| University of Southern California | 6 |
| University of Maryland | 3 |
| University of Illinois Urbana-Champaign | 3 |
| Johns Hopkins University | 3 |
| Columbia University | 2 |
| Carnegie Mellon University | 2 |
| University of Pennsylvania | 1 |
| Princeton University | 1 |
| Penn State University | 1 |
| Portland State University | 1 |
| Stony Brook University | 1 |
| University of Chicago | 1 |
| University of Washington | 1 |
| UC Berkeley | 1 |
| UCSD | 1 |
| Massachusetts Institute of Technology | 1 |
| George Washington University | 1 |
| Yale University | 1 |
| University of Toronto | 1 |
| Georgia Institute of Technology | 1 |
| National University of Singapore | 1 |
| Peking University | 1 |
| Tsinghua University | 1 |
| LinkedIn | 1 |
| Norm AI | 1 |

Table 11: Institutions of the 49 idea writing participants.

We show the institutions of the idea reviewing participants in Table 12.

| Institution | Count |
|---|---|
| Stanford University | 25 |
| UC Berkeley | 4 |
| UT Austin | 4 |
| University of Maryland | 4 |
| Princeton University | 3 |
| University of Washington | 3 |
| University of Southern California | 3 |
| Carnegie Mellon University | 3 |
| University of Chicago | 2 |
| Johns Hopkins University | 2 |
| UCLA | 2 |
| Georgia Institute of Technology | 2 |
| University of Illinois Urbana-Champaign | 2 |
| Tsinghua University | 2 |
| Stony Brook University | 1 |
| Ohio State University | 1 |
| National University of Singapore | 1 |
| University of Michigan | 1 |
| Dartmouth College | 1 |
| Massachusetts Institute of Technology | 1 |
| University of Pennsylvania | 1 |
| University of Toronto | 1 |
| Portland State University | 1 |
| Penn State University | 1 |
| New York University | 1 |
| Columbia University | 1 |
| UC Santa Barbara | 1 |
| Brown University | 1 |
| Amazon | 1 |
| LinkedIn | 1 |
| Norm AI | 1 |
| AMD | 1 |

Table 12: Institutions of the 79 reviewer participants.

### A.14 REVIEW ASSIGNMENT STATISTICS

We list the details of the review assignment in Table 13.

| Metric | Mean | Min | Max | SD |
|---|---|---|---|---|
| # Reviews | 3.8 | 2.0 | 7.0 | 1.3 |
| # Conditions | 2.5 | 2.0 | 3.0 | 0.5 |
| # Topics | 1.5 | 1.0 | 3.0 | 0.6 |

Table 13: Statistics of the review assignment.

## A.15 ADDITIONAL STATISTICAL TESTS

We present two additional statistical tests that account for potential confounders by treating each idea as one data point and each reviewer as one data point, respectively.

| Condition | Size | Mean | Median | SD | SE | Min | Max | p-value |
|---|---|---|---|---|---|---|---|---|
| **Novelty Score** | | | | | | | | |
| `Human Ideas` | 49 | 4.86 | 5.00 | 1.26 | 0.18 | 1.50 | 7.00 | – |
| `AI Ideas` | 49 | 5.62 | 5.50 | 1.39 | 0.20 | 1.50 | 8.33 | **0.03*** |
| `AI Ideas + Human Rerank` | 49 | 5.78 | 6.00 | 1.07 | 0.15 | 3.00 | 8.33 | **0.00**** |
| **Excitement Score** | | | | | | | | |
| `Human Ideas` | 49 | 4.56 | 4.33 | 1.16 | 0.17 | 2.00 | 7.00 | – |
| `AI Ideas` | 49 | 5.18 | 5.50 | 1.33 | 0.19 | 2.50 | 7.33 | 0.08 |
| `AI Ideas + Human Rerank` | 49 | 5.45 | 5.50 | 1.36 | 0.19 | 1.00 | 7.33 | **0.00**** |
| **Feasibility Score** | | | | | | | | |
| `Human Ideas` | 49 | 6.53 | 7.00 | 1.50 | 0.21 | 3.00 | 9.00 | – |
| `AI Ideas` | 49 | 6.30 | 6.00 | 1.27 | 0.18 | 2.50 | 8.50 | 1.00 |
| `AI Ideas + Human Rerank` | 49 | 6.41 | 6.50 | 1.06 | 0.15 | 4.00 | 9.00 | 1.00 |
| **Expected Effectiveness Score** | | | | | | | | |
| `Human Ideas` | 49 | 5.10 | 5.33 | 1.14 | 0.16 | 3.00 | 7.00 | – |
| `AI Ideas` | 49 | 5.48 | 5.50 | 1.23 | 0.18 | 2.00 | 7.50 | 0.58 |
| `AI Ideas + Human Rerank` | 49 | 5.57 | 5.50 | 0.99 | 0.14 | 3.00 | 7.50 | 0.17 |
| **Overall Score** | | | | | | | | |
| `Human Ideas` | 49 | 4.69 | 4.67 | 1.16 | 0.17 | 2.00 | 6.67 | – |
| `AI Ideas` | 49 | 4.83 | 5.00 | 1.34 | 0.19 | 1.50 | 7.50 | 1.00 |
| `AI Ideas + Human Rerank` | 49 | 5.32 | 5.50 | 1.24 | 0.18 | 2.00 | 7.50 | 0.06 |

Table 14: Scores across all conditions by averaging the scores for each idea and treating each idea as one data point (Test 2). Size is the number of ideas for each condition, and the p-values are computed with two-tailed Welch's t-tests with Bonferroni correction. We **bold** results that are statistically significant ($^*p < 0.05;^{**} p < 0.01$). AI ideas are judged as significantly better than human ideas in terms of novelty while being comparable on all other metrics.

| | N | Mean Diff | p-value |
|---|---|---|---|
| **Novelty Score** | | | |
| `AI Ideas` vs `Human Ideas` | 70 | 0.94 | **0.00**** |
| `AI Ideas + Human Rerank` vs `Human Ideas` | 65 | 0.86 | **0.00**** |
| **Excitement Score** | | | |
| `AI Ideas` vs `Human Ideas` | 70 | 0.73 | **0.01*** |
| `AI Ideas + Human Rerank` vs `Human Ideas` | 65 | 0.87 | **0.00**** |
| **Feasibility Score** | | | |
| `AI Ideas` vs `Human Ideas` | 70 | -0.29 | 0.36 |
| `AI Ideas + Human Rerank` vs `Human Ideas` | 65 | -0.08 | 0.74 |
| **Effectiveness Score** | | | |
| `AI Ideas` vs `Human Ideas` | 70 | 0.42 | 0.16 |
| `AI Ideas + Human Rerank` vs `Human Ideas` | 65 | 0.39 | 0.16 |
| **Overall Score** | | | |
| `AI Ideas` vs `Human Ideas` | 70 | 0.24 | 0.36 |
| `AI Ideas + Human Rerank` vs `Human Ideas` | 65 | 0.66 | **0.01*** |

Table 15: Mean score differences between AI ideas and human ideas by treating each reviewer as a data point (Test 3). All p-values are computed with one-sample t-tests with Bonferroni correction. We **bold** results that are statistically significant ($^*p < 0.05;^{**} p < 0.01$).

### A.16 MIXED-EFFECTS MODELS

One way to combine all the statistical tests above is to fit a linear mixed-effects model where we treat the condition as the fixed effect and other factors including reviewer and idea as random effects, while also accounting for the differences among different topics. This way, we can rely on the regression to account for all the possible confounders as the random effects. Specifically, for each metric, we fit the following linear mixed-effects model:

```
model = smf.mixedlm("Score ~ Condition", df,
                    groups=df["Topic"],
                    re_formula="~Condition",
                    vc_formula={"ReviewerID": "0 + C(ReviewerID)",
                            "IdeaID": "0 + C(IdeaID)"})
```

This mixed-effects model analyzes the relationship between *Score* and *Condition*, while accounting for the hierarchical structure of the data. Fixed effects estimate the average effect of *Condition* on *Score*. Random intercepts for *Topic* allow for varying baseline scores across topics, and random slopes for *Condition* within each topic allow the effect of *Condition* to vary by topic. Additionally, variance components for *ReviewerID* and *IdeaID* account for variability in scores specific to individual reviewers and ideas, respectively.

The results are shown in Table 16. The intercepts in the mixed-effects models represent the estimated mean score of the baseline condition, which in this context is the `Human Ideas`. The coefficients for Condition[`AI Ideas`] and Condition[`AI Ideas + Human Rerank`] in the mixed-effects models represent the difference in the mean score for each metric between the AI ideas and the baseline (human ideas). For example, a positive coefficient of 0.761 for the novelty score means that `AI Ideas`, on average, score 0.761 points higher than `Human Ideas` on the novelty score metric; conversely, a negative coefficient of -0.330 for the feasibility score means that `AI Ideas`, score 0.330 points lower than `Human Ideas` on feasibility on average. The topic (group) variance in the mixed-effects model represents the variability in the outcome metric that can be attributed to differences between the topics, which is relatively small in general. Similarly, the idea variance and reviewer variance in the mixed-effects model represent the variability in the outcome metric that can be attributed to differences between individual ideas and between reviewers, respectively. The reviewer variances are high in general, suggesting that there is substantial variability in how different reviewers rate the same ideas. This implies that reviewer differences play a significant role in the observed scores, with some reviewers consistently giving higher or lower ratings.

Overall, the results from the mixed-effects models confirm our main conclusion that AI ideas are rated as significantly more novel than human ideas.

|  | Coef. | SE | $p$ |
|---|---|---|---|
| **Novelty Score** | | | |
| Intercept | 4.826 | 0.217 | **0.000*** |
| Condition[AI Ideas] | 0.756 | 0.331 | **0.023*** |
| Condition[AI Ideas + Human Rerank] | 0.902 | 0.305 | **0.003**** |
| Idea Var | 0.412 | 0.178 | |
| Reviewer Var | 0.803 | 0.202 | |
| **Excitement Score** | | | |
| Intercept | 4.493 | 0.212 | **0.000*** |
| Condition[AI Ideas] | 0.626 | 0.303 | **0.039*** |
| Condition[AI Ideas + Human Rerank] | 0.879 | 0.298 | **0.003**** |
| Idea Var | 0.495 | 0.227 | |
| Reviewer Var | 0.782 | 0.167 | |
| **Feasibility Score** | | | |
| Intercept | 6.595 | 0.224 | **0.000*** |
| Condition[AI Ideas] | -0.300 | 0.294 | 0.307 |
| Condition[AI Ideas + Human Rerank] | -0.183 | 0.314 | 0.561 |
| Idea Var | 0.476 | 0.188 | |
| Reviewer Var | 1.035 | 0.261 | |
| **Expected Effectiveness Score** | | | |
| Intercept | 5.156 | 0.211 | **0.000*** |
| Condition[AI Ideas] | 0.310 | 0.140 | **0.027*** |
| Condition[AI Ideas + Human Rerank] | 0.383 | 0.242 | 0.114 |
| Idea Var | 0.200 | 0.151 | |
| Reviewer Var | 0.469 | 0.141 | |
| **Overall Score** | | | |
| Intercept | 4.660 | 0.242 | **0.000*** |
| Condition[AI Ideas] | 0.137 | 0.294 | 0.640 |
| Condition[AI Ideas + Human Rerank] | 0.610 | 0.320 | 0.056 |
| Idea Var | 0.262 | 0.154 | |
| Reviewer Var | 1.071 | 0.225 | |

Table 16: Results of linear mixed-effects models. We **bold** results that are statistically significant ($^*p < 0.05;^{**} p < 0.01;^{***} p < 0.001$). Our main conclusion on AI ideas being more novel than human ideas still holds here.

## A.17 SCORE BREAKDOWN BY TOPIC

We show the breakdown of all scores across all conditions by topic. Note that due to the smaller sample sizes for the per-topic breakdown, most results are not statistically significant and only offer an intuitive understanding of the trends.

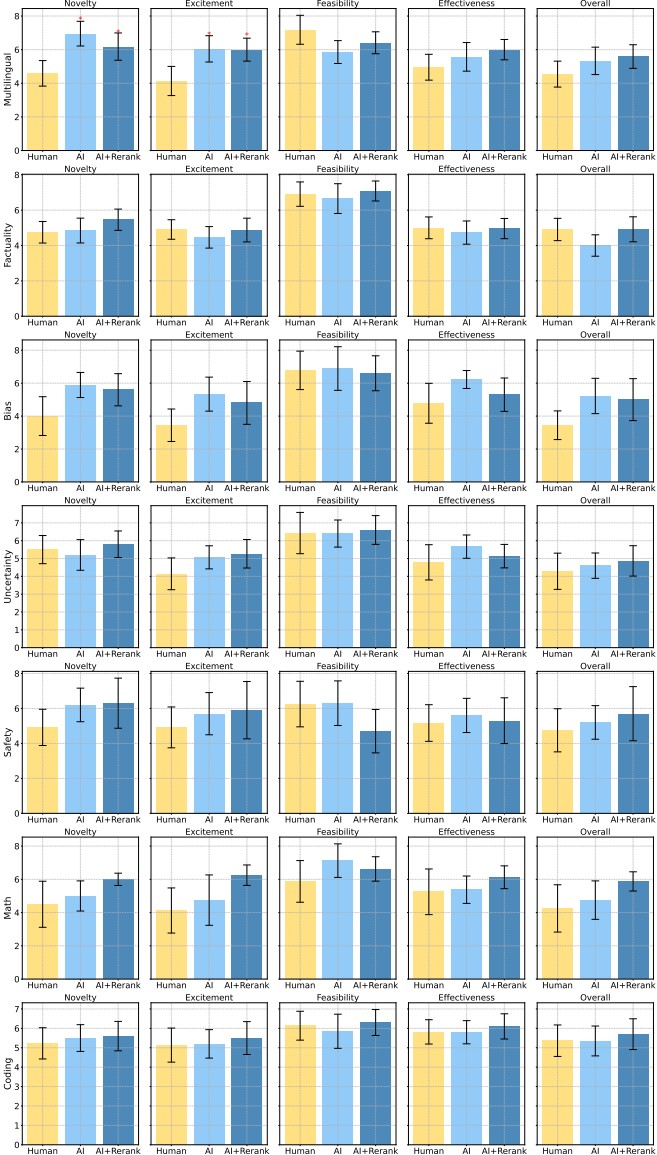

Figure 5: Breakdown of all scores by topic.

A.18   ANALYSIS OF FREE-TEXT REVIEWS

Following recent practices of using LLMs to extract patterns from text corpora (Zhong et al., 2022; 2023), we use Claude-3.5 to extract and cluster the main points from all reviews. We then manually verified and labeled each cluster.

Many reviews reinforce our quantitative finding that AI ideas tend to be more novel. For example, reviewers noted: "The idea of [...] is quite novel in an in-context learning setting.", "The idea of exploring [...] using an LLM-based iterative approach is novel.", "The idea of [...] when constructing prompts to improve cross-lingual transfer is one that I have not heard of before.", "I like the idea to [...], and think it will be helpful for other researchers in the community.", "Combining [...] is a unique way of attempting to preserve the gist of the information while likely losing specific identifiers.", and "Safeguarding using [...] is clearly novel. Similar ideas have not been seen in the related work.".

Next, we summarize some common failure modes of AI ideas:

1. **Being too vague on implementation details.** For example, one reviewer noted: "I'm not super clear on the details of this lattice and how the model will be prompted, so I'm not super sure how well the model will complete these subtasks and how well-suited this particular structure is to completing the overall task." and another reviewer noted: ""For analyzing the effectiveness of the method, the proposal only provides a very ad-hoc + hand-wavy suggestion to compare responses across predefined questions." In another case, the AI idea is criticized for not considering practical implementation details: "I think in each of the steps, there is something hard to execute. For example, in step Constellation Formation, how do we do the weighted sum?" Similarly, other reviews noted: "It's unclear how CLIP is connected to the language model and how training a CLIP model would enable the LM to understand images.", and "There's no mentioning on how to prompt the model to generate defensive strategies and refine the model's responses using these strategies." Such vagueness often makes it difficult for reviewers to make confident judgments: "Because this idea is too general and vague, I can't really answer the previous question. An idea needs a certain level of details to be determined if it fits for a conference/journal but this one misses them."

2. **Misuse of datasets.** For example: "I'm not sure about the datasets picked. StereoSet is not a QA dataset; it simply contains statements. Also, I don't understand why Dialogue NLI responses require empathy.", "I'm concerned the datasets proposed are the right test cases for security of the code (since they are really just ML/programming problems, not system-level programming).", and "the choice of datasets might not be the best to show the effect of incorporating multiple perspectives, especially TruthfulQA and ScienceQA, which seems to have a single correct interpretation and answer." In another example, the benchmark datasets chosen are considered too easy by the reviewer: "none of the chosen datasets (MATH, GSM8K, and MMLU) uses complicated math concepts".

3. **Missing or inappropriate baselines.** For example: "The proposed method needs to be compared to simply asking the model to think of one (or several) facts about the question before answering using more turns. This could be an additional baseline to verify the scoring process is meaningful." and "Although the proposal includes some baselines that should be compared to, it does not mention some methods which seem to do quite well with LLMs." Sometimes, "the chosen baselines may not be suitable", for example, because they are not directly comparable with the proposed method.

4. **Making unrealistic assumptions.** For example: "The assumption that model can (mostly) accurately flag its own hallucinations is quite tricky.", "there is a presupposed assumption that hallucinations in LLMs are ungrounded and independent of the data they are trained on, which is generally not considered true", "The big issue for the effectiveness of the proposed method is that, it asserts very strong assumptions on downstream tasks, such as there must exist only two extremes.", "Some assumptions (e.g., [...]) are unlikely to be true in practice, especially when low-resource languages and less represented cultures are included in the study.", and "A major assumption in this approach is that the model is able to [...]. However, [...]".

5. **Being too resource-demanding.** Despite the fact that we explicitly prompted the agent to consider feasibility when generating ideas, some of the generated ideas are still too resource-demanding. For example, one reviewer noted: "The biggest issue to feasibility

I see is that the project calls for fine-tuning BLOOM (See step 5). BLOOM has 176B parameters so it's going to take quite a lot of GPUs to fine-tune. From a systems perspective, I see this as causing delays." In some other cases, manual data annotation is being criticized for feasibility: "The bottleneck seems to be the dataset collection process if there are no existing datasets that fit the requirements of the paper.", and "the manual evaluation by native speakers or cultural experts could be time-consuming and resource-intensive".

6. **Not well-motivated.** For example: "Not well-motivated and there is not a clear intuition that this work can work to increase the factuality.", "And in general the method is not well-motivated and needs reasons why retrieving from model itself is meaningful by use cases or specific tasks.", and "The idea simply doesn't make sense to me. Given current LLMs' ability, I'm pretty sure they can simply recite code like inserting data to a binary search tree."

7. **Not adequately following existing best practices.** For example: "The proposal does not seem to include awareness of what has been previously tried, or more strategic ways to evaluate success/failures..."

We contrast these with some of the unique strengths and weaknesses of human ideas:

1. **Human ideas are generally more grounded in existing research and practical considerations, but may be less innovative.** For example, these ideas might be applying existing techniques to new problems: "Multilinguality as a debiasing method has already been considered in the literature, although not necessarily in the prompt engineering framework." Sometimes people apply incremental changes to existing techniques: "The overall idea shares quite a similar idea with program-of-thought (PoT). The only difference is that there is an additional step where an LLM is prompted to decide whether to use code or not." Some ideas try to combine existing techniques: "Query decomposition and RAG separately are well studied, if there is no existing work that combines both (which I'm not aware of), then it's reasonably novel." As some reviewers noted, human ideas tend to build on known intuitions and results: "There are already existing works on using available lexicons to improve the translation capabilities of LLMs in general."

2. **Human ideas tend to be more focused on common problems or datasets in the field.** For example: "The problem of models not handling negation properly is a very common problem, especially among propriety LMs such as claude-3-5-sonnet.", "The data exist. This project mainly entails plugging in these datasets to a prompt template and finetuning for a bit. There is little left unspecified, and it should be quite simple to execute on.", "I haven't found any work using this idea to solve this specific problem, but [...] is definitely not new.", and "While existing works have explored the problem of calibration in long-form answers (e.g. [...]), the specific method for calibration is different."

3. **Human ideas sometimes prioritize feasibility and effectiveness rather than novelty and excitement.** For example, reviewers noted: "I don't think this will be a groundbreaking finding, but it will probably work." and "while the idea is promising and could lead to significant improvements, it may not be groundbreaking enough to be considered transformative or worthy of a best paper award".

## A.19 RANDOMLY SAMPLED HUMAN AND AI IDEAS WITH REVIEWS

We randomly sample four pairs of ideas from different topics to ground our numerical results with actual examples. In each pair, there is one AI idea and one human idea. To save space, we include the full project proposal of each idea along with the full set of reviews in the Appendix, but we list their titles, topics, and average scores here for quick reference (we reveal whether each idea is AI-generated or human-written in Appendix A.28):

1. Modular Calibration for Long-form Answers: Appendix A.20
   Topic: Uncertainty; Average Overall Score: 5.5

2. Semantic Resonance Uncertainty Quantification: Calibrating LLM Confidence through Multi-Path Reasoning: Appendix A.21
   Topic: Uncertainty; Average Overall Score: 6

3. Translation with LLMs through Prompting with Long-Form Context: Appendix A.22
   Topic: Multilingual; Average Overall Score: 4

4. Linguistic Pivot Constellation: Enhancing Cross-Lingual Transfer for Low-Resource Languages and Dialects: Appendix A.23
   Topic: Multilingual; Average Overall Score: 6.7

5. LLM Directed Retrieval Querying for Improving Factuality: Appendix A.24
   Topic: Factuality; Average Overall Score: 4.7

6. Semantic Divergence Minimization: Reducing Hallucinations in Large Language Models through Iterative Concept Grounding: Appendix A.25
   Topic: Factuality; Average Overall Score: 3.3

7. Autoprompting: Generate Diverse Few-shot Examples for Any Application: Appendix A.26
   Topic: Coding; Average Overall Score: 5

8. Temporal Dependency Unfolding: Improving Code Generation for Complex Stateful Systems: Appendix A.27
   Topic: Coding; Average Overall Score: 6.7

## A.20 EXAMPLE IDEA: MODULAR CALIBRATION FOR LONG-FORM ANSWERS

---

**Modular Calibration for Long-form Answers (Part 1)**

**1. Problem Statement:** Calibrating the confidence of Large Language Models (LLMs) when generating long-form answers, such as essays and code, remains an open challenge in the field of natural language processing.

**2. Motivation:** While numerous methods have been developed to calibrate the performance of LLMs on multiple-choice questions or open-domain questions with short answers, extending these approaches to tasks requiring lengthy responses presents significant difficulties. For instance, in code generation tasks (e.g., the HumanEval dataset), traditional confidence extraction methods like perplexity may prove inadequate due to the substantial variation in answer length across questions. Verbalized confidence can be affected by instruction tuning artifacts or unclear scope, while the reliability of metrics such as Expected Calibration Error (ECE) and Macro-averaged Calibration Error (MacroCE) may be compromised by differences in task settings. Our aim is to propose a novel pipeline for confidence extraction and calibration of LLMs for long-form answers, drawing inspiration from methods used for short or fixed-set answers. This approach will enable us to monitor the model's long-form answer generation process and apply targeted external augmentation when necessary, thereby enhancing both performance and efficiency.

**3. Proposed Method:** We introduce Modular Calibration, a process comprising four core steps:

1. **Extend:** Prompt the model to elaborate on the original question in relation to the answer, identifying which components of the question are addressed in the long-form response.

2. **Decompose:** Instruct the LLM to break down the extended question and long-form answer into multiple modules.

3. **Extract Confidence:** Utilize verbalized confidence or perplexity to determine the confidence level for each module.

4. **Merge:** Based on the relationships between the modular questions/answers and the overall questions/answers, prompt the model to combine the modular confidence scores into an overall score representing the confidence in the long-form answer.

Each of these steps is executed by prompting the same LLM in different ways to elicit the desired response.

**4. Step-by-Step Experiment Plan:**

1. **Gather Datasets:** Select datasets featuring long answers with correctness annotations. Potential candidates include GSM8K, Code Gen, and Essay Writing.

2. **Construct Prompts:**

   (a) Establish a baseline using direct prompting, where a query is presented without special techniques.

   (b) Analyze outputs to refine prompts for the Extend and Decompose steps.

   (c) For the Confidence step, employ vanilla perplexity or verbalized confidence extraction. If performance is unsatisfactory, explore advanced methods built upon these techniques, such as those presented in recent research (e.g., FaR paper).

3. **Select Models:** Evaluate GPT-3.5 (Text-Davinci-003) and GPT-4 from the OpenAI API, as well as the open-source LLaMA-3-70B-chat.

4. **Get Results:** Obtain confidence predictions from the models on the selected datasets using both baseline methods and the proposed Modular Calibration approach.

5. **Analyze Results:** Compare the calibration performance of LLMs using the new method against the baselines (e.g., the perplexity of the entire long-form answer). Conduct qualitative and quantitative analyses on each component of the Modular Calibration process.

---

---

**Modular Calibration for Long-form Answers (Part 2)**

**5. Test Case Examples:**

- **Test Case 1: Verbalized Confidence Prompting**

  - Input: <Q> <A> Confidence (0-1)
  - Output: [Model generates a confidence score between 0 and 1]

- **Test Case 2: Modular Calibration Step 1 (Extend)**

  - Input: Given the answer, can you extend the question and elaborate on what points are covered in the answer?
  - Output: The answer covers these points of the question: (1) how fast A runs; (2) how fast B runs; (3) if A is faster than B.

- **Test Case 3: Modular Calibration Step 2 (Decompose)**

  - Input: Please decompose the above extended question and answers into modules.
  - Output:
    * How fast A runs: [relevant excerpt from the original answer]
    * How fast B runs: [relevant excerpt from the original answer]
    [Additional modules as needed]

- **Test Case 4: Modular Calibration Step 3 (Extract)**

  - Input: How fast A runs: [relevant excerpt from the original answer] Confidence (0-1)
  - Output: 1. 0.9; 2. 0.6 [Additional confidence scores for other modules]

- **Test Case 5: Modular Calibration Step 4 (Merge)**

  - Input: For each of these points related to question X, the confidence is: 0.9, 0.6, ... What is the overall confidence for the whole problem?
  - Output: [Model generates an overall confidence score]

**6. Fallback Plan:** If the proposed Modular Calibration method does not demonstrate improvement over the baseline, we will execute each sub-question and module individually to assess whether calibration is enhanced for each component. This approach will facilitate debugging of the proposed method and potentially yield interesting insights into the relationships between performance/calibration of decomposed modules and overall problems. Alternatively, we may analyze the model's ability to effectively decompose questions and answers into appropriate modules. These analyses will inform potential refinements to the method or provide valuable insights into the limitations and capabilities of LLMs in handling complex, long-form responses.

## Reviewer 1

**Novelty:** 6 (reasonably novel - there are some notable differences from existing ideas and probably enough to turn into a new paper)
**Rationale:** Focus on the long-form setting is novel at the moment. The idea of obtaining modular confidence estimates for different claims in a long-form output, and synthesizing them into a single uncertainty estimate is not that complicated, but it does seem to be underexplored.

**Feasibility:** 8 (Highly Feasible: Straightforward to implement the idea and run all the experiments.)
**Rationale:** The only part of the project that seems challenging is obtaining correctness annotations for one of the datasets (e.g., Essay Writing). GSM8K and code datasets like HumanEval seem like very natural long-form output settings to try out the idea. Other than this, iterating on the prompts for decomposition / verbalized UQ for each of the modules will be important, but the author mentions this.

**Expected Effectiveness:** 6 (Somewhat effective: There is a decent chance that the proposed idea can beat existing baselines by moderate margins on a few benchmarks.)
**Rationale:** It's possible that first obtaining verbalized uncertainty estimates for each module, and then synthesizing into a single score, will outperform the standard baselines of self-consistency over the entire long-form output (using majority vote as the confidence score). However, I don't expect this to be dramatically better. If the paper instead set out with the goal of actually producing the UQ estimates for each claim, then almost no prior work does this, and the baselines would be less strong.

**Excitement:** 5 (Leaning negative: it has interesting bits but overall not exciting enough)
**Rationale:** This seems like the most straightforward possible way to obtain uncertainty estimates for a long-form generation with an LLM. This means the project could produce some useful engineering artifacts, but it doesn't really push the idea to its logical conclusion. Therefore I don't consider it "exciting enough". There is some mention of "using the uncertainty estimates to possibly condition on more information" but this is not fleshed out – it could be more interesting. For example, studying how the fine-grained uncertainty estimates could be used to selectively retrieve factual information from Wikipedia etc. on a knowledge-intensive task.

**Overall Score:** 5 (Decent idea but has some weaknesses or not exciting enough, marginally below the acceptance threshold of major AI conferences)
**Rationale:** I like the focus on long-form generations. However, this proposal is a very straightforward baseline and extension of existing work to the long-form generation setting (just produce the long generation, decompose it, apply verbalized uncertainty on each claim, and finally aggregate them). I could see the paper being well-cited, but I don't see an interesting/novel angle here.

**Confidence:** 5 (You are absolutely certain that the evaluation is correct and very familiar with the relevant literature)

**Reviewer 2**

**Novelty:** 6 (reasonably novel - there are some notable differences from existing ideas and probably enough to turn into a new paper)
**Rationale:** While existing works have explored the problem of calibration in long-form answers (e.g. https://arxiv.org/abs/2402.06544), the specific method for calibration is different. Also seems related to FactScore (https://arxiv.org/abs/2305.14251) where the task was different (getting a factuality score) but the idea of breaking long form generations into smaller units, evaluating each separately and then combing does seem related.

**Feasibility:** 8 (Highly Feasible: Straightforward to implement the idea and run all the experiments.)
**Rationale:** The idea seems simple enough to implement with API access, considering all the steps involved in the method can be done via prompting with API. The proposal does mention using LLaMA3-70B as an additional model, which would require GPUs I guess.

**Expected Effectiveness:** 6 (Somewhat effective: There is a decent chance that the proposed idea can beat existing baselines by moderate margins on a few benchmarks.)
**Rationale:** Since it has been shown that LLMs are quite well calibrated when asked to verbalize the confidence for short answers, I'm guessing the calibration scores would be pretty good for individual modules. Also LLMs might be decent at combining confidence scores (especially with detailed instructions and some examples in the prompt), so overall the method might work well. But it's unclear if it would do better than the methods proposed in - https://arxiv.org/abs/2402.06544.

**Excitement:** 6 (Learning positive: exciting enough to be accepted at a major AI conference, but still has some weaknesses or somewhat incremental)
**Rationale:** If the method does work well in getting calibration for long-form answers, I think that would be pretty exciting. One thing which is missing from the proposal (and why the score was not higher) was that it does not touch upon the issue that for long-form answers we won't have a binary correct/incorrect decision but answers can be partially correct.

**Overall Score:** 6 (Marginally above the acceptance threshold of major AI conferences)
**Rationale:** The overall idea makes sense to me, but the score is not higher right now because: (a) it's unclear what exactly is meant by 'modules' especially for essay writing which the proposal mentions as one of the tasks ; (b) the issue for partial correctness which was mentioned above.

**Confidence:** 3 (You are fairly confident that the evaluation is correct)

### A.21 Example Idea: Semantic Resonance Uncertainty Quantification

---

**Semantic Resonance Uncertainty Quantification (SRUQ) (Part 1)**

**1. Problem Statement:** Current uncertainty quantification methods for Large Language Models (LLMs) often rely on simple statistical measures or model-specific attributes, which may not capture the nuanced semantic uncertainties in complex reasoning tasks. This limitation can lead to overconfident or poorly calibrated model outputs, potentially resulting in unreliable decision-making in critical applications.

**2. Motivation:** Existing approaches typically use softmax probabilities, entropy measures, or ensemble disagreement to quantify uncertainty. However, these methods often fail to capture the semantic nuances and reasoning complexities in tasks that require deep understanding and multi-step reasoning. Human experts, on the other hand, gauge their uncertainty by considering how well their reasoning 'resonates' with their broader knowledge and experience. By mimicking this process in LLMs, we can potentially develop a more robust and semantically grounded approach to uncertainty quantification.

**3. Proposed Method:** We propose Semantic Resonance Uncertainty Quantification (SRUQ), which prompts the LLM to generate multiple independent reasoning paths for a given problem, then quantifies uncertainty based on the semantic coherence and mutual reinforcement among these paths. The process involves five key steps:

1. Generating diverse solution attempts using different prompting strategies.

2. Cross-evaluating each solution attempt against the others, assessing logical consistency and mutual support.

3. Constructing a 'resonance graph' where nodes are solution attempts and edges represent semantic reinforcement.

4. Computing a resonance score based on graph properties like connectivity and centrality.

5. Mapping the resonance score to a calibrated uncertainty estimate.

---

**Semantic Resonance Uncertainty Quantification (SRUQ) (Part 2)**

**4. Step-by-Step Experiment Plan:**

1. **Dataset Preparation**

    - Utilize three datasets covering different reasoning tasks:
        (a) GSM8K for mathematical problem-solving
        (b) EntailmentBank for logical deduction
        (c) HotpotQA for multi-hop question answering
    - Split each dataset into train, validation, and test sets if not already done.

2. **Baseline Implementation**

    - Implement three baseline uncertainty quantification methods:
        (a) Softmax probabilities
        (b) Monte Carlo Dropout
        (c) Ensemble disagreement (using different few-shot prompts)
    - Generate predictions and uncertainty estimates on the validation and test sets for each baseline.

3. **SRUQ Implementation**

    (a) Generate 5 diverse solution attempts using different few-shot prompts and temperature settings.
    (b) For each pair of solutions, prompt the LLM to evaluate their consistency and mutual support.
    (c) Construct the resonance graph using the pairwise evaluations.
    (d) Compute the resonance score using graph centrality measures (e.g., PageRank).
    (e) Map the resonance score to a calibrated uncertainty estimate using isotonic regression on the validation set.

4. **Evaluation**

    - Compare SRUQ against the baselines using the following metrics:
        (a) Expected Calibration Error (ECE)
        (b) Brier score
        (c) Area Under the Precision-Recall Curve (AUPRC) for uncertainty ranking
    - Evaluate the correlation between uncertainty estimates and actual errors.

5. **Analysis**

    - Visualize the resonance graphs for high and low uncertainty examples.
    - Analyze the relationship between graph properties and prediction accuracy.
    - Investigate cases where SRUQ significantly outperforms or underperforms compared to baselines.

6. **Ablation Studies**

    - Vary the number of solution attempts.
    - Compare different graph centrality measures.
    - Evaluate the impact of the cross-evaluation step.

7. **Generalization Test**

    - Test the generalization of SRUQ on out-of-distribution samples by applying the method trained on one dataset to examples from the other datasets.

---

**Semantic Resonance Uncertainty Quantification (SRUQ) (Part 3)**

**5. Test Case Examples:**

- **Baseline Example:**
    - Input: Q: If a train travels at 60 miles per hour, how far will it travel in 2.5 hours?
    - Softmax Output: The train will travel 150 miles in 2.5 hours. (Confidence: 0.92)
    - Explanation: The softmax probability is high, but it does not capture the reasoning process or potential uncertainties in the calculation.

- **SRUQ Example:**
    - Input: Q: If a train travels at 60 miles per hour, how far will it travel in 2.5 hours?
    - Solution Attempts:
        1. Distance = Speed × Time
            * Distance = 60 miles/hour × 2.5 hours
            * Distance = 150 miles
        2. In 1 hour, the train travels 60 miles
            * In 2 hours, it's 120 miles
            * In 0.5 hours, it's 30 miles
            * Total: 120 + 30 = 150 miles
        3. Let's break it down:
            * 1 hour: 60 miles
            * 1 hour: 60 miles
            * 0.5 hour: 30 miles
            * Sum: 60 + 60 + 30 = 150 miles
    - Cross-Evaluation: All three solutions are consistent and mutually supportive. They use different approaches but arrive at the same result.
    - Resonance Graph: Fully connected graph with high edge weights
    - Resonance Score: 0.95
    - Calibrated Uncertainty: 0.05
    - Final Output: The train will travel 150 miles in 2.5 hours. (Uncertainty: 0.05)
    - Explanation: SRUQ generates multiple solution paths, evaluates their consistency, and quantifies uncertainty based on their semantic resonance. The high resonance score indicates low uncertainty, which is then calibrated to provide a final uncertainty estimate.

**6. Fallback Plan:** If SRUQ does not significantly outperform baselines, we can pivot to an analysis paper exploring why semantic resonance might not capture uncertainty effectively. We could investigate the quality and diversity of generated solution attempts, potentially improving the prompting strategies. Additionally, we could examine the effectiveness of the cross-evaluation step, possibly incorporating external knowledge or more structured reasoning. Furthermore, we could explore the relationship between graph properties and actual uncertainty, which might reveal insights about how LLMs represent confidence internally. We could also consider combining SRUQ with traditional uncertainty quantification methods, creating a hybrid approach that leverages both statistical and semantic information.

## Reviewer 1

**Novelty:** 6 (reasonably novel - there are some notable differences from existing ideas and probably enough to turn into a new paper)
**Rationale:** I haven't seen (and couldn't find) any prior work which exactly has the same idea as in this proposal. The proposed idea is definitely related to using consistency among multiple solutions to estimate uncertainty (e.g. https://arxiv.org/abs/2405.18711 does this across solutions decoded from different layers) but I have not seen the idea of constructing resonance graph and using graph properties to estimate uncertainty.

**Feasibility:** 8 (Highly Feasible: Straightforward to implement the idea and run all the experiments.)
**Rationale:** The proposed method, SRUQ, should be pretty easy to implement given that LLM API access is abundant. SRUQ involves multiple steps all of which can be done through prompting via API — getting multiple solutions, prompting LLMs to get a consistency score between each pair of solutions etc. The parts which cannot be implemented through API are the baselines e.g. Monte Carlo dropout, and would require GPUs. To do a fair comparison to the baselines, I imagine SRUQ will also have to be done on open models which could also require GPUs.

**Expected Effectiveness:** 6 (Somewhat effective: There is a decent chance that the proposed idea can beat existing baselines by moderate margins on a few benchmarks.)
**Rationale:** Although the proposal includes some baselines that should be compared to, it does not mention some methods which seem to do quite well with LLMs (especially getting better with scale) – e.g. methods like P(True) (https://arxiv.org/abs/2207.05221) or verbalized confidence (https://arxiv.org/abs/2305.14975). It's not clear/obvious to me that the proposed method should do better than these baselines.

**Excitement:** 6 (Learning positive: exciting enough to be accepted at a major AI conference, but still has some weaknesses or somewhat incremental)
**Rationale:** While the method is novel and feasible, I'm not too excited by it since some of the other existing methods out there mentioned above (like https://arxiv.org/abs/2207.05221, https://arxiv.org/abs/2305.14975) are much simpler and work quite well. Compared to that SRUQ is more complex, and hence maybe has less chance of being very impactful (unless it works really better).

**Overall Score:** 6 (Marginally above the acceptance threshold of major AI conferences)
**Rationale:** The above accept score is assuming the idea does work better than the baselines on some category of tasks. Overall, given that the idea is novel, the proposal includes comparison to other baselines as well analysis & ablations, I think that could be enough to get accepted into an AI conference.

**Confidence:** 4 (You are confident but not absolutely certain that the evaluation is correct)

**Reviewer 2**

**Novelty:** 6 (reasonably novel - there are some notable differences from existing ideas and probably enough to turn into a new paper)
**Rationale:** The proposed approach shares some similar ideas with self-consistency (which suggests the consistency of sampled LLMs outputs is relatively well calibrated). But the approach is more generalized and fine-grained than existing work if the approach uses more advanced 'mutual support evaluation' beyond simply comparing the final answers.

**Feasibility:** 5 (Moderately feasible: It can probably be executed within the given time frame but would require careful planning, efficient use of APIs or some advanced computational strategies to overcome the limited GPU resources, and would require some modifications to the original proposal to make it work.)
**Rationale:** There lacks some important details in terms of the cross-evaluation part. How is the mutual support evaluated (by prompting or some other methods?). This part is crucial for implementing the whole pipeline of this approach.

**Expected Effectiveness:** 6 (Somewhat effective: There is a decent chance that the proposed idea can beat existing baselines by moderate margins on a few benchmarks.)
**Rationale:** I think it has some chances to beat the proposed baselines. If the cross-evaluation part is properly executed. Again, the success of this proposal is highly dependent on that part.

**Excitement:** 6 (Learning positive: exciting enough to be accepted at a major AI conference, but still has some weaknesses or somewhat incremental)
**Rationale:** If this idea actually works, at least it tells something new about how to use multiple samples to provide better confidence estimation than simple consistency. But the idea itself is still somewhat incremental given the existence of current consistency-based calibrators.

**Overall Score:** 6 (Marginally above the acceptance threshold of major AI conferences)
**Rationale:** Overall there are some incremental contributions, but not too exciting. The algorithm itself can be neat. I think it can be worth a borderline acceptance.

**Confidence:** 4 (You are confident but not absolutely certain that the evaluation is correct)

**Reviewer 3**

**Novelty:** 6 (reasonably novel - there are some notable differences from existing ideas and probably enough to turn into a new paper)
**Rationale:** I think the idea is reasonable and indeed identifies some limitations of current works on uncertainty estimation. However, the consistency between reasoning paths is somehow similar to self-consistency reasoning from Google and SelfCheckGPT.

**Feasibility:** 7
**Rationale:** I think it could be easy to implement and quickly be tried by PhD students or even undergrads. Also, in the test case example, the setting is straightforward and well-defined.

**Expected Effectiveness:** 6 (Somewhat effective: There is a decent chance that the proposed idea can beat existing baselines by moderate margins on a few benchmarks.)
**Rationale:** Based on my experience, the consistency-based methods, although not fully theoretically grounded, can work pretty well in current uncertainty estimation questions. I believe working this on the reasoning path level could also work to some extent.

**Excitement:** 6 (Learning positive: exciting enough to be accepted at a major AI conference, but still has some weaknesses or somewhat incremental)
**Rationale:** Overall, this idea identified a good research question, although the method might not be very exciting to me.

**Overall Score:** 6 (Marginally above the acceptance threshold of major AI conferences)
**Rationale:** The novelty and the actual application of this method in the area is limited, but could be an inspiring idea.

**Confidence:** 4 (You are confident but not absolutely certain that the evaluation is correct)

## A.22 EXAMPLE IDEA: TRANSLATION WITH LLMS THROUGH PROMPTING WITH LONG-FORM CONTEXT

---

### Translation with LLMs through Prompting with Long-Form Context (Part 1)

**1. Problem Statement:** Stable generation of text in low-resource languages is an unsolved issue in large language models.

**2. Motivation:** While LLMs can often produce surprisingly good translations despite not being explicitly trained for this task, this does not hold for lower-resource languages. LLMs are both more likely to generate off-target text (text in another language than intended) when prompted to translate to a lower-resource language, and show increased instability in translation quality across prompt templates in lower-resource languages.

**3. Proposed Method:** Our proposed method investigates the use of long-form templates to improve generated translation quality and reduce off-target translations in lower-resource languages. We propose to provide additional prompt context by translating multi-sentence input, with additional views of the target language with the langid template provided as context. We do so in multiple stages:

1. **Querying the language model** to first generate a paragraph containing the source sentence to be translated.
2. **Prepending monolingual text in the target language**, with langid: tags, above the translation prompt.
3. **Presenting both these additional sources of content**, prompting the LLM for a translation.

**4. Step-by-Step Experiment Plan:**

1. **Choose datasets:** Evaluate on the FLORES-200 datasets, which allow for wide language coverage on the Wikipedia domain, as well as the WMT-21 test sets for news and law/medical domain.

2. **Choose languages:** Opt for English-centric translation with:
   - 5 high-resource languages with different scripts (French, German, Russian, Chinese, Japanese)
   - 5 mid-resource languages (Farsi, Vietnamese, Arabic, Korean, Hebrew)
   - 5 low-resource languages with considerably lower likelihood of incidental bilingualism (Gujarati, Thai, Tajik, Sindhi, Pashto)

3. **Choose models:** Include the API-based GPT-3.5 (Text-Davinci-003) and GPT-4 model from OpenAI and Gemini from Google, as well as the open-weight LLaMA-3, Gemma, and Aya models which enable additional analysis.

4. **Gather translation results:** Systematically compare standard MT prompt templates to our proposed method across different models and language pairs. Additionally ablate the steps of the new method (removing langid templates; replacing langid templates with endonymic langid tags; provide only the generated paragraph; only the monolingual content).

5. **Perform analysis:** Evaluate whether the new method improves the performance of LLMs in these tasks as compared to the baselines using multiple standard automatic metrics for MT (chrF, COMET, BLEU) and token-level LID to measure off-target translations. Assess which component(s) are necessary for this improvement and whether or not there are changes across language pair direction and language resource levels or scripts.

---

**Translation with LLMs through Prompting with Long-Form Context (Part 2)**

**5. Test Case Example:**

- **Baseline Prompt Template:**

```
[English] This is an example.
[Spanish]
```

- **Baseline Prompt Expected Output:**

```
[Spanish] Este es un ejemplo.
```

- **Proposed Prompt Input (step 1):**

```
Write a paragraph containing the following sentence:
[English] This is an example.
```

- **Expected model output (step 1):**

```
This is an example link. Italics and Bold: use asterisks and
underscores to indicate spans of emphasis. Use one asterisk
(or underscore) for italics and two asterisks (or two
underscores) for bold. For example...
```

- **Proposed Prompt Input (step 2):**

```
[Spanish] Computadoras simples son lo suficientemente pequeñas
para residir en los dispositivos móviles.
[Spanish] Las computadoras portátiles, tales come tabletas,
netbooks, notebooks, ultrabooks, pueden ser alimentadas por
pequeñas baterías.
[Spanish] Las computadoras personales en sus diversas formas
son iconos de la llamada era de la información y son lo que la
mayoría de la gente considera como «computadora».

Translate this paragraph from English to Spanish:
[English] This is an example link. Italics and Bold: use
asterisks and underscores to indicate spans of emphasis. Use
one asterisk (or underscore) for italics and two asterisks
(or two underscores) for bold. For example...
[Spanish]
```

- **Proposed Prompt Expected Output:**

```
[Spanish] Este es un enlace de ejemplo. Cursiva y negrita:
utilice asteriscos y guiones bajos para indicar intervalos
de énfasis. Utilice un asterisco (o guión bajo) para cursiva
y dos asteriscos (o dos guiones bajos) para negrita.
Por ejemplo...
```

**Translation with LLMs through Prompting with Long-Form Context (Part 3)**

**5. Test Case Example (Continued):**

- **Proposed Prompt Input (step 3):**

```
Which of the following sentences are translations of the
following English sentence? Multiple sentences can be chosen.
[English] This is an example.

1. Este es un enlace de ejemplo.
2. Cursiva y negrita: utilice asteriscos y guiones bajos para
indicar intervalos de énfasis.
3. Utilice un asterisco (o guión bajo) para cursiva y dos
asteriscos (o dos guiones bajos) para negrita.
4. Por ejemplo...
```

- **Proposed Prompt Expected Output:**

```
The sentence "This is an example." can be translated to
Spanish as:
1. Este es un ejemplo.
2. Por ejemplo...
These two options correctly translate the meaning of
"This is an example." into Spanish.
```

**6. Fallback Plan:** If the proposed method does not help as compared to the baseline, analyzing the results of step 3 would likely provide further insights into how the template should be modified. In addition to potentially identifying off-target errors, it may be that the model is unable to identify correct translations even if they have been generated, and results are likely to vary across languages based on their training data. Using the generated paragraph as provided context and still querying the model to translate at only the sentence level could be compared. Restricting monolingual text to be retrieved text within the domain of the source sentence could be explored. Adding few-shot examples in the prompt and comparing other MT prompt templates may also help debug the proposed method. Including an additional query where the model is first asked to label each generated token by langid and then asked to re-translate the source including those tokens which are correctly labelled in target may reinforce langid and guide generation in the target language. Performing layer-wise analyses of likelihood of generating the next token in-language and in-script for open-weight models may also help debug where and why off-target issues persist.

**Reviewer 1**

**Novelty:** 5 (somewhat novel - there are differences from existing ideas but not enough to turn into a new paper)
**Rationale:** While I'm not aware of papers that have used this exact prompting strategy, I don't think that this proposal will be enough to justify a publication. I think that there should be a variety of strategies suggested + an analysis of multiple prompting strategies rather than suggesting one strategy. I think that a thorough analysis of the effects of additional context / langids could potentially turn this into a paper.

**Feasibility:** 9
**Rationale**: Such a project that only uses LLM APIs could be executed very quickly without much expertise in coding/architecture. The only time-consuming part might be iterating and adjusting the prompts in the ablation studies.

**Expected Effectiveness:** 7
**Rationale:** I think that this proposal could work well to guide LLMs to translate in the desired target language, since this is a known problem with current prompt-based MT strategies (as the writers have suggested).

**Excitement:** 5 (Leaning negative: it has interesting bits but overall not exciting enough)
**Rationale:** I'm not sure how well this method will transfer to future models, and this could be a limiting factor in the longevity of this research. (But this is a limitation of all prompting research...)

**Overall Score:** 5 (Decent idea but has some weaknesses or not exciting enough, marginally below the acceptance threshold of major AI conferences)
**Rationale:** I think that the work should focus on the ablation studies and comparison of multiple prompting strategies / analysis, rather than focusing on one new strategy.

**Confidence:** 3 (You are fairly confident that the evaluation is correct)

**Reviewer 2**

**Novelty:** 1 (not novel at all - there are many existing ideas that are the same)
**Rationale:** There are multiple existing works on prompting LLMs on low-resource translation, usually using few-shot demo. https://proceedings.mlr.press/v202/garcia23a/garcia23a.pdf https://arxiv.org/pdf/2305.14857 Also work explaining why few-shot prompt would work: https://arxiv.org/pdf/2305.10266

**Feasibility:** 5 (Moderately feasible: It can probably be executed within the given time frame but would require careful planning, efficient use of APIs or some advanced computational strategies to overcome the limited GPU resources, and would require some modifications to the original proposal to make it work.)
**Rationale:** The prompting experiment is mostly feasible given one can afford the API calls. The model, prompts, and evaluation metrics are concrete, although unclear if the proposed experiment is useful for proving the research idea, e.g., a few high-resource languages are listed for a research idea that focuses on low-resource languages.

**Expected Effectiveness:** 3 (Low Effectiveness: The idea might work in some special scenarios but you don't expect it to work in general.)
**Rationale:** The proposed experiment can help find a set of relatively high-performing prompts, but it is unclear among the prompts proposed if any of them will bring any improvement.

**Excitement:** 3 (Mediocre: this idea makes marginal contributions and is very incremental)
**Rationale:** The ability to do prompting/few-shot translation is fundamentally tied to the training data, see https://arxiv.org/pdf/2305.10266, so trying to solve this problem from the prompting space is inherently limited.

**Overall Score:** 3 (Clear rejection for major AI conferences)
**Rationale:** There is similar work on prompting LLMs to generate translation in low-resource languages, hence the idea is not very novel. Moreover, in terms of the goal to generate high-quality low-resource translation, the gains likely are not going to come from prompting.

**Confidence:** 4 (You are confident but not absolutely certain that the evaluation is correct)

A.23  EXAMPLE IDEA: LINGUISTIC PIVOT CONSTELLATION: ENHANCING CROSS-LINGUAL TRANSFER FOR LOW-RESOURCE LANGUAGES AND DIALECTS

---

### Linguistic Pivot Constellation (LPC): Enhancing Cross-Lingual Transfer for Low-Resource Languages and Dialects (Part 1)

**1. Problem Statement:** Large language models struggle with cross-lingual transfer, especially for low-resource languages and dialects. This limitation hinders the models' ability to perform well on multilingual tasks involving these languages, potentially exacerbating digital language divides.

**2. Motivation:** Current approaches often rely on parallel data or multilingual pretraining, which are limited for many language pairs. Inspired by how polyglots leverage similarities between known languages to learn new ones, we propose creating a network of conceptual bridges across languages. This method could potentially overcome the limitations of existing approaches by leveraging the model's broad knowledge to create connections between known and unknown linguistic territories.

**3. Proposed Method:** We introduce Linguistic Pivot Constellation (LPC), a novel prompting technique that constructs a dynamic network of linguistic pivot points. For a given task, LPC first identifies conceptually similar languages or dialects to the target language. It then generates a constellation of prompts in these pivot languages, each capturing a different aspect of the task. The model is guided to 'triangulate' the correct response by considering these multiple perspectives. For example, to translate a rare dialect, LPC might use prompts in related languages, regional lingua francas, and even etymologically connected languages.

**4. Step-by-Step Experiment Plan:**

1. **Data Collection**
   - Gather datasets for translation and question-answering tasks across a diverse set of low-resource languages and dialects.
   - Utilize the FLORES-101 dataset for machine translation and the TyDi QA dataset for question answering.

2. **Baseline Implementation**
   - Implement standard few-shot prompting and existing cross-lingual transfer methods (e.g., zero-shot cross-lingual transfer) as baselines.

3. **LPC Implementation**
   (a) Create a language similarity matrix based on language families and geographical proximity.
   (b) Implement a function to select the most relevant pivot languages for a given target language.
   (c) Design prompts for each pivot language that capture different aspects of the task.

4. **Prompt Construction**
   (a) Select 3-5 pivot languages based on the similarity matrix.
   (b) Generate task-specific prompts in each pivot language.
   (c) Combine these prompts into a 'constellation' prompt that includes the original task in the target language.

5. **Model Selection**
   - Use GPT-4 as the primary model for experiments.
   - Test with GPT-3.5-turbo for comparison.

6. **Experiment Execution**
   (a) Run the baseline methods.
   (b) Run the LPC method with varying numbers of pivot languages (1, 3, and 5).
   (c) Record the model outputs and performance metrics.

**Linguistic Pivot Constellation (LPC): Enhancing Cross-Lingual Transfer for Low-Resource Languages and Dialects (Part 3)**

**4. Step-by-Step Experiment Plan (Continued):**

7. **Evaluation**

   - Evaluate the results using task-specific metrics:
     - BLEU score for translation tasks
     - F1 score for question answering tasks

8. **Analysis**

   - Analyze the effectiveness of different pivot language combinations and the method's scalability to extremely low-resource scenarios.
   - Compare LPC performance against baselines across different language families and resource levels.

**5. Test Case Examples:**

- **Test Case 1:**

  - **Baseline Prompt Input:** Translate the following Sicilian sentence to English: 'Unni c'è fumu c'è focu.'
  - **Baseline Prompt Expected Output:** Where there's smoke, there's fire.
  - **Proposed Prompt Input:** We will translate a Sicilian sentence to English. To help with this task, consider the following related phrases:

    ```
    In Italian: 'Dove c'è fumo c'è fuoco.'
    In Neapolitan: 'Addò ce sta 'o fummo ce sta 'o ffuoco.'
    In Latin: 'Ubi fumus, ibi ignis.'
    ```

    Now, translate the Sicilian sentence to English: 'Unni c'è fumu c'è focu.'
  - **Proposed Prompt Expected Output:** Where there's smoke, there's fire.
  - **Explanation:** The LPC method provides context from related languages (Italian, Neapolitan, and Latin), which can help the model better understand and translate the Sicilian phrase. This is especially useful for low-resource languages like Sicilian, where direct translation data might be limited.

**6. Fallback Plan:** If the LPC method does not significantly outperform baselines, we will pivot the project towards an in-depth analysis of cross-lingual transfer mechanisms. We will investigate the relationship between language similarity and transfer effectiveness, the impact of pivot language selection on performance, and how different aspects of language (lexical, syntactic, semantic) transfer across the constellation. This analysis could provide valuable insights into the strengths and limitations of large language models in cross-lingual tasks, potentially informing future research directions in multilingual Natural Language Processing.

**Reviewer 1**

**Novelty:** 9
**Rationale:** The idea of using a linguistic similarity matrix to form conceptual bridges when constructing prompts to improve cross-lingual transfer is one that I have not heard of before. I think this could be an interesting way of leveraging existing information about related languages for NLP tasks in general.

**Feasibility:** 8 (Highly Feasible: Straightforward to implement the idea and run all the experiments.)
**Rationale:** I think the idea makes sense, but more details should be shared about how exactly this language similarity matrix is constructed and what algorithms will be used for determining language similarity. More details should be provided on how the prompts for different languages will be obtained and how the data will be collected, which might be a time bottleneck.

**Expected Effectiveness:** 6 (Somewhat effective: There is a decent chance that the proposed idea can beat existing baselines by moderate margins on a few benchmarks.)
**Rationale:** I think that this idea could work well just by providing more context in different languages. The effectiveness sounds like it might be highly variable on the selection of pivot languages, though.

**Excitement:** 7
**Rationale:** I think that this could be interesting beyond the context of prompting, such as the use of pivot languages in traditional machine translation.

**Overall Score:** 7 (Good idea, would be accepted by major AI conferences)
**Rationale:** I think that the idea is sufficiently novel, and if it is executed well with good results, could produce a quality paper at a top NLP conference.

**Confidence:** 3 (You are fairly confident that the evaluation is correct)

### Reviewer 2

**Novelty:** 8 (clearly novel - major differences from all existing ideas)
**Rationale:** The LPC method introduces a novel way of leveraging related languages and dialects to improve cross-lingual transfer. While cross-lingual transfer and language similarity have been explored, the idea of dynamically creating a constellation of prompts using pivot languages for specific tasks is a fresh and innovative approach.

**Feasibility:** 5 (Moderately feasible: It can probably be executed within the given time frame but would require careful planning, efficient use of APIs or some advanced computational strategies to overcome the limited GPU resources, and would require some modifications to the original proposal to make it work.)
**Rationale:** Implementing LPC could be challenging due to the complexities involved in selecting optimal pivot languages and designing effective prompts for each. While the concept is sound, the practical execution—such as building the language similarity matrix and dynamically generating prompts—may require substantial effort and experimentation.

**Expected Effectiveness:** 6 (Somewhat effective: There is a decent chance that the proposed idea can beat existing baselines by moderate margins on a few benchmarks.)
**Rationale:** The LPC method has the potential to improve cross-lingual performance, especially in low-resource languages. By leveraging linguistic similarities, the model might better understand and translate languages with limited training data.

**Excitement:** 7
**Rationale:** The LPC method is exciting because it tackles a critical challenge in multilingual NLP—improving performance for low-resource languages. If successful, it could significantly enhance the accessibility and usability of AI models across diverse linguistic contexts, particularly in underrepresented languages.

**Overall Score:** 6 (Marginally above the acceptance threshold of major AI conferences)
**Rationale:** The idea is a promising candidate for exploration in the field of multilingual NLP. It introduces a novel approach that could potentially improve cross-lingual transfer, particularly for low-resource languages and dialects. However, the challenges in implementation and the uncertain effectiveness of the method warrant a cautious overall rating.

**Confidence:** 4 (You are confident but not absolutely certain that the evaluation is correct)

**Reviewer 3**

**Novelty:** 8 (clearly novel - major differences from all existing ideas)
**Rationale:** Leveraging language similarity is often quite well studied in machine translation, but there hasn't been one studying using similar language as demonstration in multilingual in-context learning. It would be interesting to see how the model behavior change with different pivots.

**Feasibility:** 8 (Highly Feasible: Straightforward to implement the idea and run all the experiments.)
**Rationale:** The implementation will mostly involve building the similarity matrix and formatting the prompts. The similarity matrix should be able to get from some existing works. The prompt formatting and experiments part should be pretty straightforward with enough API quota.

**Expected Effectiveness:** 6 (Somewhat effective: There is a decent chance that the proposed idea can beat existing baselines by moderate margins on a few benchmarks.)
**Rationale:** The idea is pretty interesting, but it's not exactly sure whether similar languages are informative enough for the model, since it still requires the model to understand the similarity between languages and reason over the relationship between target language and the given languages.

**Excitement:** 8 (Exciting: would deepen the community's understanding or make major progress in this research direction)
**Rationale:** It would be informative to the community to see whether such demonstration can lead to good performance for in-context learning. Even if this idea doesn't work, the analysis will be quite informative.

**Overall Score:** 7 (Good idea, would be accepted by major AI conferences)
**Rationale:** This work studies an important problem for the multilingual community. The experiment results and analysis will be quite informative for multilingual in-context learning.

**Confidence:** 4 (You are confident but not absolutely certain that the evaluation is correct)

A.24 EXAMPLE IDEA: LLM DIRECTED RETRIEVAL QUERYING FOR IMPROVING FACTUALITY

---

**LLM Directed Retrieval Querying for Improving Factuality (Part 1)**

**1. Problem Statement:** Large language models can generate flexible, long-form language generations, but LLM-generated responses often contain hallucinated or factually inconsistent content. Particularly in high-risk settings, there is a need for methods to improve the factuality of LLMs.

**2. Motivation:** A common framework for improving the factuality of LLM generations is retrieval augmented generation (RAG). In a RAG framework, a retriever takes a query as input and retrieves external knowledge from a high-quality knowledge base from reliable sources. The retrieved content is incorporated into the prompt for generating the response. One issue with this approach is that the quality of the generation can be bottlenecked by the quality of the retrieved content. Retrieval can be challenging for tasks where the query objective is underspecified or additional reasoning (or multi-step reasoning) on the query is required to retrieve content that supports the query.

**3. Proposed Method:** Our method refines the query by using an LLM to decompose the problem into sub-questions and generate candidate answers to expand each sub-question. The key steps include:

1. Decomposing the original question into sub-questions using an LLM.
2. Generating candidate answers for each sub-question using the LLM.
3. Expanding each sub-question with generated candidate answers to create retrieval queries.
4. Retrieving passages for each expanded query.
5. Filtering retrieved passages based on retrieval model score.
6. Aggregating filtered passages across sub-questions.
7. Prompting the generative LLM with the aggregated passages as context to answer the original question.

**4. Step-by-Step Experiment Plan:**

1. **Choose RAG datasets** where the retrieval task has underspecified/unique objectives or requires multi-hop reasoning, such as BIRCO and HotpotQA.
2. **Select a retriever,** such as an E5 or BGE model, and a generative LLM, such as GPT or LLaMA-3.
3. **Establish Baseline:**
   (a) Use the example question as the query to the retriever to retrieve relevant content from the retrieval passage pool.
   (b) Construct a prompt that provides the retrieved context passages and the question.
   (c) Prompt the generative LLM to answer the question using the context.
4. **Implement Proposed Method:**
   (a) Prompt the generative LLM to decompose the question into sub-questions.
   (b) For each sub-question, prompt the generative LLM to generate candidate answers.
   (c) Use semantic similarity to cluster the generated candidate answers and sample for semantic diversity.
   (d) Construct retrieval queries by expanding each sub-question with sampled candidate answers.
   (e) Retrieve passages using each query and aggregate results for each sub-question.
   (f) Deduplicate retrieved passages and filter based on retrieval model score.
   (g) Prompt the generative LLM with filtered passages as context to answer the original question.

---

---

**LLM Directed Retrieval Querying for Improving Factuality (Part 2)**

**5. Test Case Examples:**

- **Test Case 1:**
  - **Original Question:** In which region is the village after which lager "Fucking Hell" is named?
  - **Baseline:**
    * **Retrieval Query:** In which region is the village after which lager "Fucking Hell" is named?
    * **Retrieved Passage:** Fucking Hell is a German pale lager, a Pilsner, with an alcohol content of 4.9%. It is named after Fucking, the previous name of the village of Fugging in Austria; hell is the German word for 'pale' and a typical description of this kind of beer. The beer's name was initially controversial. Both the local authorities in Fucking and the European Union's Trade Marks and Designs Registration Office initially objected to the name. It was eventually accepted and the lager is sold internationally.
    * **Prompt:** Given the retrieved passage(s) as context and the question, answer the question using the context.
    * **Answer:** The village after which the lager "Fucking Hell" is named is located in Austria.
  - **Proposed Method:**
    * **Sub-Questions:**
      1. What village is the lager "Fucking Hell" named after?
      2. In which country is this village located?
      3. In which specific region or state within that country is the village located?
    * **Example Retrieval Query:** What village is the lager "Fucking Hell" named after? The lager "Fucking Hell" is named after the village previously known as Fucking, which is now called Fugging, in Austria.
    * **Retrieved Passages:**
      1. Fucking Hell is a German pale lager, a Pilsner, with an alcohol content of 4.9%. It is named after Fucking, the previous name of the village of Fugging in Austria; hell is the German word for 'pale' and a typical description of this kind of beer. The beer's name was initially controversial. Both the local authorities in Fucking and the European Union's Trade Marks and Designs Registration Office initially objected to the name. It was eventually accepted and the lager is sold internationally.
      2. Fugging, spelled Fucking until 2021, is an Austrian village in the municipality of Tarsdorf, located in the Innviertel region of western Upper Austria. It is 33 km (21 mi) north of Salzburg and 4 km (2.5 mi) east of the Inn river, which forms part of the German border.
    * **Prompt:** Given the retrieved passage(s) as context and the question, answer the question using the context.
    * **Answer:** The village after which the lager "Fucking Hell" is named is located in the Innviertel region of western Upper Austria.

**6. Fallback Plan:** If the proposed method does not satisfy the success criteria, alternative approaches could be explored. These may include quantifying the difficulty of various examples and analyzing whether this correlates with method improvement. The method is likely to be more effective for questions about esoteric facts, where the model is less likely to have internal knowledge of the answer, or its generated answers are more likely to disagree. Additionally, the method may be more beneficial for questions requiring information from multiple passages. Further analysis could help debug why the proposed method did not work, informing alternative new methods or transforming the project into an analysis paper by offering interesting ablations and insights.

### Reviewer 1

**Novelty:** 1 (not novel at all - there are many existing ideas that are the same)
**Rationale:** I find this idea is extremely similar to "GenDec: A robust generative Question-decomposition method for Multi-hop reasoning" by Wu et al. (2024). Link: https://arxiv.org/html/2402.11166v1

**Feasibility:** 8 (Highly Feasible: Straightforward to implement the idea and run all the experiments.)
**Rationale:** Technically, this idea can be quickly re-produced based on the aforementioned paper. Though the motivations and evaluations are different from the existing work, it shouldn't take too long to figure them out.

**Expected Effectiveness:** 3 (Low Effectiveness: The idea might work in some special scenarios but you don't expect it to work in general.)
**Rationale:** Given that the idea is too similar to an existing one, the author may need to create a new but related idea as a follow-up study of the aforementioned paper. This idea does have a different motivation from the aforementioned one, so it uses different evaluation methods, though.

**Excitement:** 2
**Rationale:** Reviewers may argue the originality and novelty of this idea if it's submitted to a venue. They may not find it exciting, either.

**Overall Score:** 1 (Critically flawed, trivial, or wrong, would be a waste of students' time to work on it)
**Rationale:** The students should probably think one-step-further of the existing study and they may eventually find a way to improve the existing system.

**Confidence:** 5 (You are absolutely certain that the evaluation is correct and very familiar with the relevant literature)

### Reviewer 2

**Novelty:** 6 (reasonably novel - there are some notable differences from existing ideas and probably enough to turn into a new paper)
**Rationale:** Query decomposition and RAG separately are well studied, if there is no existing work that combines both (which I'm not aware of), then it's reasonably novel.

**Feasibility:** 10 (Easy: The whole proposed project can be quickly executed within a few days without requiring advanced technical skills.)
**Rationale:** It's just a series of prompting which should be easy for a CS PhD student.

**Expected Effectiveness:** 8 (Probably Effective: The idea should offer some significant improvement over current methods on the relevant benchmarks.)
**Rationale:** This method involves multiple fine-grained retrieval operations and should naturally outperform existing retrieval methods without decomposition.

**Excitement:** 6 (Learning positive: exciting enough to be accepted at a major AI conference, but still has some weaknesses or somewhat incremental)
**Rationale:** Although I believe in the effectiveness of the proposed method, the high latency compared to baselines is a concern—training an end-to-end model to reduce latency might be a good add-on.

**Overall Score:** 7 (Good idea, would be accepted by major AI conferences)
**Rationale:** This is a good idea. If there is no identical existing work and the authors conduct comprehensive experiments, it would be a good paper.

**Confidence:** 4 (You are confident but not absolutely certain that the evaluation is correct)

**Reviewer 3**

**Novelty:** 5 (somewhat novel - there are differences from existing ideas but not enough to turn into a new paper)
**Rationale:** The idea aims to tackle a question by breaking it down and solving it one by one with RAG. But it seems to be a more specialized way of CoT with RAG.

**Feasibility:** 5 (Moderately feasible: It can probably be executed within the given time frame but would require careful planning, efficient use of APIs or some advanced computational strategies to overcome the limited GPU resources, and would require some modifications to the original proposal to make it work.)
**Rationale:** The idea assumes a question can be broken down into subquestions where each subquestion is independent of the others. In cases where they are not independent, the method might suffer from issues or inefficiency. But maybe the distribution of these questions is more like a long tail and predominantly questions that can be easily broken down. And is there a case where the question is high-level mathematics and difficult to the point where it breaks down into a non-linear scale of the question text token?

**Expected Effectiveness:** 5 (Somewhat ineffective: There might be some chance that the proposed idea can work better than existing baselines but the improvement will be marginal or inconsistent.)
**Rationale:** The main question is how the sub-questions are created. We can break the question into conditioned parts from $p(q_0|q_0,...q_n)...p(q_n|q_0,...q_{n-1})$ where we assume them to be dependent, or we can use LLM to reason about their dependency. We can also ask the question by asking leveled sub-questions like "where is this person from" into "which country is this person from", "which city is this person from", "which district is this person from". The concern is that different methods might affect the performance differently.

**Excitement:** 6 (Learning positive: exciting enough to be accepted at a major AI conference, but still has some weaknesses or somewhat incremental)
**Rationale:** The idea seems exciting as it prevents LLM from shortcutting the question and hallucinating. But it needs more method formulation on how the question should be broken down. The very baseline implementation will just degrade to a CoT reasoning with RAG for each step. Because this could just be a subset of CoT methods in some sense.

**Overall Score:** 6 (Marginally above the acceptance threshold of major AI conferences)
**Rationale:** I believe there could be more comparison with CoT as motivation. Why should this be better with prompting the model step by step using RAG, and why are they different? And for problem formulation, it would be great if we can list more edgy examples of how questions can be divided to help pilot the prompting methods.

**Confidence:** 4 (You are confident but not absolutely certain that the evaluation is correct)

## A.25 EXAMPLE IDEA: SEMANTIC DIVERGENCE MINIMIZATION: REDUCING HALLUCINATIONS IN LARGE LANGUAGE MODELS THROUGH ITERATIVE CONCEPT GROUNDING

---

### Semantic Divergence Minimization: Reducing Hallucinations in Large Language Models through Iterative Concept Grounding (Part 1)

**1. Problem Statement:** Large language models often generate hallucinations by diverging from the core semantic content of the input, especially in complex reasoning tasks. This problem undermines the reliability and trustworthiness of LLMs in critical applications that require accurate and factual responses.

**2. Motivation:** Current approaches like chain-of-thought prompting focus on generating intermediate steps but do not explicitly constrain semantic drift. By continuously grounding generated content to the original semantic space of the input, we can reduce hallucinations while preserving reasoning capabilities. This method leverages the LLM's own ability to extract and compare semantic concepts, creating a self-correcting mechanism that does not require external knowledge bases or complex architectures.

**3. Proposed Method:** We introduce Semantic Divergence Minimization (SDM) prompting. For each reasoning step, we prompt the model to:

1. Generate a candidate next step.
2. Extract key semantic concepts from the original input.
3. Measure semantic similarity between the candidate step and extracted concepts.
4. If similarity is below a threshold, regenerate the step with explicit instructions to incorporate more relevant concepts.
5. Repeat until convergence or maximum iterations.

This creates a semantic 'gravity well' that keeps reasoning tethered to the input's conceptual core.

---

**Semantic Divergence Minimization: Reducing Hallucinations in Large Language Models through Iterative Concept Grounding (Part 2)**

**4. Step-by-Step Experiment Plan:**

1. **Dataset Preparation:**

   - Use two datasets: HotpotQA for multi-hop reasoning and GSM8K for complex math word problems.
   - For HotpotQA, utilize the dev set (7,405 questions).
   - For GSM8K, employ the test set (1,319 problems).

2. **Baseline Implementation:**

   - Implement two baselines:
     - Standard prompting: directly asking the model to answer the question.
     - Chain-of-thought (CoT) prompting: asking the model to show its work step-by-step before giving the final answer.

3. **SDM Implementation:**

   - Implement the SDM method with the following sub-steps for each reasoning iteration:
     - Generate next step.
     - Extract key concepts from input.
     - Measure semantic similarity.
     - Regenerate if below threshold.
     - Repeat until convergence or maximum iterations.

4. **Prompt Engineering:**

   - Design prompts for each step of SDM. For example:
     - "Generate the next step in solving this problem:"
     - "Extract key concepts from the original question:"
     - "Rate the semantic similarity between these concepts and the generated step on a scale of 0-10:"
     - "Regenerate the step, focusing more on these key concepts:"

5. **Hyperparameter Tuning:**

   - Experiment with different similarity thresholds (e.g., 5, 6, 7 out of 10) and maximum iteration limits (e.g., 3, 5, 7) to find the optimal balance between performance and computational cost.

6. **Model Selection:**

   - Use GPT-4 as the primary model due to its advanced reasoning capabilities.
   - Test GPT-3.5-turbo for comparison.

7. **Evaluation:**

   - For HotpotQA, use the official evaluation script to compute Exact Match (EM) and F1 scores.
   - For GSM8K, use accuracy as the metric.
   - Manually review a subset of responses to assess qualitative improvements in reasoning and reduction of hallucinations.

8. **Analysis:**

   - Compare SDM results with baselines.
   - Analyze the correlation between semantic similarity scores and answer correctness.
   - Examine cases where SDM significantly improves or fails to improve over baselines.

**Semantic Divergence Minimization: Reducing Hallucinations in Large Language Models through Iterative Concept Grounding (Part 3)**

**5. Test Case Examples:**

- **Example 1 (HotpotQA):** Input: Who was born first, Alec Guinness or Billie Holiday?
  Baseline Output: Alec Guinness was born first. He was born on April 2, 1914, while Billie Holiday was born on April 7, 1915.
  SDM Output: The SDM method provides a more structured and transparent reasoning process, reducing the risk of hallucination by grounding each step in the key concepts (birth dates) from the original question.

- **Example 2 (GSM8K):** Input: Janet's ducks lay 16 eggs per day. She eats three for breakfast every morning and sells the rest at the farmers' market daily for $2 per egg. How much money does she make every week?
  Baseline Output: Let's solve this step by step: Janet makes $182 every week.
  SDM Output: The SDM method not only provides a clear step-by-step solution but also explicitly verifies that all key concepts from the original question are incorporated, reducing the risk of introducing irrelevant information or hallucinating facts.

**6. Fallback Plan:** If the proposed SDM method does not significantly outperform baselines, we can pivot the project in several ways. First, we could conduct an in-depth analysis of where and why SDM fails, potentially uncovering insights about LLM reasoning processes. We might find that SDM works better for certain types of questions or reasoning tasks, which could lead to a more nuanced application of the method. Second, we could explore variations of SDM, such as using different prompts for concept extraction or similarity measurement, or incorporating a dynamic threshold that adjusts based on the complexity of the question. Third, we could combine SDM with other prompting techniques like chain-of-thought or self-consistency to create a hybrid approach. Finally, if the semantic grounding aspect proves challenging, we could shift focus to analyzing how LLMs interpret and maintain semantic consistency throughout multi-step reasoning, which could provide valuable insights for future work on reducing hallucinations.

## Reviewer 1

**Novelty:** 8 (clearly novel - major differences from all existing ideas)
**Rationale:** The use of semantic similarity to constrain CoT-styled generation is very new. I have not seen similar work on it.

**Feasibility:** 5 (Moderately feasible: It can probably be executed within the given time frame but would require careful planning, efficient use of APIs or some advanced computational strategies to overcome the limited GPU resources, and would require some modifications to the original proposal to make it work.)
**Rationale:** The pipeline is feasible to me. The major challenge would be finding the similarity threshold for each dataset.

**Expected Effectiveness:** 3 (Low Effectiveness: The idea might work in some special scenarios but you don't expect it to work in general.)
**Rationale:** I see some drawbacks in this pipeline. First, manually tuning the similarity threshold seems not the best practice for scalable applications. The GSM8K math dataset contains pretty elementary math problems. In that case, the semantic similarity threshold should be set very high, since these basic math concepts involved in the prompt and the CoT breakdown would be determined as highly similar by most existing embedding methods. This brings the question of whether this similarity threshold is non-trivial at all for some tasks.

**Excitement:** 6 (Learning positive: exciting enough to be accepted at a major AI conference, but still has some weaknesses or somewhat incremental)
**Rationale:** Constraining CoT breakdowns is a novel idea and deserves more work and exploration. While the use of semantic similarity has many drawbacks (such as tuning the threshold, task-sensitive, non-scalable), it can still show us some valuable results about constraining CoT breakdowns.

**Overall Score:** 5 (Decent idea but has some weaknesses or not exciting enough, marginally below the acceptance threshold of major AI conferences)
**Rationale:** There are some clear drawbacks inherent to the method, as discussed earlier. If the authors can overcome these limitations, this idea could yield some interesting findings useful for our understanding of CoT behavior and could pass above a major conference threshold.

**Confidence:** 3 (You are fairly confident that the evaluation is correct)

## Reviewer 2

**Novelty:** 4
**Rationale:** Generally this method is a way of rejection sampling to improve factuality. It is somewhat not too different from previous literature for "constrained decoding" for improving factuality: - Constrained Abstractive Summarization: Preserving Factual Consistency with Constrained Generation - Don't Say What You Don't Know: Improving the Consistency of Abstractive Summarization by Constraining Beam Search

**Feasibility:** 9
**Rationale:** Simple prompting approach that is easy to implement. Evaluation is simple.

**Expected Effectiveness:** 3 (Low Effectiveness: The idea might work in some special scenarios but you don't expect it to work in general.)
**Rationale:** 1. Right now most LLMs hallucinate in a subtle way: they say things in semantically correct or reasonable ways, but the precise fact is incorrect. Using semantic similarity as a measurement to gauge/control hallucination might not be able to solve the problem. 2. The rejection sampling is based on another LLM—what if the LLM also hallucinates?

**Excitement:** 3 (Mediocre: this idea makes marginal contributions and is very incremental)
**Rationale:** The method is not that novel and I think the method is not that effective and might not solve the problem at all.

**Overall Score:** 3 (Clear rejection for major AI conferences)
**Rationale:** The experiment design is kind of simple and the evaluation is not comprehensive. I think the idea is in the range of 4 but the experiment plan further reduces my score.

**Confidence:** 5 (You are absolutely certain that the evaluation is correct and very familiar with the relevant literature)

## Reviewer 3

**Novelty:** 3 (mostly not novel - you can find very similar ideas)
**Rationale:** The idea of extracting key semantic concepts, measuring the relevance of the candidate next step, and possibly rejecting/revising the step is very similar to incorporating self-critique into multi-step reasoning problems. Different versions of this are already commonly used, especially for solving math problems.

**Feasibility:** 8 (Highly Feasible: Straightforward to implement the idea and run all the experiments.)
**Rationale:** The proposed approach should be straightforward to implement: it only requires prompt engineering to extract semantic concepts and evaluate the relevance of a candidate next step.

**Expected Effectiveness:** 3 (Low Effectiveness: The idea might work in some special scenarios but you don't expect it to work in general.)
**Rationale:** Compared to chain-of-thought prompting, there's a reasonable chance this method could work better: it could help identify when a reasoning step becomes irrelevant to the original question. However, since such self-critique methods have already been explored, it's unlikely that this instantiation will work significantly better than previous ones. Also, the proposed idea of extracting relevant semantic concepts and measuring semantic similarity seems a bit vague, and it's not reflected in the provided examples.

**Excitement:** 2
**Rationale:** The proposed method is too similar to existing works; it doesn't contain novel insights that would meaningfully boost current LM performance or introduce new ideas worth building on. It would not be an exciting paper.

**Overall Score:** 2 (Strong rejection for major AI conferences)
**Rationale:** Similar to the reasoning above: the proposal is too similar to existing works, it doesn't introduce new ideas or insights, and is unlikely to meaningfully improve current LM performance.

**Confidence:** 4 (You are confident but not absolutely certain that the evaluation is correct)

## A.26 EXAMPLE IDEA: AUTOPROMPTING: GENERATE DIVERSE FEW-SHOT EXAMPLES FOR ANY APPLICATION

---

**Autoprompting: Generate Diverse Few-Shot Examples for Any Application (Part 1)**

**1. Problem Statement:** Adding natural language capabilities to existing software requires manually crafting few-shot prompts, which is tedious and does not guarantee high coverage.

**2. Motivation:** Integrating natural language capabilities into software applications often necessitates manually creating few-shot prompts, a process that is time-consuming and may not ensure comprehensive coverage. An "Autoprompting" system capable of automatically generating diverse and relevant few-shot examples tailored to specific applications would significantly reduce manual effort, improve coverage and versatility, and enable rapid prototyping and iteration of natural language capabilities. Large Language Models can iteratively test different functionalities of an application and make adjustments to few-shot prompts akin to a human developer. This approach would ultimately democratize the integration of such capabilities across a wide range of applications and industries.

**3. Proposed Method:** This method leverages a Large Language Model (LLM) with coding capabilities. It involves the following core steps:

1. Extract all user-facing functions and gather their documentation and unit tests, if available.
2. Generate diverse natural language prompts to utilize each function, defining the expected output.
3. Generate code from the natural language prompts and execute the corresponding functions.
4. If the code fails:
   - Update the code and retry
   - If the code runs but produces an incorrect result, update it using insights from unit tests or general reasoning.
5. Once you have a few exemplar prompts for all (or desired) functions, generate prompts that compose multiple functions together and repeat step 4.

By iteratively refining code generation from natural language and leveraging available documentation and tests, this process aims to create an LLM capable of correctly implementing functions based on natural language instructions.

**4. Step-by-Step Experiment Plan:**

- **Applications:** When collecting applications from GitHub, prioritize those with clear, well-written documentation and comprehensive test suites. Include applications from different domains and with varying levels of complexity to ensure a diverse dataset.

- **Few shots and feasibility:** Create manual few-shot examples to understand the complexity of the functions and the quality of the documentation. Begin by creating 4-5 examples for any function, which could also serve as a starting point for the LLM.

- **Extract functions and metadata:** Utilize static code analysis tools to ensure accurate and comprehensive extraction of functions, documentation, and test cases. Consider extracting additional metadata, such as function signatures, dependencies, and comments, as they can provide valuable context.

- **NL Module:** Generate diverse user utterances and incorporate techniques to handle variations in natural language. For each user utterance, generate the expected outcome. Consider generating negative test cases to improve the model's ability to handle invalid or ambiguous inputs.

- **Execution Module:** Incorporate sandboxing or containerization techniques to ensure a secure and isolated execution environment when executing the generated code. Implement logging and reporting mechanisms to capture and analyze errors and unexpected behavior.

---

---

**Autoprompting: Generate Diverse Few-Shot Examples for Any Application (Part 2)**

**4. Step-by-Step Experiment Plan (Continued):**

- **Exploration:** Incorporate techniques such as code summarization, call graph analysis, and type inference to provide more contextual information to the agent. Specifically, in any code snippet, if there are other user-defined functions, retrieve their metadata and use it in the next iteration of prompt generation.

- **Store:** Utilize a vector database or other structured storage mechanism that supports efficient retrieval and querying for storing few-shot examples and their outputs. Incorporate mechanisms for versioning and updating the stored data as the codebase and the underlying models evolve.

- **Experiments:** Once few-shot examples for different functionalities and their compositions are obtained, simulate different users with various intents and calculate goal completion and error rates using different models. Initially, start with a strong model, and once few-shot examples are available, test with weaker and open-source models.

**5. Test Case Examples:** Select a toy application from GitHub implemented in Python or JavaScript.

- **Direct prompting:** Provide the few-shot examples created and check the goal completion and error rates for the following scenarios.

- **Toy example:** Calculator app and different utterances to try.

  - Provide a complete user utterance with no ambiguity. For example:
    * Can you add 4 to 8.
    * Divide 6 by 9 and multiply it by 6.

  - Provide a user utterance with some ambiguity. For example:
    * Take 6 and 9, add them, and then subtract 8. Also, add 2 to the first one. – here the "first" one is ambiguous as it could be 6 or the intermediate answer (6+9=15).

  - Provide a user utterance that is not related to the function. For example:
    * Please add A and J. The correct result would be refusing to answer instead of generating add("A", "J").

**6. Fallback Plan:** If the proposed methodology does not yield satisfactory results, there are several areas to investigate. First, examine the documentation to ensure it adequately explains the basic functionality of each function. Then, assess the coding style to confirm it aligns with recommended practices. Subsequently, evaluate each module separately. For the NL module, verify that the examples are diverse and that the generated test cases are aligned. For the execution module, ensure that the correct error messages are being passed and explore ways to enhance them. The exploration module is the most challenging aspect; if any function has a high dependency on other functions, traversing it will be difficult. Therefore, initially focus on examples with limited to no function dependency and gradually increase the complexity.

**Reviewer 1**

**Novelty:** 4
**Rationale:** The proposed method is similar to https://arxiv.org/abs/2210.03493; https://aclanthology.org/2023.findings-acl.216/

**Feasibility:** 6 (Feasible: Can be executed within the given constraints with some reasonable planning.)
**Rationale:** The experiments can be done with sufficient API access. The dataset collection needs some planning but is in general feasible to do. Setting up the vector database may take extra time.

**Expected Effectiveness:** 5 (Somewhat ineffective: There might be some chance that the proposed idea can work better than existing baselines but the improvement will be marginal or inconsistent.)
**Rationale:** The proposal is vague as it doesn't mention what's the final evaluation metric, and does not provide sufficient description of the compared baseline. The prompt in the direct prompt baseline is confusing to me as well. Overall it's hard to discuss the effectiveness.

**Excitement:** 4
**Rationale:** Given that the proposed method is vague, I am unsure about its contributions and effectiveness, and therefore I feel less excited about it.

**Overall Score:** 4 (Ok but not good enough, rejection for major AI conferences)
**Rationale:** The descriptions are confusing and I'm not really sure what's the focus or contribution. The title problem statement mentioned ensuring "diversity"/"high coverage" as the goal but doesn't describe how this is ensured in later subsections. The "Test Case Examples" doesn't explain how the components in the "Step-by-Step Experiment Plan" are used.

**Confidence:** 3 (You are fairly confident that the evaluation is correct)

**Reviewer 2**

**Novelty:** 7
**Rationale:** Mapping natural language to custom applications is a hugely impactful capability, and doing so automatically is really interesting. I like the focus on autoprompting for these types of translations, as the task is feasible since it builds off some of the "few-shot prompting" that developers might normally do to add NL functionality, with a more automatic process that has real system checks/verifications (e.g., running the applications through containers). A related work from HCI tries to enable individual developers to add such NL functionality to their own applications via a DSL + NL program signatures (https://jackieyang.me/reactgenie/). This work is distinguished, as it would empower adding such NL functionality to any application, without changing the code.

**Feasibility:** 4
**Rationale:** The project infrastructure seems more difficult than simply choosing some prompting methods. It would be an iterative process choosing real example applications from Github, and developing the few-shot prompts manually to get a feel for this task. Then, some of the modules seem like 1-2 week tasks (Execution Module, Exploration, Storage) which I estimate would make the project more like 3 - 4 months to complete all modules AND to do the evaluations.

**Expected Effectiveness:** 7
**Rationale:** The baseline here is a zero-shot prompt, asking to do the NL intent and feeding in all the documentation of the API. Assuming the author is correct to say that such NL function mapping requires good few & diverse few-shot examples, I expect the method to work well. It uses a number of external systems to enrich the code dataset to give the LLM context and uses system errors to inform. So in some ways, Autoprompting is allowing an agent to make use of all these SWE tools for understanding the software, which then will allow it to maximize its understanding and better retrieve good few-shot examples for the task at hand.

**Excitement:** 7
**Rationale:** Seems like an impactful and ambitious outcome if completed. I am curious how such an approach fits into the conversation about general agents, which can leverage API/tool/functions calls. It's a little unclear from the toy example why existing function-calling models can't translate NL intents into.

**Overall Score:** 6 (Marginally above the acceptance threshold of major AI conferences)
**Rationale:** The results would be really exciting and the technical infrastructure to enable the Auto-prompting agent would be impressive. However, I'm missing a bit of which cases will be really difficult for other generalist web/system agents, but where finding the few-shot examples for this task is really needed. Thus, the core idea of the method doesn't seem clarified enough to result in a really clear takeaway on the method.

**Confidence:** 3 (You are fairly confident that the evaluation is correct)

## A.27 EXAMPLE IDEA: TEMPORAL DEPENDENCY UNFOLDING: IMPROVING CODE GENERATION FOR COMPLEX STATEFUL SYSTEMS

---

### Temporal Dependency Unfolding: Improving Code Generation for Complex Stateful Systems (Part 1)

**1. Problem Statement:** Generating code for complex, stateful systems or applications with intricate temporal dependencies remains challenging for current code generation models. Most existing approaches focus on generating individual functions or small code snippets without fully considering the temporal aspects and state changes in larger systems. This limitation hinders the applicability of AI-assisted programming in areas such as distributed systems, game development, and real-time applications.

**2. Motivation:** Many real-world applications require careful management of state over time. Existing code generation models struggle with capturing the full complexity of temporal dependencies and state changes in larger systems. A method that can effectively reason about and generate code for systems with complex temporal dependencies could significantly improve the applicability of AI-assisted programming in critical areas. Our proposed Temporal Dependency Unfolding method is inspired by how human developers approach complex system design, first identifying key states and their relationships before implementing the detailed logic.

**3. Proposed Method:** We propose Temporal Dependency Unfolding, a novel prompting technique that guides the model to generate code by explicitly reasoning about state changes and temporal relationships. The method consists of five steps:

1. State Identification: Prompt the model to identify key states and variables that change over time in the target system.
2. Temporal Graph Construction: Guide the model to create a conceptual graph of how these states evolve and interact over time.
3. Staged Code Generation: Generate code in stages, focusing on different temporal slices or state transitions in each stage.
4. Consistency Verification: After each stage, prompt the model to verify temporal consistency and make necessary adjustments.
5. Integration: Finally, guide the model to integrate the stage-wise generated code into a cohesive system, ensuring proper handling of all temporal dependencies.

**4. Step-by-Step Experiment Plan:**

1. **Dataset Preparation:**
   - Create a dataset of programming tasks that involve complex temporal dependencies.
   - Include tasks from three domains: 1) Multi-threaded applications, 2) Game logic, and 3) Distributed systems.
   - For each domain, prepare 50 task descriptions, each with a clear specification of the desired functionality and temporal requirements.

2. **Baseline Implementation:**
   - Implement two baseline methods:
     - Direct prompting: Simply provide the task description to the model and ask it to generate the code.
     - Chain-of-Thought (CoT) prompting: Append 'Let's approach this step-by-step:' to the task description.
   - Use GPT-4 for both baselines.

**Temporal Dependency Unfolding: Improving Code Generation for Complex Stateful Systems (Part 2)**

**4. Step-by-Step Experiment Plan (Continued):**

3. **Temporal Dependency Unfolding Implementation:**

- Implement our proposed method with the following sub-steps for each task:
  (a) State Identification: Prompt GPT-4 with 'Identify the key states and variables that change over time in this system:'.
  (b) Temporal Graph Construction: Prompt with 'Create a conceptual graph showing how the identified states evolve and interact over time:'.
  (c) Staged Code Generation: For each major state or transition identified, prompt with 'Generate code for the following state/transition: [state/transition]'.
  (d) Consistency Verification: After each stage, prompt with 'Verify the temporal consistency of the generated code and suggest any necessary adjustments:'.
  (e) Integration: Finally, prompt with 'Integrate the generated code segments into a cohesive system, ensuring proper handling of all temporal dependencies:'.

4. **Evaluation Metrics:**

- Correctness: Percentage of generated code that passes predefined test cases.
- Temporal Consistency: Manual evaluation of how well the code handles temporal dependencies (scale 1-5).
- Code Quality: Automated metrics like cyclomatic complexity and maintainability index.
- Execution Efficiency: Runtime performance on benchmark inputs.

5. **Human Evaluation:**

- Recruit 5 experienced developers to review a subset of 30 generated solutions (10 from each domain).
- They will rate the code on a scale of 1-5 for readability, maintainability, and correct handling of temporal dependencies.

6. **Experiment Execution:**

- For each task in the dataset:
  (a) Generate solutions using both baseline methods and our Temporal Dependency Unfolding method.
  (b) Apply all evaluation metrics to the generated solutions.
  (c) Collect human evaluations for the subset of solutions.

7. **Analysis:**

(a) Compare the performance of Temporal Dependency Unfolding against the baselines across all metrics.
(b) Analyze the effectiveness of each step in our method (State Identification, Temporal Graph Construction, etc.) by examining intermediate outputs.
(c) Identify patterns in tasks where our method shows significant improvement or underperforms.
(d) Correlate automated metrics with human evaluations to validate their reliability.

**Temporal Dependency Unfolding: Improving Code Generation for Complex Stateful Systems (Part 3)**

**5. Test Case Examples:**

- **Test Case 1:**

  - Baseline Prompt Input (Direct Prompting): Generate Python code for a simple multi-threaded producer-consumer system with a shared buffer. The producer should generate random numbers and add them to the buffer, while the consumer should remove and process these numbers. Implement proper synchronization to avoid race conditions.

  - Baseline Prompt Expected Output (Direct Prompting): [Python code for a simple producer-consumer system]

  - Proposed Prompt Input (Temporal Dependency Unfolding; Step 1: State Identification): For a multi-threaded producer-consumer system with a shared buffer, identify the key states and variables that change over time in this system:

  - Proposed Prompt Expected Output (Temporal Dependency Unfolding; Step 1: State Identification): [List of key states and variables]

  - Proposed Prompt Input (Temporal Dependency Unfolding; Step 2: Temporal Graph Construction): Create a conceptual graph showing how the identified states evolve and interact over time for the producer-consumer system:

  - Proposed Prompt Output (Temporal Dependency Unfolding; Step 2: Temporal Graph Construction): [Conceptual graph of state evolution and interactions]

  - Proposed Prompt Input (Temporal Dependency Unfolding; Step 3: Staged Code Generation): Generate code for the producer functionality in the producer-consumer system, focusing on its interaction with the buffer and synchronization mechanisms:

  - Proposed Prompt Output (Temporal Dependency Unfolding; Step 3: Staged Code Generation): [Python code for producer functionality]

  - Proposed Prompt Input (Temporal Dependency Unfolding; Step 4: Consistency Verification): Verify the temporal consistency of the generated producer code and suggest any necessary adjustments:

  - Proposed Prompt Output (Temporal Dependency Unfolding; Step 4: Consistency Verification): [Verification and adjustment suggestions]

  - Proposed Prompt Input (Temporal Dependency Unfolding; Step 5: Integration): Integrate the generated producer code with a consumer and main control logic to create a complete producer-consumer system, ensuring proper handling of all temporal dependencies:

  - Proposed Prompt Output (Temporal Dependency Unfolding; Step 5: Integration): [Complete Python code for producer-consumer system]

  - **Explanation:** The Temporal Dependency Unfolding method produces a more comprehensive and robust solution compared to the baseline. It explicitly handles temporal dependencies, includes proper synchronization, and provides mechanisms for graceful termination. The staged approach allows for better handling of edge cases and improved overall system design.

**6. Fallback Plan:** If the Temporal Dependency Unfolding method does not show significant improvement over the baselines, we can pivot the project in several ways. First, we could conduct an in-depth analysis of where and why the method fails, which could provide valuable insights into the limitations of current language models in handling temporal reasoning tasks. This analysis could involve examining the intermediate outputs (state identification, temporal graphs) to understand where the reasoning breaks down. Second, we could explore combining our method with other techniques, such as retrieval-augmented generation, to see if providing relevant examples improves performance. Third, we could focus on developing a new evaluation framework specifically designed to assess temporal reasoning in code generation, which could be a valuable contribution to the field even if our primary method doesn't outperform baselines. Lastly, we could investigate whether the method performs better on certain types of temporal dependencies or specific programming domains, which could lead to a more targeted approach for improving code generation in those areas.

**Reviewer 1**

**Novelty:** 6 (reasonably novel - there are some notable differences from existing ideas and probably enough to turn into a new paper)
**Rationale:** The construction of Temporal Graph sounds novel. The research question is also relatively underexplored, but necessary for coding in domains like distributed systems.

**Feasibility:** 6 (Feasible: Can be executed within the given constraints with some reasonable planning.)
**Rationale:** The data collection part should be the most challenging part. Collecting high-quality coding problems that involve complex temporal dependencies could be hard. Also, the human evaluation might also take time to execute.

**Expected Effectiveness:** 6 (Somewhat effective: There is a decent chance that the proposed idea can beat existing baselines by moderate margins on a few benchmarks.)
**Rationale:** With specific prompting techniques, the proposed method should outperform baselines in terms of temporal dependencies.

**Excitement:** 7
**Rationale:** I think this should be more exciting than most of the borderline papers since we are working on a new problem. The collected data should also be super useful.

**Overall Score:** 7 (Good idea, would be accepted by major AI conferences)
**Rationale:** Again, working on a novel problem makes it better than most of the prompting papers.

**Confidence:** 4 (You are confident but not absolutely certain that the evaluation is correct)

## Reviewer 2

**Novelty:** 5 (somewhat novel - there are differences from existing ideas but not enough to turn into a new paper)

**Rationale:** Although I am not entirely familiar with the field of generating temporally adaptive programs, I suspect some similar ideas can be found in software engineering works (e.g., ICSE). More concretely on the method, it is rather similar to code generation with intermediate state reasoning, which has been explored in several multi-step, conversational code generation works, e.g:

1. Zheng, Tianyu, et al. "Opencodeinterpreter: Integrating code generation with execution and refinement."

2. Cao, Liuwen, et al. "Beyond Code: Evaluate Thought Steps for Complex Code Generation." Proceedings of the 2024 Joint International Conference on Computational Linguistics, Language Resources and Evaluation (LREC-COLING 2024). 2024.

3. Nijkamp, Erik, et al. "Codegen: An open large language model for code with multi-turn program synthesis."

**Feasibility:** 3 (Very challenging: there are flaws in the proposed method or experiments, or the experiments require compute/human resources beyond any academic lab)

**Rationale:** It would be pretty hard to collect such datasets (e.g., would mostly require a whole repository), further, it would be difficult to generate executable test cases to verify the multiple problems created. Especially because the task targets temporally-dependent modules in the program, it may necessitate domain experts to carefully construct examples and tests, which would demand a lot of time and costs.

**Expected Effectiveness:** 5 (Somewhat ineffective: There might be some chance that the proposed idea can work better than existing baselines but the improvement will be marginal or inconsistent.)

**Rationale:** I am not very confident that the model can solve this complex temporally-dependent programming problems with reasonable correctness. Furthermore, because the current method is basically prompting, which may have a very low performance upper bound. Therefore, I don't expect the proposed method to improve significantly on code generation.

**Excitement:** 4

**Rationale:** Overall, I don't expect this method to bring substantial improvements, hence am less excited about the potential of this method. It would still be an interesting problem to solve, particularly in bringing more challenging coding problems and proposed corresponding methods. With this being said, given the current performance of models, building a solid benchmark regarding this temporal code generation problem may be more exciting than proposing a method that is expectedly not working.

**Overall Score:** 4 (Ok but not good enough, rejection for major AI conferences)

**Rationale:** The task of temporal code generation is not the most urgent issue of current code generation models, and the proposed method is expected to not bring much improvement. The method needs to be further refined and go beyond simple prompting to convince the audience of the potential of this thread of methods.

**Confidence:** 3 (You are fairly confident that the evaluation is correct)

**Reviewer 3**

**Novelty:** 10 (very novel - very different from all existing ideas in a very interesting and clever way)
**Rationale:** This idea studies a very novel problem in LLM-based code generation. Temporal dependencies in code generation should be specifically studied in the era of LLMs.

**Feasibility:** 5 (Moderately feasible: It can probably be executed within the given time frame but would require careful planning, efficient use of APIs or some advanced computational strategies to overcome the limited GPU resources, and would require some modifications to the original proposal to make it work.)
**Rationale:** Constructing a reasonable dataset is challenging within a short time. Also, human evaluation might take more time. Whether LLM can construct high-quality graphs in this case is also to be examined.

**Expected Effectiveness:** 6 (Somewhat effective: There is a decent chance that the proposed idea can beat existing baselines by moderate margins on a few benchmarks.)
**Rationale:** One needs to build reasonable metrics to show effectiveness. Also, one might need to tune prompts carefully to construct high-quality graphs in this case.

**Excitement:** 8 (Exciting: would deepen the community's understanding or make major progress in this research direction)
**Rationale:** This is novel and could have a huge impact on those code generation cases requiring temporal dependencies. But one needs to justify why such use cases are important, and why temporal dependency is the core problem in such use cases.

**Overall Score:** 9 (Top 15% of all published ideas on this topic at major AI conferences, strong accept)
**Rationale:** Considering its novelty, valuable dataset, and comprehensiveness of experiment and evaluation design, this could be an impactful work. But one needs to make experiment results concrete by re-examining whether each step works well in practice.

**Confidence:** 4 (You are confident but not absolutely certain that the evaluation is correct)

## A.28 IDENTITIES OF EXAMPLE IDEAS

We reveal whether each example idea is AI-generated or human-written:

- Human ideas: Example A.20, Example A.22, Example A.24, Example A.26
- AI ideas: Example A.21, Example A.23, Example A.25, Example A.27

A.29 ABLATIONS OF IDEA GENERATION

In this section, we investigate several factors that influence the performance of our retrieval-augmented generation (RAG) approach for idea generation. We ablate:

1. The maximum number of papers to retrieve ($N$).

2. The number of papers ($k$) to include in the prompt.

3. Different base models and prompt setups for idea generation.

For ablating the impact of the maximum number of papers to retrieve ($N$), we vary $N$ among $\{30, 60, 120, 180\}$ and measure the average relevance score of the top 20 retrieved papers (on a scale of 1–10), as judged by an LLM. The results in Table 17 show that $N = 120$ consistently achieves strong performance and that the relevance of the top retrieved papers tends to plateau beyond this point.

Table 17: Ablation of the maximum number of papers to retrieve ($N$). We report the average relevance score of the top 20 retrieved papers (on a scale of 1–10) for two topics (Multilingual and Uncertainty) as judged by an LLM.

| N | Multilingual | Uncertainty |
|---|---|---|
| 30 | 6.80 | 6.75 |
| 60 | 7.05 | 7.50 |
| 120 | 8.10 | 8.45 |
| 180 | 8.10 | 8.40 |

We additionally ablate the impact of the number of papers ($k$) to add to the prompt for RAG by varying $k$ in $\{0, 5, 10, 20\}$, and measure how it affects the diversity of generated ideas for the topic of uncertainty prompting. In Table 18, the "Non-Duplicates (%)" metric reflects the proportion of unique ideas generated. We observe that varying $k$ has minimal impact on the diversity.

Table 18: Ablation of the number of retrieved papers $k$ added to the prompt, showing the diversity of generated ideas on the uncertainty prompting topic. "Non-Duplicates (%)" indicates the proportion of unique ideas.

| $k$ | Non-Duplicates (%) |
|---|---|
| 0 | 18.8 |
| 5 | 18.4 |
| 10 | 19.1 |
| 20 | 19.4 |

To examine whether the diversity issue exists across different base models and prompt configurations, we measure the idea diversity of multiple other models and prompt setups. We use a temperature of 1.0 for all the generations and compute the percentage of non-duplicate ideas out of 2K generated ideas on the topic of uncertainty prompting. Table 19 compares four different base models.

Table 19: Comparison of different base models regarding idea diversity on the topic of uncertainty prompting. We measure the percentage of non-duplicate ideas out of 2K generated ideas using a temperature of 1.0.

| Base Model | Non-Duplicates (%) |
|---|---|
| Claude-3.5-Sonnet | 19.1 |
| GPT-4o | 59.5 |
| o1-mini | 22.6 |
| Llama-3.1-405B-Instruct | 51.1 |

We find that different models have very different non-duplicate rates, with GPT-4o and Llama-3.1-405B-Instruct showing substantially higher diversity than Claude-3.5-Sonnet and o1-mini. However, we chose Claude-3.5-Sonnet as the base model of our agent because the *quality* of its generated ideas outperforms the others in our pilot study. Specifically, we randomly sampled 10 ideas from Claude-3.5-Sonnet and GPT-4o for a round of pilot expert scoring and found that Claude-3.5-Sonnet achieved an average score of 5.4 (on a 1–10 scale), whereas GPT-4o scored 4.8.

Next, we ablate several different prompt setups for idea generation using the Claude-3.5-Sonnet backbone. Specifically, we analyze the impact of applying RAG and the impact of appending previously generated ideas for deduplication.

Table 20 shows that appending previously generated ideas in the prompt and instructing the model to avoid repetition significantly reduces idea duplication. However, including retrieved papers in the prompt has minimal impact on diversity.

Table 20: Comparison of different prompt setups for idea generation with the Claude-3.5-Sonnet backbone. We report the percentage of non-duplicate ideas out of 2K generated ideas on the topic of uncertainty prompting.

| Prompt Setup | Non-Duplicates (%) |
|---|---|
| No RAG; No prev | 7.6 |
| No RAG; Prev | 18.8 |
| RAG (k=5); Prev | 18.4 |
| RAG (k=10); Prev | 19.1 |
| RAG (k=20); Prev | 19.4 |

## A.30 ATTEMPT ON IDEA EXECUTION AGENT

For our execution agent, the input is the generate idea (the full project proposal), and the output is a Python file that can be executed with our specified command. Since there is often a common pipeline of implementing prompting-based research ideas, we provide a manually crafted code file example as template. We attach the full template below:

```python
import random
from tqdm import tqdm
from utils import call_api, load_model
import random
random.seed(2024)

## Step 1: Generate synthetic test examples
def generate_testset():
    test_data = [
        {
            "input": "Natalia sold clips to 48 of her friends in
                April, and then she sold half as many clips in May.
                How many clips did Natalia sell altogether in April
                and May?",
            "output": "Natalia sold 48/2 = <<48/2=24>>24 clips in
                May. Natalia sold 48+24 = <<48+24=72>>72 clips
                altogether in April and May. #### 72"
        },
        {
            "input": "Weng earns $12 an hour for babysitting.
                Yesterday, she just did 50 minutes of babysitting.
                How much did she earn?",
            "output": "Weng earns 12/60 = $<<12/60=0.2>>0.2 per
                minute. Working 50 minutes, she earned 0.2 x 50 =
                $<<0.2*50=10>>10. #### 10"
        },
        {
            "input": "Tim has 30 less apples than Martha, and Harry
                has half as many apples as Tim. If Martha has 68
                apples, how many apples does Harry have?",
            "output": "Tim has 68-30 = <<68-30=38>>38 apples. Harry
                has 38/2 = <<38/2=19>>19 apples. #### 19"
        },
        {
            "input": "Four people lost a total of 103 kilograms of
                weight. The first person lost 27 kilograms. The
                second person lost 7 kilograms less than the first
                person. The two remaining people lost the same
                amount. How many kilograms did each of the last two
                people lose?",
            "output": "Second person = 27 - 7 = <<27-7=20>>20 kg 103
                - 27 - 20 = <<103-27-20=56>>56 kg 56/2 =
                <<56/2=28>>28 kg The last two people each lost 28
                kilograms of weight. #### 28"
        }
    ]

    return test_data

## Step 2: Implement the baseline method
def baseline_method(client, model_name, seed, question):
    ## zero-shot chain-of-thought
    prompt = "Answer the following question: {}\n".format(question)
    prompt += "Think step by step."
    prompt_messages = [{"role": "user", "content": prompt}]
```

```python
37      response, _ = call_api(client, model_name, prompt_messages,
            temperature=0., max_tokens=2000, seed=seed, json_output=False)
38      return response.strip()
39
40
41  ## Step 3: Implement the proposed method
42  def proposed_method(client, model_name, seed, question,
        print_all=False):
43      intermediate_outputs = ""
44
45      if print_all:
46          print ("question:\n", question)
47
48      ## collaborative reasoning step 1: task decomposition
49      prompt = "Please break down the following task into smaller
            sub-tasks or steps:: {}".format(question)
50      prompt_messages = [{"role": "user", "content": prompt}]
51      decomposition, _ = call_api(client, model_name, prompt_messages,
            temperature=0., max_tokens=2000, seed=seed, json_output=False)
52      intermediate_outputs += "task decomposition:\n" + decomposition +
            "\n"
53      if print_all:
54          print ("decomposition:\n", decomposition)
55
56      ## collaborative reasoning step 2: sub-task information generation
57      prompt = "For each of the following sub-tasks, please generate
            relevant information or intermediate results:
            \n{}".format(decomposition)
58      prompt_messages = [{"role": "user", "content": prompt}]
59      intermediate, _ = call_api(client, model_name, prompt_messages,
            temperature=0., max_tokens=2000, seed=seed, json_output=False)
60      intermediate_outputs += "sub-task results:\n" + intermediate +
            "\n"
61      if print_all:
62          print ("intermediate:\n", intermediate)
63
64      ## collaborative reasoning step 3: result combination
65      prompt = "Given the following intermediate results: \n{}, please
            combine them to generate the final answer for the task:
            \n{}".format(intermediate, question)
66      prompt_messages = [{"role": "user", "content": prompt}]
67      answer, _ = call_api(client, model_name, prompt_messages,
            temperature=0., max_tokens=2000, seed=seed, json_output=False)
68      intermediate_outputs += "result combination:\n" + answer + "\n"
69      if print_all:
70          print ("initial answer:\n", answer)
71
72      ## collaborative reasoning step 4: reflection and refinement
73      prompt = "Given the task: {}\nPlease reflect on the generated
            answer:\n{}.\n\nAre there any gaps or inconsistencies in the
            answer? If so, please identify and address them and give me
            an improved answer. If not, you don't have to edit anything
            and can just return the original answer.\n".format(question,
            answer)
74      prompt_messages = [{"role": "user", "content": prompt}]
75      final_answer, _ = call_api(client, model_name, prompt_messages,
            temperature=0., max_tokens=2000, seed=seed, json_output=False)
76      intermediate_outputs += "reflection and refinement:\n" +
            final_answer
77      if print_all:
78          print ("final answer:\n", final_answer)
79
80      return final_answer.strip(), intermediate_outputs
81
82
```

```python
83  ## Step 4: Define the style evaluator
84  def style_evaluator(client, model_name, seed, question,
        baseline_prediction, proposed_prediction):
85      ## define all the components that the proposed method outputs
            should have
86      ## and the advantages of the proposed method over the baseline
            method
87      ## just need to check the style is correct
88      prompt = "Given the task: {}\n".format(question)
89      prompt += "The baseline method produced the following
            output:\n{}\n\n".format(baseline_prediction)
90      prompt += "The proposed new method produced the following
            output:\n{}\n\n".format(proposed_prediction)
91      prompt += "Now determine if the proposed method is better by
            checking if it has satisfied the following criteria:\n"
92      prompt += "1. The proposed method's output should produce all the
            intermediate components including: task decomposition,
            sub-task information generation, result combination, and
            reflection and refinement.\n"
93      prompt += "2. The proposed method should provide a more detailed
            and comprehensive answer than the baseline method.\n"
94      prompt += "Just tell me 'yes' or 'no' for whether the criteria
            are met, nothing else is needed."
95      prompt_messages = [{"role": "user", "content": prompt}]
96      response, _ = call_api(client, model_name, prompt_messages,
            temperature=0., max_tokens=1, seed=seed, json_output=False)
97
98      judgment = False
99      if response.strip().lower() == "yes":
100         return True
101
102     return judgment
103
104
105 ## Step 5: Define the output evaluator
106 def output_evaluator(client, model_name, seed, question, gold_label,
        prediction):
107     ## check if the prediction is correct given the gold label
108     prompt = "Given the following question and reference answer,
            determine if the prediction is correct. Just tell me 'yes' or
            'no', nothing else is needed.\n\nQuestion: {}\n\nReference
            Answer: {}\n\nPrediction: {}\n\n".format(question,
            gold_label, prediction)
109     prompt_messages = [{"role": "user", "content": prompt}]
110     response, _ = call_api(client, model_name, prompt_messages,
            temperature=0., max_tokens=1, seed=seed, json_output=False)
111
112     judgment = False
113     if response.strip().lower() == "yes":
114         return True
115
116     return judgment
117
118
119 ## Step 6: Define the function that runs the experiments to obtain
        model predictions and performance
120 ## you shouldn't need to modify this function in most cases
121 def run_experiment(client, model_name, seed, testset):
122     sample_size = len(testset)
123     baseline_predictions = []
124     proposed_predictions = []
125
126     baseline_correctness = []
127     proposed_correctness = []
128
```

```
129     style_check = []
130
131     for i in tqdm(range(sample_size)):
132         question = testset[i]["input"].strip()
133         gold_label = testset[i]["output"].strip()
134
135         baseline_prediction = baseline_method(client, model_name,
                seed, question)
136         proposed_prediction_final, proposed_prediction_intermediate =
                proposed_method(client, model_name, seed, question)
137         baseline_predictions.append(baseline_prediction)
138         proposed_predictions.append(proposed_prediction_final)
139
140         baseline_correctness.append(output_evaluator(client,
                model_name, seed, question, gold_label,
                baseline_prediction))
141         proposed_correctness.append(output_evaluator(client,
                model_name, seed, question, gold_label,
                proposed_prediction_final))
142
143         style_check.append(style_evaluator(client, model_name, seed,
                question, baseline_prediction,
                proposed_prediction_intermediate))
144
145     return baseline_correctness, proposed_correctness, style_check
146
147
148 ## Step 7: Execute the experiments and compare performance
149 if __name__ == "__main__":
150     testset = generate_testset()
151     print ("simulated {} test examples for
            evaluation.".format(len(testset)))
152
153     model_name = "claude-3-opus-20240229"
154     seed = 2024
155     client = load_model(model_name)
156     print ("using model: ", model_name)
157
158     ## output correctness
159     baseline_correctness, proposed_correctness, style_check =
            run_experiment(client, model_name, seed, testset)
160     print ("baseline correctness: ", sum(baseline_correctness) /
            len(baseline_correctness))
161     print ("proposed correctness: ", sum(proposed_correctness) /
            len(proposed_correctness))
162     print ("style check pass rate: ", sum(style_check) /
            len(style_check))
```

As seen above, we have defined two different evaluator functions. The style_evaluator() checks whether all components mentioned in the proposed method are implemented, and the output_evaluator() checks whether the model predictions are correct.

We provided this template along with instructions to tell the execution agent to modify all necessary functions for the provided idea. We tested the agent on 30 safety prompting ideas and 30 factuality prompting ideas.

As shown in Table 21, over half of the generated code files can be successfully executed, and the results indicate that only a handful of these proposed methods actually outperform the baselines. While this shows some promise that the execution agents can automatically implement and identify effective methods out of all the generated ideas, a careful manual check reveals many pitfalls.

|  | Safety | Factuality |
|---|---|---|
| Total | 30 | 30 |
| Executed | 17 | 18 |
| Passed Style Evaluator | 13 | 14 |
| Proposed Method Works Better | 5 | 1 |

Table 21: Success rate of our execution agent on 30 randomly sampled AI ideas on safety and factuality.

For example, the agent implemented the following baseline for the task of identifying malicious and adversarial prompts, which is a keyword-based approach that is trivial to beat with any LLM-based methods.

```python
## Step 2: Implement the baseline method
def baseline_method(client, model_name, seed, prompt):
    # Keyword-based heuristic
    keywords = ["ignore", "pretend", "roleplay", "disregard",
        "override"]
    for keyword in keywords:
        if keyword in prompt.lower():
            return "Adversarial"
    return "Benign"
```

In another example, for the same task of detecting adversarial prompts, the agent implemented the following evaluator function:

```python
## Step 5: Define the output evaluator
def output_evaluator(client, model_name, seed, input_text,
    gold_label, prediction):
    prompt = "Given the following text and reference sentiment
        classification, determine if the predicted classification is
        correct. Just tell me 'yes' or 'no', nothing else is
        needed.\n\nText: {}\n\nReference: {}\n\nPrediction:
        {}\n\n".format(input_text, gold_label, prediction)
    prompt_messages = [{"role": "user", "content": prompt}]
    response, _ = call_api(client, model_name, prompt_messages,
        temperature=0., max_tokens=1, seed=seed, json_output=False)

    judgment = False
    if response.strip().lower() == "yes":
        return True

    return judgment
```

The agent is supposed to inject adversarial triggers into sentiment classification data to test whether the proposed method can detect those adversarial prompts while maintaining sentiment classification accuracy. However, the agent only evaluates the accuracy on the original sentiment classification task but not the task of adversarial prompt detection.

Given these errors, we believe more work is needed to carefully verify the code implementations produced by the execution agent rather than blindly trusting their executed results, and we leave such attempts to future work.

