# OpenReview forum: "Can LLMs Generate Novel Research Ideas? A Large-Scale Human Study with 100+ NLP Researchers"
_ICLR.cc/2025/Conference — ICLR 2025 Poster_

### Official Review · Reviewer_r2Ms · 2024-10-30

**Soundness:** 2
**Presentation:** 3
**Contribution:** 3
**Rating:** 5
**Confidence:** 3

**Summary:**

This paper investigates the potential of LLMs to autonomously generate innovative research ideas comparable to those created by human experts.
In a controlled study, over 100 NLP researchers evaluated ideas generated by both humans and an LLM agent across metrics like novelty, excitement, feasibility, and effectiveness. The findings reveal that LLM-generated ideas were rated significantly more novel than human ideas, though slightly lower on feasibility.

The authors addressed multiple experimental challenges, including ensuring a balanced comparison between human and LLM ideas by standardizing topics and using blind reviews.
They also tested the robustness of results through various statistical methods. The LLM utilized a retrieval-augmented generation approach and scaled inference techniques to produce numerous ideas, from which a ranking mechanism selected the best ones. Despite the promising novelty scores, the study identifies several limitations of LLMs in research ideation, particularly a lack of diversity in ideas and challenges with reliable self-evaluation.

The study highlights open questions in designing research agents and emphasizes the potential and current limitations of LLMs in academic innovation.

**Strengths:**

1. The topic is interesting, as employing LLMs to generate research ideas holds practical value for researchers. Additionally, the findings offer valuable insights that can help researchers determine whether to adopt LLMs as part of their research toolkit.

2. The study’s experimental design is well-structured, effectively controlling for potential confounding factors. Through the use of blind reviews and standardized styles and topics, the authors achieve a fair and balanced comparison between ideas generated by humans and those by LLMs.

3. This paper incorporates a large-scale human evaluation, involving over 100 expert NLP researchers to create and assess research ideas. This approach provides a robust evaluation of the ideas across multiple metrics.

**Weaknesses:**

1. As noted in L434-L435, I appreciate that the authors acknowledge the subjectivity involved in reviewing ideas rather than executed papers. However, it would still be valuable to compare LLM-generated ideas with those from published papers (as ground truth), as the latter have been deemed at least ``feasible''.

2. According to L175-L179, the LLM agent was created with a minimalist approach, incorporating only paper retrieval, idea generation, and ranking. However, in real-world research, high-quality ideas often benefit from iterative refinement and prototyping, which are essential for improving clarity and feasibility. As such, I view this paper as an initial investigation into whether LLMs, in their current, straightforward form, can generate high-quality research ideas. The limitations observed, such as the lower feasibility of LLM-generated ideas, could potentially be addressed by a more sophisticated agent framework that includes stages for refining and prototyping ideas.

3. I would not consider this as a major weakness, but given the insights provided, it would be beneficial if the authors offered best practices or recommendations for using LLMs effectively to generate research ideas.

**Questions:**

See above.

---

> ### Author Response · Authors · 2024-11-21
> **Response to Reviewer r2Ms**
>
> Thank you for your insightful review! To address your concerns:
>
> - “it would still be valuable to compare LLM-generated ideas with those from published papers”
>
> To address your comment on evaluating the inter-reviewer agreement level with published papers, we compute the inter-reviewer agreement using the same consistency metric used in the paper for papers accepted by ICLR 2024, which only gets 52.8% consistency. This means that reviewers tend to have even lower agreement for their ranking on the accepted papers (since they might have similar quality in general).
>
> This further validates our point that reviewing is an inherently difficult task and it is expected that the inter-reviewer agreement would be relatively low. As we mentioned in the response to Reviewer HdAN, typical AI conferences such as NeurIPS and ICLR also have low inter-reviewer agreement (66.0% for NeurIPS’21). Moreover, our setting of evaluating ideas without the experiments is expected to involve even higher subjectivity than typical conference reviewing. That said, we believe we are getting meaningful signals from the review results since we are recruiting highly qualified experts for all the review tasks.
>
>
> - “The limitations observed, such as the lower feasibility of LLM-generated ideas, could potentially be addressed by a more sophisticated agent framework that includes stages for refining and prototyping ideas.”
>
> As we explained in the paper, our focus is on benchmarking the capabilities of current LLMs in comparison to a human expert baseline, rather than on building the strongest agent scaffolding for the idea generation task. We leave it to future work to explore many possible ways to improve upon our current agent design.
>
> - “it would be beneficial if the authors offered best practices or recommendations for using LLMs effectively to generate research ideas”
>
> Thank you for your suggestion! We will include a list of things that we tried and found effective, for example, we discovered that appending previously generated ideas and instructing the model to avoid duplicates can reduce repetition in idea generation; and that reranking is an effective strategy to find higher-quality ideas among all the generations.

---

> > ### Comment · Reviewer_r2Ms · 2024-11-26
> > **Response to the authors**
> >
> > Thank you for the authors' response. However, I would prefer to maintain my original score, as I believe some of my concerns will only be addressed in the next version.

---

### Official Review · Reviewer_YqJH · 2024-11-04

**Soundness:** 3
**Presentation:** 3
**Contribution:** 3
**Rating:** 6
**Confidence:** 3

**Summary:**

They constructed a meticulously controlled experiment comparing human-generated and LLM-generated ideas, thus overcoming the sample size and baseline issues present in previous small-scale evaluation studies. The study recruited 138 outstanding natural language processing researchers to generate human baseline ideas and conduct blind reviews (49 were responsible for writing ideas, and 79 for blind review). To reduce the impact of confounding factors, the study strictly controlled the style of the ideas and standardized the topic distribution between human and LLM-generated ideas. After a year-long high-cost evaluation, the authors provided a human expert baseline and a standardized evaluation protocol, laying a foundation for future related research.

In nearly 300 reviews, the authors found that under multiple hypothesis tests and different statistical tests, AI-generated ideas were considered more novel than those of human experts, but less feasible. In addition to evaluating the ideas, the study also analyzed the LLM agent, revealing its limitations and unresolved issues. Despite high hopes for expanding LLMs' reasoning abilities, the authors found that they still show deficiencies in the diversity of idea generation, and the results of human consistency analysis indicate that LLMs cannot yet serve as reliable evaluators of scientific idea quality.

In summary, the contributions of the paper are as follows:

1. Standardized some evaluation criteria for scientific ideas generated using large language models while demonstrating the limitations of LLMs as judges in evaluating scientific ideas.
2. Conducted a blind review evaluation of the quality of scientific ideas generated by LLMs through the recruitment of a large number of scientists, verifying both the potential and shortcomings of LLMs in the field of scientific research.

**Strengths:**

1. The results are quality assured due to significant investment. The author recruited a total of 138 excellent natural language processing researchers and verified their quality through Google Scholar. On average, each idea took the researchers 5.5 hours to generate, and there was a reward of $300. As for the evaluators, each evaluation opinion had a reward of $25. This incentive mechanism, to a certain extent, ensures the quality of idea generation and evaluation.
2. It is relatively fair. By standardizing the topics and providing human experts and large language models with the same writing templates, and finally passing both outputs through the same style transformation module for rewriting, the consistency of style between the two is ensured, minimizing human bias towards writing style as much as possible.
3. It is very meaningful. This large-scale manual evaluation verifies the ability of large language models to generate scientific ideas comparable to those of human experts, providing motivation for the development of subsequent work.

**Weaknesses:**

1. The author concluded that ideas generated by large language models lack diversity. However, using the same prompts, especially longer prompts, it is easy for large models to produce similar responses. Therefore, the lack of diversity may not be solely due to the models themselves; it could also be due to a lack of diversity in the prompts. The author needs to test more prompts to sufficiently reach such a conclusion for this experiment.
2. The paper only verifies the generation quality of the author's simple implementation of an agent, lacking evaluation against other baselines.
3. The research is limited to the NLP field and lacks studies in other domains.

**Questions:**

1. Could you provide the specific prompt used for the Agent to generate ideas?
2. Observing that some researchers in Table 3 and Table 4 have a familiarity level of 1 with the domain (which could be considered as low-quality annotation), what are the experimental results in Table 5 after removing this data?

---

> ### Author Response · Authors · 2024-11-21
> **Response to Reviewer YqJH (Part 1)**
>
> Thank you for your insightful review! To address your concerns:
>
> - "The author needs to test more prompts to sufficiently reach such a conclusion for this experiment."
>
> To examine whether the diversity issue exists across models and prompts, we further measure the idea diversity of multiple other base models and prompt setups. We use a temperature of 1.0 for all the generations and measure the percentage of non-duplicate ideas out of 2K generated ideas on the topic of uncertainty prompting.
>
> In the table below, we compare four different base models: Claude-3.5-Sonnet, GPT-4o, o1-mini, and Llama-3.1-405B-Instruct.
>
> |        |     Non-Duplicate (%)    |
> |-----------------------------------|-----------|
> | claude-3-5-sonnet        | 19.1%     |
> | gpt-4o                            | 59.5%     |
> | o1-mini                           | 22.6%     |
> | Llama-3.1-405B-Instruct           | 51.1%     |
>
> We find that different models have very different non-duplicate rates, and overall o1-mini and Claude-3.5-sonnet have the lowest diversity. However, we note that we picked Claude-3.5-sonnet as the base model of our agent because the quality of its generated ideas outperforms other models based on our pilot study, where we randomly sampled 10 ideas from Claude-3.5-sonnet and GPT-4o for a round of pilot expert scoring and Claude-3.5-sonnet scored an average of 5.4 while GPT-4o scored 4.8.
>
> Next, we try several different prompts for idea generation with the Claude-3.5-sonnet backbone: 1) no RAG and not appending previously generated ideas for deduplication; 2) using RAG but not appending previously generated ideas; 3) using RAG with different numbers of retrieved papers included in the prompt (k=5,10,20).
>
> |  |   Non-Duplicate (%)   |
> |-------------------|------|
> | no RAG; no prev   | 7.6% |
> | no RAG; prev      | 18.8%|
> | RAG (k=5); prev   | 18.4%|
> | RAG (k=10); prev  | 19.1%|
> | RAG (k=20); prev  | 19.4%|
>
> We find that appending previously generated ideas in the prompt and asking the model to avoid repetition can significantly reduce idea duplication. However, including retrieved papers in the prompt has minimal impact on the diversity. We thus conclude that Claude-3.5-sonnet is generally bad at idea diversity across various prompt setups.
>
> - "The paper only verifies the generation quality of the author's simple implementation of an agent, lacking evaluation against other baselines."
>
> As we explained in the paper, our focus is on benchmarking the capabilities of current LLMs in comparison to a human expert baseline, rather than building the strongest agent scaffolding for the idea generation task. We leave it to future work to explore many possible ways to improve upon our current agent design.
>
> - "The research is limited to the NLP field and lacks studies in other domains."
>
> We justify our choice to focus on prompting-based NLP research in the point of the general response. We believe the evaluation framework that we established would be helpful for future works that extend the evaluation to other domains.
>
> - "Observing that some researchers in Table 3 and Table 4 have a familiarity level of 1 with the domain (which could be considered as low-quality annotation), what are the experimental results in Table 5 after removing this data?"
>
> Note that out of all the reviews that we have collected, only two indicated a familiarity score of 1. Removing these two reviews does not impact any conclusions in the paper. Moreover, only one of the idea-writing participants indicated a familiarity of 1 on their topic. Removing their idea in the statistical tests gives the same conclusions.
>
> We show the results of removing the one participant with a familiarity of 1 and the two reviews with a familiarity of 1 below:
>
> |                 | Human | AI   | AI + Human Rerank |
> |-----------------|-------|------|-------------------|
> | Size (N)        | 115   | 108  | 109               |
> | Novelty         | 4.83  | 5.63 (p=0.00) | 5.81 (p=0.00) |
> | Excitement      | 4.56  | 5.18 (p=0.01) | 5.46 (p=0.00) |
> | Feasibility     | 6.63  | 6.33 (p=0.24) | 6.44 (p=0.42) |
> | Effectiveness   | 5.19  | 5.45 (p=0.24) | 5.55 (p=0.10) |
> | Overall         | 4.72  | 4.83 (p=1.00) | 5.34 (p=0.07) |
>
> Overall, all three main conclusions still hold robustly even if we remove all the ideas and reviews with a familiarity of 1.

---

> > ### Author Response · Authors · 2024-11-21
> > **Response to Reviewer YqJH (Part 2)**
> >
> > - "Could you provide the specific prompt used for the Agent to generate ideas?"
> >
> > The main prompt consists of: 1) the retrieved papers for the given topic; 2) the idea format template; 3) demo examples; and 4) previously generated ideas for deduplication. We provide the simplified version below, and we will open-source the entire agent implementation, which includes all the prompts used.
> >
> > ```
> > You are an expert researcher in AI. Now I want you to help me brainstorm some new research project ideas on the topic of: {topic description}.
> >
> > Here are some relevant papers on this topic just for your background knowledge:
> > {retrieved papers}
> >
> > The above papers are only for inspiration and you should not cite them and just make some incremental modifications. Instead, you should make sure your ideas are novel and distinct from the prior literature. You should aim for projects that can potentially win best paper awards at top AI conferences like ACL and NeurIPS. Each idea should be described as: (1) Problem: State the problem statement, which should be closely related to the topic description and something that large language models cannot solve well yet. (2) Existing Methods: Mention some existing benchmarks and baseline methods if there are any. (3) Motivation: Explain the inspiration of the proposed method and why it would work well. (4) Proposed Method: Propose your new method and describe it in detail. The proposed method should be maximally different from all existing work and baselines, and be more advanced and effective than the baselines. You should be as creative as possible in proposing new methods, we love unhinged ideas that sound crazy. This should be the most detailed section of the proposal. (5) Experiment Plan: Specify the experiment steps, baselines, and evaluation metrics.
> >
> > You can follow these examples to get a sense of how the ideas should be formatted (but don't borrow the ideas themselves):
> > {demo examples}
> >
> > You should make sure to come up with your own novel and different ideas for the specified problem. You should try to tackle important problems that are well recognized in the field and considered challenging for current models. For example, think of novel solutions for problems with existing benchmarks and baselines. In rare cases, you can propose to tackle a new problem, but you will have to justify why it is important and how to set up proper evaluation.
> > Please write down your 5 ideas (each idea should be described as one paragraph. Output the ideas in json format as a dictionary, where you should generate a short idea name (e.g., "Non-Linear Story Understanding", or "Multi-Agent Negotiation") as the key and the actual idea description as the value (following the above format). Do not repeat idea names or contents.
> > ```

---

> > > ### Comment · Reviewer_YqJH · 2024-12-03
> > >
> > > Thanks a lot for your response. I am inclined to maintain my positive score

---

### Official Review · Reviewer_HdAN · 2024-11-04

**Soundness:** 4
**Presentation:** 4
**Contribution:** 2
**Rating:** 6
**Confidence:** 4

**Summary:**

- Study: This paper presents a novel study comparing research ideas generated by Large Language Models (LLMs) with those crafted by expert NLP researchers.
- Methodology: The study involves hired expert researchers who generate ideas and conduct blind reviews of both human- and AI-generated ideas.
- Conclusion: This work is the first to conclude that AI-generated ideas are judged as significantly more novel than those from human experts (p < 0.05).

**Strengths:**

* Thorough Study Design: This study is detailed and carefully planned, taking a year to complete and requiring significant resources. It includes human expert baselines and a clear evaluation process, with strong statistical methods that make the results reliable.

* Key Conclusion: The study makes an important finding about the differences between AI-generated and human-generated research ideas, adding to our understanding of AI's potential in coming up with research ideas. — more discussions on the absolute quality, in addition to the comparison between AI and human, would be more apprieciated (e.g. are the quality of both ideas bad).

* Interesting Findings: The study also reveals some surprising insights, such as AI-generated ideas becoming repetitive as more ideas are created, and that current AI systems aren’t yet good at reliably evaluating research ideas

**Weaknesses:**

* While this work is interesting and presents important conclusions, it does not center on machine learning methodology or technical analysis. Although it is relevant and intriguing for the ML community, the paper leans more towards evaluating scientific idea generation, aligning more with applications rather than with core ML research.
* The study is limited by its exclusive focus on prompting-based NLP research, which restricts the generalizability of its findings to other fields or other ML directions.
* Although the authors worked to standardize evaluation criteria, research ideation remains inherently subjective, as reflected in the study's inter-reviewer agreement of 56.1%

**Questions:**

- The authors used reviews from Openreview, are there any licensing issues?

---

> ### Author Response · Authors · 2024-11-21
> **Response to Reviewer HdAN**
>
> Thank you for your insightful review! To address your concerns:
>
> - “Although it is relevant and intriguing for the ML community, the paper leans more towards evaluating scientific idea generation, aligning more with applications rather than with core ML research.”
>
> We believe our work is within the scope of ICLR, since “applications” is explicitly mentioned in the Call For Papers page (https://iclr.cc/Conferences/2025/CallForPapers).
>
> In fact, there are at least 10 submissions at this ICLR conference that focus on research idea generation (e.g., “Chain of Ideas: Revolutionizing Research in Idea Development with LLM Agents”, “Review and Rebuttal: Zero-shot In-context Adversarial Learning for Improving Research Ideation”, “Two Heads Are Better Than One: A Multi-Agent System Has the Potential to Improve Scientific Idea Generation”, “CycleResearcher: Improving Automated Research via Automated Review”, “GraphEval: A Lightweight Graph-Based LLM Framework for Idea Evaluation”, etc.).
>
> Despite being evaluation-centric, we believe our work is very relevant to the ICLR community and would be an important contribution to the emerging body of work on LLM for scientific research.
>
>
> - “The study is limited by its exclusive focus on prompting-based NLP research, which restricts the generalizability of its findings to other fields or other ML directions.”
>
> We address this in the first point of our general response.
>
> - “research ideation remains inherently subjective, as reflected in the study's inter-reviewer agreement of 56.1%”
>
> Subjectivity is the inherent nature of peer-reviewing. As we showed in the paper, typical AI conferences such as NeurIPS and ICLR also have low inter-reviewer agreement (66.0% for NeurIPS’21). Moreover, our setting of evaluating ideas without the experiments is expected to involve even higher subjectivity than typical conference reviewing. That said, we believe we are getting meaningful signals from the review results since we are recruiting highly qualified experts for all the review tasks.
>
> - “The authors used reviews from Openreview, are there any licensing issues?”
>
> According to the OpenReview terms of use (https://openreview.net/legal/terms), all submitters agree that their submissions/comments shall be released to the public under the Creative Commons Attribution 4.0 International (CC BY 4.0) license, which allows free sharing and adaptation.

---

> > ### Comment · Reviewer_HdAN · 2024-12-03
> >
> > Thank you for the response. I lean towards maintaining the rating. I believe the work is somewhere in between ML / HCI / general AI, maybe more towards the audience of HCI and general AI.
> >
> > For the licensing, I re-read the page and it says "the CC BY 4.0 license applies to comments and configuration records unless restricted by readability settings in the OpenReview API." So there is still some constraints on usage based on API.

---

### Official Review · Reviewer_3XGM · 2024-11-04

**Soundness:** 3
**Presentation:** 3
**Contribution:** 2
**Rating:** 6
**Confidence:** 4

**Summary:**

This paper evaluates if LLM based RAG agents are capable of generating novel research ideas for prompting techniques by comparing them with expert researchers. The key contributions of their work are

1) Evaluating the capabilities of LLMs to generating research ideas and analyzing their strengths (novelty) and weaknesses (feasibility and diversity)
2) They focus on removing confounders by conducting a large scale human evaluations and controlling experimental setting

**Strengths:**

1) Extensive human evaluation with over 100 NLP researchers
2) Great care taken to remove confounders through style normalization, review form, grant submission form for research ideas etc.
3) Interesting insights like lack of ability of LLMs to generate diverse research ideas.

**Weaknesses:**

1) Not strong enough motivation for restricting to prompting based research. This could have been a confounder since we did we are not sure about the ability of LLMs to generate research where the solutions are more complex.

2) Usefulness: The paper does not talk about the usefulness of the generated ideas research ideas, which is more important than novelty. Research ideas are born out of deep insights into the limitations of previous research. This paper neither gives the discussion/limitations section of research papers to the LM as context nor does it do any analysis of the usefulness of the generated ideas. Using abstracts and paper titles sounds like an anti-causal way of generating ideas which I think could affect the usefulness.

**Questions:**

1) RAG pipeline ablation tests: "We keep the top k = 20 papers from each executed function call and stop the action generation when a max of N = 120 papers have been retrieved." , "For retrieval augmentation, we randomly select k = 10 papers from the top-ranked retrieved papers" . Can you please provide some small scale (self-evaluated) tests ablating these parameters to ensure their optimality?

2) Researcher Background: I understand that the researcher familiarity numbers are provided for idea generation and reviewing processes, however it is slightly different from expertise. So, a better way to ensure the robustness of results is to look at the distribution of the expertise of the reviewers.

3) Interestingness: Can you please clarify the meaning of the term? Is it that the insight behind the proposed idea a non-trivial one or is it that the human evaluating it finds it personally interesting?

---

> ### Author Response · Authors · 2024-11-21
> **Response to Reviewer 3XGM**
>
> Thank you for your insightful review! To address your concerns:
>
>
> - "Not strong enough motivation for restricting to prompting based research."
>
> We address this in the first point of our general response.
>
> - "The paper does not talk about the usefulness of the generated ideas research ideas."
>
> We made our best attempt to evaluate the usefulness of these ideas by asking expert reviewers to score the ideas on feasibility and expected effectiveness as part of their evaluation. Notably, our effectiveness metric directly asks reviewers whether they think the proposed method would outperform existing baselines, which captures this usefulness aspect of the ideas.
>
> - "Can you please provide some small scale (self-evaluated) tests ablating these parameters to ensure their optimality?"
>
> For ablating the impact of the maximum number of papers to retrieve (N), we vary N among {30, 60, 120, 180} and measure the average relevance score of the top 20 retrieved papers (as judged by the LLM). We show results on the following two topics, where we report the average relevance score of the top 20 retrieved papers (on a scale of 1 - 10), as judged by an LLM. In general, N = 120 as used in our paper achieves good performance across different topics, and the relevance of the top retrieved papers tends to plateau after N = 120.
>
> | N   | Multilingual | Uncertainty |
> |-----|--------------|-------------|
> | 30  | 6.80         | 6.75        |
> | 60  | 7.05         | 7.50        |
> | 120 | 8.10         | 8.45        |
> | 180 | 8.10         | 8.40        |
>
> For ablating the impact of the number of papers to add to the prompt for RAG, we vary k between {0, 5, 10, 20} and measure how it impacts the diversity of the generated ideas on the topic of uncertainty prompting.
>
> |  |   Non-Duplicates (%)  |
> |--------------------|------|
> | k = 0              | 18.8%|
> | k = 5              | 18.4%|
> | k = 10             | 19.1%|
> | k = 20             | 19.4%|
>
> Overall, we can see that this hyper-parameter k has minimal impact on the diversity of the generated ideas.
>
> - "A better way to ensure the robustness of results is to look at the distribution of the expertise of the reviewers."
>
> When surveying participants' familiarity (lines 1146 - 1156 in Appendix), we defined familiarity as whether they have authored or read papers on their selected topic, which in a way captures their expertise. This, combined with the qualifications of these participants (as shown in Table 2), serves as evidence that our reviewers in general have strong expertise on their reviewed topics.
>
> - "Interestingness: Can you please clarify the meaning of the term?"
>
> When reviewing ideas, we used the review form as shown in A.7 (lines 1134 - 1270). “Interestingness” in this context (line 115) refers to the novelty and excitement aspects defined there.

---

> > ### Comment · Reviewer_3XGM · 2024-12-03
> >
> > Thanks a lot for the ablation tests. However, I am inclined to maintain my score since I am not fully convinced about the two major issues i.e.
> > 1) Usefulness: The fact that something outperforms the previous methods does not prove usefulness. Usefullness is what the scientific community can learn from a paper which is not conveyed through just
> >
> > 2) Restricting to prompting based ideas: The authors mention that prompting based ideas are the easiest to implement and execute. But I am not sure why that is necessary to determine if the research idea is novel or not.

---

### Author Response · Authors · 2024-11-21
**General Response**

We sincerely thank all the reviewers for their insightful reviews and constructive feedback. We notice a common question raised by the reviewers (3XGM, HdAN, and YqJH): why do we restrict to prompting-based NLP research for the study? We address this concern below.

As we explained in our paper (lines 118-122), our motivation for restricting to prompting-based NLP research is twofold.

First, prompting-based research requires minimal computing hardware and is generally easy to implement and execute compared to many other types of AI research. We intend for our ideation study and the data / ideas generated from this to serve as a useful set of resources and a foundation from which to study the excitability of AI-generated ideas, as well as automated research execution agents. Therefore the feasibility of execution is important for facilitating such future work.

Second, prompting is an active area of research, and many prompting-based research papers have been published at AI/NLP/ML conferences in recent years. For example, in this current ICLR conference itself, 1194 submissions mentioned prompting in the title or abstract. Successful prompting research (such as chain-of-thought prompting) has had a big impact on the research community.

We believe our study in its current scope and design has substantial value for the community. Future work can take our evaluation framework and extend it to other research domains.

---

### Meta-Review · Area_Chair_YMqt · 2024-12-19

**Metareview:**

**Summary**

This paper aims to study whether LLMs can be scientifically creative and then evaluates if LLM based RAG agents are capable of generating novel research ideas for prompting techniques by comparing them with expert researchers.
This paper addresses the question that we all don't want to be answerd as discussed in [1].


**Strengths**

The extensive analysis made using NLP researchers

**Weaknesses**
Clearly, this kind of paper has many weaknesses as we don't want LLMs to succed in this task.
- The usefulness of the generated ideas does not seems to be proven (reviewer *3XGM* is not convinced)


**Final remarks**

The paper touches important issues.

** References **

[1] GASP! Generating Abstracts of Scientific Papers from Abstracts of Cited Papers

**Additional Comments On Reviewer Discussion:**

The active discussion has clarified many points.

---

### Decision · Program_Chairs · 2025-01-22

Accept (Poster)